# Generator-Mediated Bandits: Thompson Sampling for GenAI-Powered Adaptive Interventions

**Marc Brooks**[1],[*]  **Gabriel Durham**[1],[*]  **Kihyuk Hong**[1],[*]  **Ambuj Tewari**[1]

[1]Department of Statistics, University of Michigan, Ann Arbor, MI (USA)

`{marcbr, gjdurham, kihyukh, tewaria}@umich.edu`

## Abstract

Recent advances in generative artificial intelligence (GenAI) models have enabled the generation of personalized content that adapts to up-to-date user context. While personalized decision systems are often modeled using bandit formulations, the integration of GenAI introduces new structure into otherwise classical sequential learning problems. In GenAI-powered interventions, the agent selects a query, but the environment experiences a stochastic response drawn from the generative model. Standard bandit methods do not explicitly account for this structure, where actions influence rewards only through stochastic, observed treatments. We introduce *generator-mediated bandit–Thompson sampling* (GAMBITTS), a bandit approach designed for this action/treatment split, using mobile health interventions with large language model-generated text as a motivating case study. GAMBITTS explicitly models both the treatment and reward generation processes, using information in the delivered treatment to accelerate policy learning relative to standard methods. We establish regret bounds for GAMBITTS by decomposing sources of uncertainty in treatment and reward, identifying conditions where it achieves stronger guarantees than standard bandit approaches. In simulation studies, GAMBITTS consistently outperforms conventional algorithms by leveraging observed treatments to more accurately estimate expected rewards.

## 1 Introduction

Bandit algorithms are ubiquitous in the design of personalized interventions as they provide a flexible framework for learning from context and outcomes over time. Recently, researchers have begun integrating generative artificial intelligence (GenAI) models into these systems to enable richer forms of personalization in areas such as healthcare, e.g., patient care and mobile health (mHealth) [2, 23], personalized education [24], music [48], and marketing [28]. By generating rich, tailored content on-the-fly, GenAI integration expands the scope of personalization beyond what fixed libraries can offer, since predefined intervention options are unable to accommodate the full range of user contexts encountered in practice. While this real-time generative capability opens new possibilities for intervention design, it disrupts the action/reward feedback loop that underpins traditional bandit learning. Rather than directly acting on the environment, the agent selects a query—e.g., a prompt to a large language model (LLM)—and the environment receives the stochastic output generated in response to that query, which we refer to as the "treatment." This separation between the agent's action and the delivered treatment introduces additional stochasticity into the intervention process and motivates the need for algorithms that explicitly account for the generative mechanism.

Broadly, we focus on online decision-making problems where the treatment delivered to the environment is a random function of the agent's action. As a motivating case study, we consider the use of LLMs in sequential mHealth interventions. Traditionally, mHealth systems deliver text-based

---

[*]Equal contribution.

39th Conference on Neural Information Processing Systems (NeurIPS 2025).

support using data collected from smartphones and wearables, adapting to rapid fluctuations in user context, physiology, and behavior [39, 50]. Bandit algorithms have become a workhorse for this sequential personalization, with personalized Just-in-Time Adaptive Interventions (pJITAIs) serving as a popular setting in which such methods are developed and evaluated [51, 33, 14].

This framework for mHealth intervention has inspired countless recent advances in bandit algorithms; e.g., allowing partial pooling of information across users [19, 52, 20], accommodating non-stationarity to adapt to shifting user behavior patterns [33], and integrating high-dimensional covariates [3]. While these advancements further optimize bandit learning within this intervention framework, such approaches are fundamentally limited by reliance on a fixed library of text intervention options.

Dynamic message generation via LLMs offers a path forward. Throughout our analyses, we assume access to a generator whose responses are guaranteed to be safe and suitable for delivery (see Section 7). In this setting, the agent selects a query based on user context, submits both query and user context to the LLM, and delivers the generated text response as the treatment. The user then produces a reward signal (e.g., behavior or engagement). Crucially, while the agent selects the query, it cannot control the LLM's response.

This setup motivates a bandit framework where the agent's action affects the environment only through a high-dimensional, stochastic treatment. Standard bandit algorithms are not designed to exploit this indirect treatment mechanism, leading to inefficient learning by failing to make use of the structure linking treatments to rewards. Section 3 more precisely defines this *generator-mediated bandit* (GAMBIT) framework.

We develop a Thompson sampling-based framework tailored to this action/treatment split, incorporating explicit models of the treatment and reward generation processes. This framework serves as the backbone of our approach due to its central role in pJITAI designs, which often rely on Thompson sampling both for its theoretical guarantees and for its compatibility with downstream causal analyses using logged data, a key consideration in mobile health research [33]. We call the proposed approach *generator-mediated bandit–Thompson sampling* (GAMBITTS).

GAMBITTS selects actions by reasoning over both the distribution of treatments induced by each query and the reward those treatments generate, using a two-stage sampling procedure that captures uncertainty across the intervention process. In settings where the agent has simulation access to the treatment-generating mechanism, this structure enables offline estimation of the treatment model, accelerating online adaptation. To address the high dimensionality of treatments in generative settings (e.g., LLM-generated text), GAMBITTS projects each observed treatment into a fixed low-dimensional representation, enabling scalable and sample-efficient learning.

This work responds to the core challenge of learning when actions yield stochastic, observed treatments. The primary contributions are: $(a)$ the GAMBITTS framework for bandit learning with stochastic treatments, $(b)$ algorithmic instantiations of GAMBITTS that vary in their approach to modeling the treatment-generation process, ranging from fully online updates to approaches that pretrain the treatment model, as well as ensemble-based variants that support nonlinear reward modeling, $(c)$ theoretical regret bounds for GAMBITTS, including decompositions that isolate contributions from treatment and reward uncertainty, showing when modeling treatment generation leads to improved learning efficiency, and extending guarantees to flexible, nonlinear reward models, $(d)$ empirical results demonstrating the limitations of standard methods and the performance gains achieved by GAMBITTS-based approaches.

## 2 Related Work

In the bandit literature, researchers have illustrated how leveraging the causal structure underlying sequential interventions can improve policy learning [30, 37, 35]. Subsequent work has supported this conclusion, developing bandit algorithms in a wide variety of causal structures [57, 56]. These approaches model a set of binary, interconnected random variables $\mathbf{X} = \{X_1, \ldots, X_N\}$ with general causal pathways to a reward $Y$. Actions are framed as interventions on subsets of $\mathbf{X}$, and algorithms are aimed at optimizing reward in settings where the action space corresponds to modifying causally linked features that influence $Y$. In parallel, Sen et al. [2017] and Maiti et al. [2022] explore related environments where unobserved confounding can impact bandit learning [45, 38].

While the works discussed above emphasize general causal structures with limited treatment representations, our focus is the inverse: a specific mediation structure, where actions influence reward through an observed stochastic intermediate, but with rich, high-dimensional treatments (i.e., the intermediate generative output). This designed mediation, where the effect of the agent's action operates through the generator's stochastic response, parallels the instrumental variable (IV) framework, in which an instrument affects outcomes only indirectly via a stochastic channel [22]. Recent work has connected IV estimation and bandit design in classical econometric settings [58, 9], while related research has explored similar structures under noncompliance, where the realized action may differ from the one selected by the agent [49, 29]. Kallus [2018] makes this connection explicit through the *instrument-armed bandit* framework [25]. Most recently, Zou et al. [2025] study mediated bandit environments from a psychometric perspective [59].

Appendix A discusses further connections and distinctions between our framework and the methods discussed above, along with recent work on LLMs and bandits and related, application-focused, literature from the mHealth community.

## 3 Problem Formulation and Notation

We consider a sequential decision-making setting in which, at each time $t = 1, \ldots, T$, an agent: $(i)$ observes a context $X_t \in \mathcal{X} \subseteq \mathbb{R}^{d_C}$, $(ii)$ selects an action $A_t \in \mathcal{A}$, where $|\mathcal{A}| = K \in \mathbb{N}$, $(iii)$ sends $(A_t, X_t)$ to a generator, $(iv)$ receives response $G_t \in \mathcal{G}$ and delivers $G_t$ to the environment, and $(v)$ observes reward $Y_t \in \mathbb{R}$. The goal of the agent is to learn a policy for choosing actions that maximizes cumulative expected reward over time.

As a concrete example, consider an mHealth intervention designed to encourage physical activity. At time $t$, the agent: $(i)$ observes user context/covariates $X_t$ (e.g., steps taken the previous day and current location), $(ii)$ selects an action $A_t$, corresponding to a query from a finite set of options (e.g., "Please write a message that encourages the user to walk more; I have attached the user's current location and previous day's step count"), $(iii)$ sends the query, along with the context, to an LLM, $(iv)$ delivers the LLM's response, $G_t$, to the user, and $(v)$ observes reward $Y_t$ (e.g., user's subsequent step count).

In our motivating example, $G$ represents the output of an LLM, and the treatment space $\mathcal{G}$ consists of natural language text. In such GenAI settings, $\mathcal{G}$ is exceedingly high-dimensional, making it natural to posit the effect of $G$ on $Y$ occurs through a lower-dimensional embedding $Z^* \in \mathcal{Z}^* \subseteq \mathbb{R}^{d^*}$, which serves as a sufficient representation of the treatment for predicting reward. Recent work in high-dimensional causal inference supports this assumption, showing that such lower-dimensional embeddings can yield representations sufficient for treatment effect estimation (see, e.g., Veitch et al. [2019]) [53]. Because $Z^*$ is not observed in practice, the analyst must instead specify a working embedding $h(G) = Z \in \mathcal{Z} \subseteq \mathbb{R}^d$. We discuss strategies for constructing $h$ in Section 4. Furthermore, in practice, the agent may also restrict which components of $X_t$ are passed to the generator; we discuss this case in Appendix B.1.

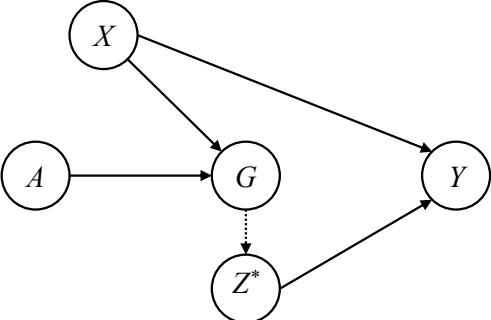

Figure 1: Generator-Mediated Bandit Causal Structure

*Notes: (i) Dotted lines represent deterministic relationships.*
*(ii) Arrows from X to A are omitted to represent the data-generating process rather than the decision logic.*
*(iii) A: Action/Query; X: Full Context; G: Generated Response; $Z^*$: True Response Embedding; Y: Reward.*

We refer to the resulting learning setup as a *generator-mediated bandit* (GAMBIT) problem. This causal structure is summarized in the directed acyclic graph (DAG) in Figure 1.

# 4    Methodology: Generator-Mediated Bandit–Thompson Sampling

As discussed above, we seek a Thompson sampling-based framework for our bandit agent, in line with its prevalence in mHealth interventions. We observe two key conditional structures to model:

$$Z \mid A, X, \text{ and } \mathbb{E}\left[Y \mid Z, X\right],$$

which correspond to the treatment and reward models. Modeling these two components comprises the two stages of the proposed Thompson sampler and induces the following parametrized environment:

$$\text{Treatment model: } Z_t \sim f_1\left(z; A_t, X_t, \theta_1\right) \quad \text{Treatment prior: } \theta_1 \sim \pi_1\ ,$$
$$\text{Reward model: } \mathbb{E}\left[Y_t \mid Z_t, X_t\right] = m_2\left(Z_t, X_t; \theta_2\right) \quad \text{Reward prior: } \theta_2 \sim \pi_2\ .$$

The construction of working treatment representation $Z = h(G)$ depends on the application domain. In marketing applications, for instance, $G$ may represent AI-generated images, while in our motivating mHealth setting, $G$ corresponds to LLM-generated text, necessitating different approaches to representation. For implementation, all algorithmic steps operate on the observed working embedding $Z$, not the unobserved benchmark $Z^*$. While the GAMBITTS framework is agnostic to the form of $h$, the choice can substantially influence empirical performance. The effectiveness of the GAMBITTS projection approach depends on how well $Z$ preserves reward-relevant information; when $Y \perp\!\!\!\perp Z^* \mid Z, X$, modeling $Y \mid Z, X$ captures the full signal in $Z^*$ (and thus that in $G$ as well). As the conditional dependence $Y \mid Z, X$ diverges from $Y \mid Z^*, X$, performance may degrade. We explore this phenomenon empirically in Section 6 and further discuss robustness to embedding misspecification in Appendix C. In our motivating setting, many projection strategies have been proposed for natural language-based interventions; see, e.g., [18, 53, 43, 11, 21, 54].

We propose two general approaches for GAMBITTS implementation: a *fully online* variant that updates both treatment and reward models sequentially, and a *partially online* variant that pretrains the treatment model using offline data and updates only the reward model during deployment. To improve flexibility and accommodate nonlinear reward relationships, we extend both variants with the ensemble-based sampling strategies introduced in Lu and Van Roy [2017] [36].

The algorithms below focus on settings where the decision to intervene (e.g., to send the user a text message) has already been made. In practice, mHealth interventions often include an initial decision of whether to send a message at all; GAMBITTS can be trivially extended to accommodate this preliminary selection step, as discussed in Appendix B.2.

## 4.1    Fully Online Generator-Mediated Bandit–Thompson Sampling

Let $\mathcal{D}_t = \{X_s, A_s, G_s, Y_s\}_{s=1}^{t-1}$ represent the data collected up to time $t$. Algorithm 1 below describes a fully online approach for selecting action $A_t$ based on $\mathcal{D}_t$.

---

**Algorithm 1** `Fully Online GAMBITTS (foGAMBITTS)`

---

    **Inputs**: Data $\mathcal{D}_t$, priors $\pi_1, \pi_2$, models $f_1, m_2$, current context $x_t$.
1: **Derive** posterior distributions $P_1\left(\theta_1 \mid \mathcal{D}_t\right)$ and $P_2\left(\theta_2 \mid \mathcal{D}_t\right)$.
2: **Sample** $\theta_{1,t} \sim P_1\left(\theta_1 \mid \mathcal{D}_t\right)$ and $\theta_{2,t} \sim P_2\left(\theta_2 \mid \mathcal{D}_t\right)$.
3: **For** each $a \in \mathcal{A}$, calculate

$$\mathbb{E}_{\theta_{1,t}, \theta_{2,t}}\left[Y \mid a, x_t\right] = \int_{\mathcal{Z}} m_2\left(z, x_t; \theta_{2,t}\right) f_1(z; a, x_t, \theta_{1,t}) dz.$$

4: **Set** $A_t = \underset{a' \in \mathcal{A}}{\arg\max}\, \mathbb{E}_{\theta_{1,t}, \theta_{2,t}}\left[Y \mid a', x_t\right].$

---

The integral in Step 3 can often be intractable in closed form, but can be efficiently approximated via Monte Carlo approaches. Furthermore, because GAMBITTS selects agents via posterior sampling, it

induces well-defined randomization probabilities that can be used for downstream causal inference or offline evaluation. Specifically, the probability of selecting action $a$ at time $t$ is

$$p_{t,a}^* = \int_{\Theta_2} \int_{\Theta_1} \mathbb{1} \left\{ a = \arg\max_{a' \in \mathcal{A}} \mathbb{E}_{\theta_1, \theta_2} \left[ Y \mid a', x_t \right] \right\} P_1 \left( \theta_1 \mid \mathcal{D}_t \right) P_2 \left( \theta_2 \mid \mathcal{D}_t \right) d\theta_1 d\theta_2.$$

In settings where these probabilities are required for post hoc causal analyses (e.g., analysis of pJITAIs), the full set $\left\{ p_{t,a}^* \right\}_{a \in \mathcal{A}, t=0,\dots,T}$ can be approximated retrospectively, after the study period, when additional computational resources are available [4, 33]. Alternatively, if sufficient computing power is available at deployment time, the agent may compute these probabilities online and sample from them directly. As with the integral in Step 3, the quantities $p_{t,a}^*$ can be estimated via Monte Carlo sampling from the posterior distributions over $\theta_1$ and $\theta_2$.

## 4.2 Partially Online Generator-Mediated Bandit–Thompson Sampling

In settings where the agent has simulation access to the generator, or access to additional data sources, it may be possible to learn the treatment model independently of user interactions. For example, in our motivating application, an mHealth intervention designer could query an LLM multiple times without sending any generated text to users.

This capability motivates a *partially online GAMBITTS* (`poGAMBITTS`). The `poGAMBITTS` framework exploits simulation access to the generator to offload learning of the first-stage distribution, $Z \mid A, X$ to an offline phase. Empirically, this decomposition can substantially improve online performance, as discussed in Section 6.

Here, we consider the generator $G$ to be a random function, which the agent can query using prompt-context pairs $(a, x) \in \mathcal{A} \times \mathcal{X}$. In the `poGAMBITTS` framework, the agent can obtain an offline approximation of the treatment distribution $f_1$ by: $(i)$ simulating $M_{off}$ prompt-context pairs $\left\{ \left( a^1, x^1 \right), \dots, \left( a^{M_{off}}, x^{M_{off}} \right) \right\}$, $(ii)$ using the generator to simulate $G(a^i, x^i)$, with $g^i(a^i, x^i)$ denoting the observed output and $z^i$ denoting its observed $\mathcal{Z}$ embedding, for $i = 1, \dots, M_{off}$, and $(iii)$ taking $f_1^{off}(z; A_t, X_t)$ to be the empirical conditional distribution observed in $\mathcal{D}^{off} := \left\{ \left( a^i, x^i, z^i \right) \right\}_{i=1}^{M_{off}}$.

In cases where $|\mathcal{X}|$ is finite, the agent can repeatedly prompt the generator for each prompt-context pair $(a, x) \in \mathcal{A} \times \mathcal{X}$ and directly use the empirical distribution of $Z \mid A, X$ in $\mathcal{D}^{off}$ as $f_1^{off}$. When $|\mathcal{X}|$ is infinite, the agent must instead fit a model to estimate the conditional distribution of $Z \mid A, X$ based on $\mathcal{D}^{off}$, and use this estimate as the treatment model.

---

**Algorithm 2** Partially Online GAMBITTS (`poGAMBITTS`)

---

    **Inputs**: Data $\mathcal{D}_t$, prior $\pi_2$, model $m_2$, current context $x_t$, pretrained model $f_1^{off}$.
1: **Derive** the posterior distribution $P_2 \left( \theta_2 \mid \mathcal{D}_t \right)$.
2: **Sample** $\theta_{2,t} \sim P_2 \left( \theta_2 \mid \mathcal{D}_t \right)$.
3: **For** each $a \in \mathcal{A}$, calculate

$$\mathbb{E}_{\theta_{2,t}} \left[ Y \mid a, x_t \right] = \int_{\mathcal{Z}} m_2 \left( z, x_t; \theta_{2,t} \right) f_1^{off}(z; a, x_t) dz.$$

4: **Set** $A_t = \arg\max_{a' \in \mathcal{A}} \mathbb{E}_{\theta_{2,t}} \left[ Y \mid a', x_t \right]$.

---

As in `foGAMBITTS`, the probability of selecting action $a$ at time $t$ can be written explicitly, here as

$$\int_{\Theta_2} \mathbb{1} \left\{ a = \arg\max_{a' \in \mathcal{A}} \mathbb{E}_{\theta_2} \left[ Y \mid a', x_t \right] \right\} P_2 \left( \theta_2 \mid \mathcal{D}_t \right) d\theta_2,$$

which can be computed or approximated retrospectively if randomization probabilities are needed for post hoc causal analyses. Again, we can approximate $\mathbb{E}_{\theta_{2,t}} \left[ Y \mid A_t = a, X_t = x_t \right]$ via Monte Carlo.

Moreover, one may wish to blend the approaches in Algorithms 1 and 2 by using simulated offline data to form a posterior distribution over the treatment model parameters, $P\left(\theta_1 \mid \mathcal{D}^{off}\right)$. During online deployment, the agent may either continue updating this posterior online, or fix it and sample throughout learning (avoiding the computational cost of full online $\theta_1$ posterior updates).

### 4.3   Ensemble-Based GAMBITTS Approaches

Algorithms 1 and 2 require explicit posterior updates for reward model parameters ($\theta_2$), restricting the class of models that can be feasibly used. This restriction can be particularly limiting for two reasons. First, flexible models are often desirable to avoid strong parametric assumptions on $\mathbb{E}\left[Y \mid Z, X\right]$. Second, when $Z$ is an imperfect proxy for $Z^*$, flexible models may help capture residual information and improve the approximation of $\mathbb{E}\left[Y \mid Z^*, X\right]$ through $m_2(Z, X)$.

To address these challenges, we propose an ensemble-based posterior approximation strategy, drawing on the ensemble sampling framework introduced in Lu and Van Roy [2017] [36]. This approach maintains an ensemble of models to approximate the posterior distribution over parameters, making posterior sampling tractable even for complex or nonlinear models. We present ensemble-based variants of Algorithms 1 and 2 (`ens-foGAMBITTS` and `ens-poGAMBITTS`, respectively) in Appendix D. In Section 6, we evaluate the performance of `ens-poGAMBITTS` using a multilayer perceptron (MLP) model for $\mathbb{E}\left[Y \mid Z, X\right]$.

## 5   Theoretical Guarantees

This section explores regret guarantees for the GAMBITTS algorithms introduced in Section 4. We revisit the generator-mediated bandit learning problem from Section 3, now from a theoretical perspective. Throughout, we assume a finite context space (i.e., $|\mathcal{X}| = C < \infty$) and focus on the correctly specified setting where the working treatment projection matches the true underlying treatment embedding (i.e., $Z = Z^*$).

Under these assumptions, treatment and reward generation at time $t$ follows a two-stage stochastic process. First, given context $x_t$ and action $a_t$, the generator samples an intermediate treatment variable $z_t \in \mathbb{R}^d$ from a distribution $\xi_{x_t,a_t}^{\theta_1}$ (with density $f_1\left(z; x_t, a_t, \theta_1\right)$) supported on the ball $\mathbb{B}_d(B) = \{z \in \mathbb{R}^d : \|z\|_2 \le B\}$. Second, the agent receives a noisy reward $y_t = m_2(z_t, x_t; \theta_2) + \eta_t$, where $\eta_t$ is $\sigma_2$-subgaussian noise. Here, $\theta = \{\theta_1, \theta_2\}$ are unknown parameters. We also assume that the mean reward distribution $m_2\left(Z, x; \theta_2\right)$, under $Z \sim \xi_{x,a}^{\theta_1}$, is $\sigma_1$-subgaussian for all $(x, a) \in \mathcal{X} \times \mathcal{A}$.

The goal of the agent is to minimize the time-$T$ cumulative Bayesian regret, defined as

$$
\mathrm{BR}_T := \mathbb{E}\left[\sum_{t=1}^{T}\left(\mathbb{E}_{\xi_{X_t,A_t^*}^{\theta_1}}\left[m_2(Z, X_t; \theta_2) \mid X_t, A_t^*, \theta\right] - \mathbb{E}_{\xi_{X_t,A_t}^{\theta_1}}\left[m_2(Z, X_t; \theta_2) \mid X_t, A_t, \theta\right]\right)\right],
$$

where the inner expectations are over $Z$ and the outer expectations over $X$, $\theta_1$, $\theta_2$, and agent actions $A$. The benchmark $a_t^*$ denotes the Bayes-optimal action in context $x_t$, defined by $A_t^* := \arg\max_{a \in \mathcal{A}} \mathbb{E}_{\xi_{x_t,a}^{\theta_1}}\left[m_2(Z, x_t; \theta_2) \mid \theta\right]$.

As a baseline, we consider a Thompson sampling agent that models the problem as a conventional multi-armed contextual bandit over $\mathcal{X} \times \mathcal{A}$, ignoring the generator-mediation structure. We refer to this as a "standard" Thompson sampling approach, as it captures a natural modeling choice that treats rewards as a direct consequence of actions. This serves as a point of comparison in both our theoretical and empirical results. As shown in Appendix E, its Bayesian regret satisfies

$$
\mathrm{BR}_T^{standard} \le \widetilde{\mathcal{O}}\left((\sigma_1 + \sigma_2)\sqrt{CKT}\right) \tag{1}
$$

While the regret bound for the standard Thompson sampling agent is tight (i.e., matching known lower bounds), it becomes unfavorable when either the context or action space is large. This motivates the GAMBITTS approach, which can exploit shared structure in the treatment space to support generalization across arms. We begin by analyzing `poGAMBITTS`, as this approach aligns more closely with our motivating setting, then turn to the `foGAMBITTS` variant. Appendix E presents the proofs of all results discussed below, along with additional theoretical results.

## 5.1 Regret Analysis for `poGAMBITTS`

We begin by analyzing the regret of `poGAMBITTS`. This algorithm uses empirical conditional distributions, with densities $f_1^{off}(z; x, a)$, that approximate the true treatment distributions $\xi_{x,a}^{\theta_1}$ and define the estimated mean reward

$$\widehat{m}_2(x, a; \theta_2) = \int_{\mathcal{Z}} m_2(z, x; \theta_2) f_1^{off}(z; x, a) dz.$$

`poGAMBITTS` can be viewed as a Thompson sampling algorithm operating on the estimated mean reward function $\widehat{m}_2$. More generally, we can consider other traditional bandit algorithms operating in the "partially online stochastic treatment" setting, where actions are selected based on an estimated reward model derived from an empirical treatment distribution. Although actions are chosen with respect to $\widehat{m}_2$, rewards are generated according to the true mean $m_2$, introducing a form of model misspecification. Nevertheless, we show that if the empirical treatment model is sufficiently close to the true distribution, then running a no-regret contextual bandit algorithm on $\widehat{m}_2$ still yields no-regret guarantees for the original problem.

In the following results, we let $\varepsilon > 0$ and consider empirical treatment models $f_1^{off}$ satisfying

$$KL\left(f_1^{off}(\cdot; x, a) \big\| f_1(.; x, a, \theta_1)\right) \leq \varepsilon \quad \text{for all } (x, a) \in \mathcal{X} \times \mathcal{A}. \tag{2}$$

As shown in Appendix E, under regularity conditions on $\xi_{X,A}^{\theta_1}$, $f_1^{off}$ will satisfy (2) given $\text{poly}\left(d, \varepsilon^{-1}\right)$ draws from the simulator for each $(x, a) \in \mathcal{X} \times \mathcal{A}$.

**Theorem 1.** *Let $f_1^{off}$ be an empirical model for the treatment distribution satisfying (2). Let Alg be a contextual bandit algorithm that selects actions based on the estimated reward function $\widehat{m}_2$. Then, running Alg in the partially online stochastic treatment setting gives $T$-step Bayesian regret $BR_T^{Alg}$, with*

$$BR_T^{Alg} \leq \mathcal{O}\left(\widehat{BR}_T^{Alg} + T\sqrt{\varepsilon T}\right),$$

*where $\widehat{BR}_T^{Alg}$ is the $T$-step Bayesian regret of Alg with respect to $\widehat{m}_2$.*

Theorem 1 bounds the agent's true regret in terms of its regret under the estimated reward model induced by its misspecified treatment distribution. For `poGAMBITTS` with a linear reward model, we get the following result.

**Corollary 1.** *Let $f_1^{off}$ satisfy (2). If $m_2(z, x_t; \theta)$ is linear in $z \in \mathbb{R}^d$, then the Bayesian regret of `poGAMBITTS` (Algorithm 2) satisfies $BR_T^{po:lin} \leq \widetilde{\mathcal{O}}\left(\sigma_2 d\sqrt{T} + T\sqrt{\varepsilon T}\right).$*

When $\varepsilon \leq (\sigma_2 d/T)^2$, the regret simplifies to $\widetilde{\mathcal{O}}\left(\sigma_2 d\sqrt{T}\right)$. This is sharper than the standard Thompson sampling bound whenever $d < \left(1 + \frac{\sigma_1}{\sigma_2}\right)\sqrt{CK}$.

We now consider `poGAMBITTS` with a nonlinear reward model. In Theorem 2, the complexity measures $\dim_E(\mathcal{F}, T^{-2})$ and $\mathcal{N}(\mathcal{F}, T^{-2}, \|\cdot\|_\infty)$ denote the eluder dimension and the covering number of $\mathcal{F}$, respectively. Appendix E provides formal definitions of these measures.

**Theorem 2.** *Let $f_1^{off}$ satisfy (2). For any function class $\mathcal{F}$ with $\widehat{m}_2 \in \mathcal{F}$, the Bayesian regret of `poGAMBITTS` (Algorithm 2) satisfies*

$$BR_T^{po:\mathcal{F}} \leq \widetilde{\mathcal{O}}(\sigma_2\sqrt{\dim_E(\mathcal{F}, T^{-2})\log\mathcal{N}(\mathcal{F}, T^{-2}, \|\cdot\|_\infty)T} + T\sqrt{\varepsilon T}).$$

## 5.2 Regret Analysis for `foGAMBITTS`

We now present regret guarantees for `foGAMBITTS` (Algorithm 1) in the linear reward setting, which applies when the agent does not have simulator access to the distributions $\{\xi_{x,a}\}_{(x,a)\in\mathcal{X}\times\mathcal{A}}$.

**Theorem 3.** *If $m_2(z, x_t; \theta)$ is linear in $z \in \mathbb{R}^d$, the Bayesian regret of `foGAMBITTS` (Algorithm 1) satisfies*

$$BR_T^{fo:lin} \leq \widetilde{\mathcal{O}}\left(\sigma_1\sqrt{CKT} + \sigma_2 d\sqrt{T}\right).$$

The bound nearly matches the minimax lower bound of $\Omega(\max\{\sigma_1\sqrt{CKT}, \sigma_2\sqrt{dT}\})$, presented in Appendix E. Comparing Theorem 3 and Equation 1 shows `foGAMBITTS` achieves a sharper regret bound than standard Thompson sampling whenever $d \ll \sqrt{CK}$. When the working treatment representation is low-dimensional and well-specified, the benefits from generalizing across arms outweigh the cost of estimating the treatment distribution. Moreover, the advantage grows when $\sigma_2 \gg \sigma_1$, since generalizing across arms helps more when reward noise dominates. In addition, when $\sigma_1 < 1/\sqrt{CK}$, the regret of `foGAMBITTS` becomes comparable to that of `poGAMBITTS`. In such cases, where the generator is sufficiently concentrated, simulator access offers little improvement.

Having established regret guarantees for the GAMBITTS algorithms, we now turn to their empirical evaluation in Section 6.

# 6    Simulation Results

To evaluate GAMBITTS-based algorithms, we designed a simulation study modeled on the 2023 Intern Health Study (IHS) which was aimed at supporting mental health among medical interns [40]. At each decision point, the agent $(i)$ observes user context (location, recent step count), $(ii)$ selects a prompt from a finite list to submit to an LLM (Llama 3.1 8.0B), and $(iii)$ delivers the generated text response as the intervention [15]. The environment then generates a reward based on select semantic dimensions of the generated text (optimism, severity, formality, clarity, and encouragement). The semantic dimensions were constructed by training one-dimensional variational autoencoders on text generated to vary along each axis, yielding mappings from text to scores used in the reward model (see Appendix G.3 for more details).

While no deployed JITAIs currently integrate LLMs for real-time message generation (and thus no real-world dataset exists for this setting), we calibrate our simulation using empirical distributions from the 2023 IHS to model realistic conditional reward structures. Details on the compute resources, generative model, prompt design, and reward specification are provided in Appendix G, with code and further documentation available on GitHub. Additionally, Appendix F includes further simulations exploring different first- and second-stage variance decompositions, varying the $\mathcal{Z}$ dimensionality $d$, altering the level of simulation access available to `poGAMBITTS`, incorporating direct covariate influence on reward, and evaluating performance under nonlinear data-generating mechanisms. All figures are based on 250 Monte Carlo runs per agent, with 95% confidence intervals shown.

## 6.1    Illustrative Example

We begin with a simple example where the outcome at time $t$ is given by $Y_t = \beta Z_t^{\text{optimism}} + \varepsilon_t$, with $Z_t^{\text{optimism}}$ denoting the optimism score of the generated message and $\varepsilon_t \overset{\text{IID}}{\sim} \mathcal{N}(0, \sigma)$. As discussed above, parameters $\beta$ and $\sigma$ are derived from IHS data. This setup isolates a single semantic dimension, where we expect GAMBITTS to outperform standard Thompson sampling by leveraging shared reward structure. We compare GAMBITTS algorithms to both contextual and non-contextual standard Thompson sampling agents (`StdTS:Contextual` and `StdTS`, respectively); as shown in Figure 2, `StdTS` performs better here and is used as the baseline in subsequent simulations. As expected, GAMBITTS methods achieve lower regret than standard Thompson sampling, with `poGAMBITTS` performing especially well in this setting.

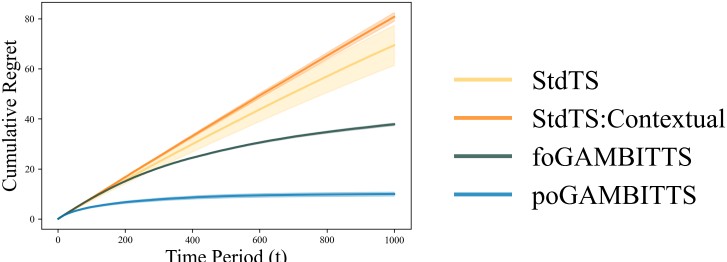

Figure 2: Cumulative Regret Under Single-Dimension Reward Model

## 6.2 Embedding Misspecification

The assumption that the agent has correctly identified how $G$ influences $Y$ (i.e., that $Z = Z^*$) is quite strong. To assess the impact of misspecifying this mechanism, we revisit the illustrative setting from Section 6.1, but evaluate performance when the agent uses a linear model with an alternative embedding. As discussed in Section 4, we expect GAMBITTS performance to degrade as the working embedding departs from the true reward-relevant dimension (optimism).

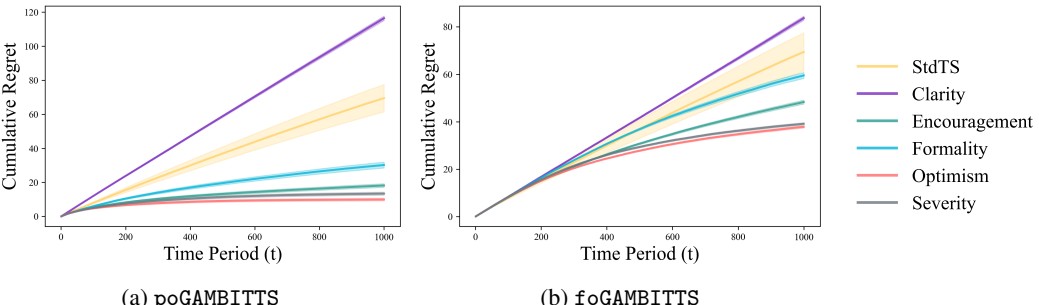

(a) `poGAMBITTS`                    (b) `foGAMBITTS`

Figure 3: Cumulative Regret Under Treatment Embedding Misspecification

Figure 3 confirms this pattern: GAMBITTS performs well when the working embedding is correlated with the true one, but deteriorates as the correlation declines (see Table G.3.4 in Appendix G). In low-correlation settings, it can even underperform standard Thompson sampling, as policy learning fails when working treatment representations are misaligned with rewards.

## 6.3 Scaling Number of Arms in a More Complex Reward Structure

We next vary the number of treatment arms to test Corollary 1, which predicts `poGAMBITTS` regret does not scale with $K$ in linear settings. Furthermore, in this section and throughout Appendix F, we turn to a more complex linear reward structure based on $Z^{\text{optimism}}$, $Z^{\text{formality}}$, and $Z^{\text{severity}}$. As we move to more complex data-generating environments, we introduce an `ens-poGAMBITTS` agent with an MLP reward model.

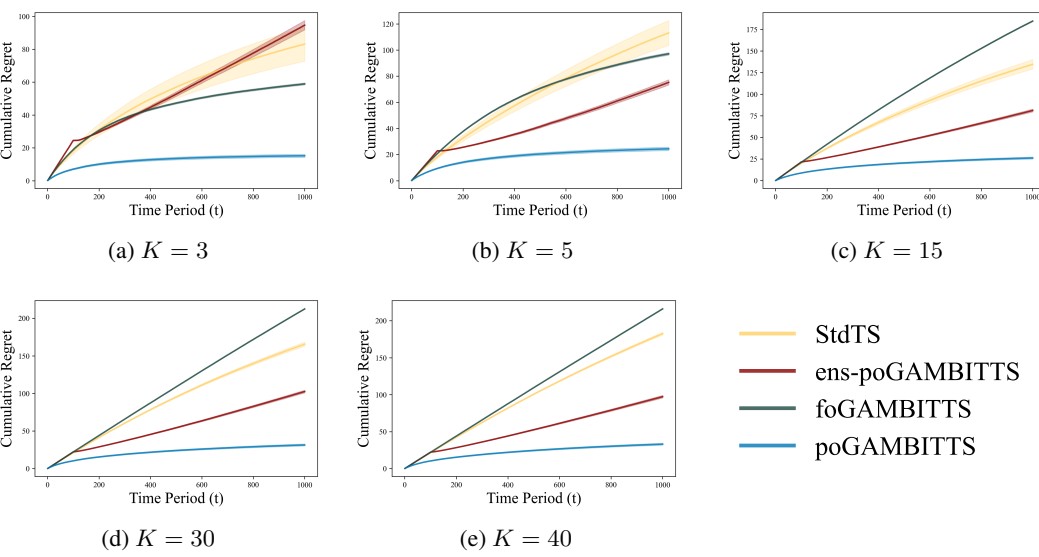

(a) $K = 3$          (b) $K = 5$          (c) $K = 15$

(d) $K = 30$         (e) $K = 40$

Figure 4: Cumulative Regret for Varying Across Sizes of Action Space

The simulations varying the size of the action set largely aligned with our expectations: the regret of the linear `poGAMBITTS` algorithm was stable across values of $K$, and the regret of `ens-poGAMBITTS` increased slightly with $K$, but not substantially. This pattern is encouraging, as it suggests that

additional intervention options can be incorporated by increasing $K$ without incurring a large regret penalty, though computational cost scales linearly with $K$. foGAMBITTS performed noticeably worse in this setting, which is expected given that it must estimate the distribution of $Z$ for each $(x, a)$ pair. A more natural comparator for foGAMBITTS is the contextual bandit, and we examine this comparison in Appendix F.

# 7 Discussion and Future Work

This paper introduces GAMBITTS for online learning in generator-mediated bandit environments, showing promise for a synergistic relationship between classical bandit methods and modern advances in generative modeling. By formalizing settings in which actions (e.g., prompts) produce stochastic treatments (e.g., generated responses), this framework opens new opportunities for personalization in domains such as marketing, mHealth, and education, where treatments cannot be easily enumerated and real-time generation can yield more deeply-tailored interventions.

This work opens up a wide range of directions for future research. One immediate area concerns the structure and specification of the working treatment embedding. As shown in Section 6, GAMBITTS can still perform well under embedding misspecification, and flexible reward models can help mitigate this issue. Still, a natural extension would be to learn the embedding online, (e.g., through online sufficient dimensionality reduction). Another direction involves moving beyond a static generator. While this manuscript assumes a fixed model, fine-tuning the generator based on observed outcomes could offer new opportunities for intervention design. Finally, this manuscript assumes that the agent always delivers the generated response to the environment. While this may be appropriate in some applications, in others it may be essential to ensure that any content sent to users is safe. In such cases, intervention designers may wish to incorporate safety checks or content filters prior to delivery. Incorporating these constraints into the generator-mediated bandit framework may require new GAMBITTS variants and accompanying theoretical analysis.

## Acknowledgments

The authors thank researchers at d3center for their comments in the early stages of this work; in particular: Daniel Almirall, Walter Dempsey, Mason Ferlic, Easton Huch, Wenchu Pan, Yao Song, and Shiyu Zhang. We also thank Jordy Berne, Liza Levina, and Abhiti Mishra for their comments on later drafts of the manuscript, as well as Elena Frank and Srijan Sen, director and principal investigator of the Intern Health Study, for their support and coordination. Lastly, Gabriel Durham and Ambuj Tewari acknowledge support from the NIH via grants 5R01DA039901 and P50DA054039, and 5R01MH131617 and 5U01MH136025, respectively.

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

# A  Further Related Work

Eldowa et al. [2024] study *bandits with mediator feedback* from an information-theoretic perspective, broadly considering settings where the effect of an action is funneled through a stochastic mediator [12].[2] Among works on bandits with mediator feedback, our approach is most closely related to those that draw connections to instrumental variables and noncompliance. Zhang et al. [2022] and Della Vecchia and Basu [2025] adopt an econometric perspective, developing OFUL-based algorithms for bandit problems with endogenous covariates. While both consider continuous treatments, they do not focus on high-dimensional treatment spaces or settings in which treatments can be generated offline without interacting with the environment [58, 9]. Moreover, these works emphasize interpretable parameter estimation, particularly in the structural model of the reward, whereas our primary focus is on predictive modeling for policy learning. This distinction is important in generative settings, where treatments (e.g., LLM-generated text) are difficult to interpret directly, and causal inference is used to support adaptive decision-making rather than uncover underlying mechanisms.

Other related efforts include Stirn and Jebara [2018], which introduces Thompson sampling in the context of noncompliance, assuming a shared, discrete action and treatment space [49]. Kveton et al. [2023] also study noncompliance and employ a Thompson sampling-based approach, aiming to identify actions with the highest compliance-weighted mean reward, rather than modeling the full stochastic treatment-generation process [29].

The generator-mediated bandit setup most closely resembles the noncompliance-inspired *instrument-armed bandit* (IAB) framework of Kallus [2018], where each arm pull represents the choice of an instrument, and the mediated bandit environment of Zou et al. [2025] [25, 59]. One key departure in our setting is the presence of a stochastic generator that produces the treatment, which leads us to focus on questions of treatment representation, generator access, and nonlinear outcome modeling, areas not emphasized in the IAB or other mediated bandit frameworks.

In contrast to Kallus' focus, our setting does not require the treatment space $\mathcal{G}$ to match the action space $\mathcal{A}$. While assuming a shared action and treatment space is natural for modeling noncompliance, where the intended and realized actions are ideally aligned, our setting explicitly separates the two. Here, the distinction is fundamental rather than incidental, reflecting the design goal of producing personalized responses rather than enforcing direct action execution.

Like Zou et al. [2025], we consider a general framework for learning in mediated bandit environments. However, motivated by the complex nature of generative outputs, we focus on flexible, potentially nonlinear models for both the mediator and the reward. In contrast, Zou et al. frame their approach more explicitly as a surrogate-reward method,[3] emphasizing linear models and aiming for robustness in cases where the action may influence the reward directly, outside the mediation pathway.[4]

The works discussed above present algorithms that can be broadly understood as instances of a general *noisy action* framework, where the environment observes a stochastic transformation of the agent's chosen action.[5] From this perspective, GAMBITTS represents a specific instantiation of a noisy-action Thompson sampler. While GAMBITTS is motivated by the GAMBIT framework, with high-dimensional and continuous treatment spaces, taking $h = I$ (as defined in Section 3) recovers algorithms suited to the econometric and psychometric mediated bandit environments discussed above. Further taking $\mathcal{G} = \mathcal{A}$ yields algorithms appropriate for noncompliance settings.

In addition to bandit methods with comparable structure, described above, another line of research has explored intersections between LLMs and bandits. Although these approaches address different statistical questions, they nonetheless relate to our setting through their shared focus on combining LLMs with bandit methods. Within this broader category, the most directly related line of work concerns using bandits for prompt selection, where prompts are treated as arms and evaluated based on task-specific performance. In contrast with our approach, much of this literature considers settings with deterministic rewards once the LLM output is observed, or with white-box access to the model

---

[2]Their analysis focuses on discrete mediators and introduces an EXP4-based algorithm.

[3]We note that GAMBITTS can also be viewed in this way, as discussed in Section 5.

[4]In our motivating application, the action affects the outcome only through the generated treatment, making full mediation a reasonable assumption.

[5]We use the term *noisy action* to emphasize that, from the environment's perspective, the observed treatment is a noisy representation of the agent's intended action.

[7, 8, 34, 46]. Another line of recent research examines LLMs themselves as the decision-making agents in multi-armed bandit [27, 17, 26, 41]. For a broader overview of work at the intersection of LLMs and bandits, see Bouneffouf and Feraud [2025] [5].

In the mHealth literature, researchers have recently begun exploring how LLMs can support intervention design, motivated by the the parallel rise of mobile interventions interventions and generative models. Haag et al. [2024] compared the quality of JITAI intervention text generated by GPT-4 to that authored by both domain experts and non-experts. Their findings showed that LLM-generated content performed favorably compared with human-authored text [16]. Additionally, James et al. [2024] conducted a randomized trial comparing LLM- and human-generated goals in a gamified mHealth application, evaluating their impact on participant engagement. The study found no significant difference in engagement between the two groups, suggesting that LLMs may offer a viable and scalable alternative for content generation in such settings [23]. However, as the goals were pre-written and not personalized in real time; the study did not evaluate the use of LLMs for on-the-fly message generation or adaptive personalization. Lastly, the IHS-COMPASS studies used LLMs to generate JITAI intervention text in recent deployments. However, as in James et al. [2024], these messages were pre-written rather than generated on-the-fly [13].

# B    Practical Variations

## B.1    Restricting Context for the Generator

As discussed in Section 3, in many practical applications, the agent may restrict which features of $X_t$ are passed to the generator, selecting a subset $X_t^G \subseteq X_t$ to include in the query. While some components of $X_t$ may be useful for outcome modeling, they may not be appropriate for inclusion in the prompt sent to the generator. For example, a variable like weight could improve reward prediction but would likely be excluded from the generator input for ethical or design reasons. The corresponding causal structure is shown in Figure 5.

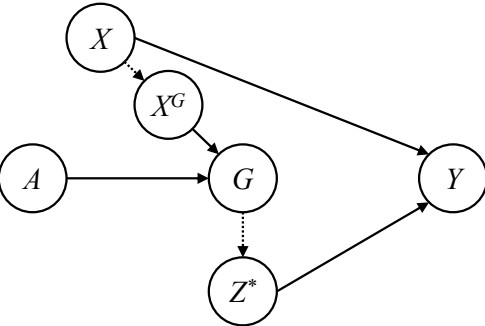

Figure 5: Generator-Mediated Bandit Causal Structure (Restricted Context in Query)

*Notes: (i) Dotted lines represent deterministic relationships.*
*(ii) Arrows from $X$ to $A$ are omitted to represent the data-generating process rather than the decision logic.*
*(iii) $A$: Action/Query; $X$: Full Context; $X^G$: Context Sent to Generator; $G$: Generated Response; $Z^*$: True Response Embedding; $Y$: Reward.*

The methods and analyses in this manuscript change only superficially when $X_t^G \neq X_t$. In this case, the treatment model introduced in Section 4, $f_1 \left( z; A_t, X_t, \theta_1 \right)$, is replaced by $f_1 \left( z; A_t, X_t^G, \theta_1 \right)$. Additionally, all algorithms that invoke $f_1$ inherit this modification.

## B.2    Optional Prompting Extension

As discussed in Section 4, mHealth interventions frequently begin with a decision of whether to deliver any message at all. GAMBITTS can be naturally extended to incorporate this structure. In this variant, a primary agent first decides whether to act (e.g., to send a message). A GAMBITTS-based agent then operates only at the time points where the primary agent chooses to act. Algorithm 3 outlines this extension.

---

**Algorithm 3** `GAMBITTS with Optional Prompting`

---

**Inputs:** Full history $\mathcal{D}_t^{(1)}$, GAMBITTS history $\mathcal{D}_t^{(2)}$, prompting policy $\mu^{(1)}$, GAMBITTS policy $\mu^{(2)}$, context $x_t$.

1: **Sample** $R_t \sim \mu^{(1)}\left(\mathcal{D}_t^{(1)}, x_t\right)$ according to Agent 1.
2: **If** $R_t = 1$ then
3:     Select query $A_t \sim \mu^{(2)}\left(\mathcal{D}_t^{(2)}, x_t\right)$ according to Agent 2.
4:     Update $\mathcal{D}_{t+1}^{(2)} \leftarrow \mathcal{D}_t^{(2)} \cup \{(x_t, A_t, Z_t, Y_t)\}$.
5:     Update policy $\mu^{(2)}$.
6: **Update** $\mathcal{D}_{t+1}^{(1)} \leftarrow \mathcal{D}_t^{(1)} \cup \{(x_t, R_t, Y_t)\}$.
7: **Update** policy $\mu^{(1)}$.

---

## C    Embedding Misspecification

As discussed in Section 4, GAMBITTS performance hinges on the construction of working embedding $Z = h(G)$, with the agent modeling $Y \mid Z, X$ in place of $Y \mid G, X$. Section 6.2 supports this notion, showing GAMBITTS is robust to mild misspecification but suffers under severe misspecification. With this phenomenon in mind, this appendix discusses two approaches aimed at improving robustness to embedding misspecification.

One option for promoting robustness is to construct a hedged reward model. In particular, given any initial embedding construct $Z$ and model $m_2$, an analyst can consider the modified form:

$$\tilde{m}_2(A_t, Z_t, X_t; \beta^\cdot, \theta_2) = \sum_{a \in \mathcal{A}} \beta^a \mathbb{1}_{A_t = a} + m_2(Z_t, X_t; \theta_2).$$

When the goal is online learning of the optimal action $A$, this structure offers a hedge against misspecification of $Z$ and $m_2$. In particular, if $m_2$ fails to capture signal, its estimated contribution to $\tilde{m}_2$ will shrink towards zero. In that case, the $\tilde{m}_2$ model asymptotically behaves like a standard Thompson sampling agent that learns a reward distribution over actions directly. Conversely, if $Z$ and $m_2$ are well-specified, then the action $A_t$ provides no additional information beyond what is already captured through $m_2(Z_t, X_t; \theta_2)$. In this case, the estimated $\beta^\cdot$ terms will shrink to zero and $\tilde{m}_2$ will converge to $m_2$. However, the agent must first learn that the embedding carries signal, potentially delaying efficient generalization early in learning.

Figure 6 revisits the analysis presented in Figure 3 of Section 6.2, now including hedged models. All hedged `poGAMBITTS` agents perform similarly, as do all hedged `foGAMBITTS` agents. While every hedged GAMBITTS variant outperforms standard Thompson sampling in this setting, the well-specified models no longer exhibit the performance advantage they had in the original (unhedged) analysis. This supports the hypothesis that hedging reduces downside risk but limits gains under well-specified embeddings.

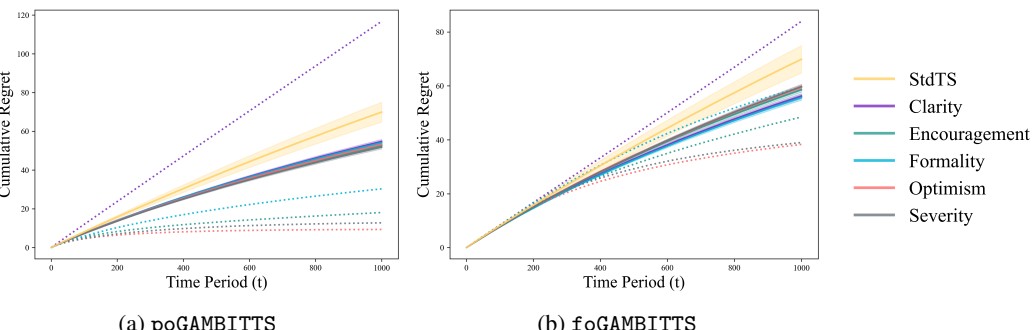

        (a) `poGAMBITTS`                    (b) `foGAMBITTS`

Figure 6: Cumulative Regret Under Embedding Misspecification - Original vs Hedged Approaches
*Note: Dotted lines represent cumulative regret under original (unhedged) approaches.*

A second option is to maintain a set of candidate reward models, each potentially using different treatment embeddings. Periodically, the agent can evaluate these models based on predictive performance and switch to the one that best predicts observed outcomes. This allows the algorithm to recover from poor initial choices of $Z$ (and $m_2$ more broadly) without requiring strong prior knowledge about which embedding will perform best.

Improving robustness to embedding misspecification also points to several directions for future work. For example, extending the candidate model approach by developing a corralling bandit, where a master algorithm selects among reward models, could allow the agent to adaptively shift across models and tailor the choice of embedding to the individual. Similarly, as discussed in Section 7, learning individualized treatment embeddings through methods like online sufficient dimensionality reduction would mitigate misspecification risk.

# D   Ensemble-Based Algorithms

As in Lu and Van Roy [2017], given a second-stage reward model parametrized by $\theta_2$, ensemble-based GAMBITTS approaches require initially drawing $M_{ens}$ sets of parameters $\tilde{\theta}_{2,1}^1, \ldots, \tilde{\theta}_{2,1}^{M_{ens}}$ from a pre-determined initialization distribution. Algorithms 4 and 5 then demonstrate how to select action $a_t$ for $t = 1, \ldots, T$. As in Algorithms 1 and 2, an analyst may approximate the integrals below via Monte Carlo.

Steps 5 and 4 in Algorithms 4 and 5 (respectively) require updating model parameters as described in Lu and Van Roy [2017]. These updates involve perturbations of observed reward to promote exploration [36].

---
**Algorithm 4** `Ensemble-Based Fully Online GAMBITTS (ens-foGAMBITTS)`
---

**Inputs**: Data $\mathcal{D}_t$, prior $\pi_1$, models $f_1$, $m_2$, current context $x_t$, reward model parameters $\left\{\tilde{\theta}_{2,t}^m\right\}_{m=1}^{M_{ens}}$.

1: **Sample** $\theta_{1,t} \sim P_1\left(\theta_1 \mid \mathcal{D}_t\right)$
2: **Sample** $j \sim \mathcal{U}\left(\{1, \ldots, M_{ens}\}\right)$.
3: **For** each $a \in \mathcal{A}$, calculate

$$\mathbb{E}_{\theta_{1,t}, \tilde{\theta}_{2,t}^j}\left[Y \mid a, x_t\right] = \int_{\mathcal{Z}} m_2\left(z, x_t; \tilde{\theta}_{2,t}^j\right) f_1(z; a, x_t, \theta_{1,t}) dz.$$

4: **Set** $A_t = \arg\max\limits_{a \in \mathcal{A}} \mathbb{E}_{\theta_{1,t}, \tilde{\theta}_{2,t}^j}\left[Y \mid a, x_t\right]$.
5: **Given** $(z_t, x_t, y_t)$, update $\tilde{\theta}_{2,t+1}^1, \ldots, \tilde{\theta}_{2,t+1}^{M_{ens}}$.

---

---
**Algorithm 5** `Ensemble-Based Partially Online GAMBITTS (ens-poGAMBITTS)`
---

**Inputs**: Data $\mathcal{D}_t$, model $m_2$, current context $x_t$, pretrained model $f_1^{off}$, reward model parameters $\left\{\tilde{\theta}_{2,t}^m\right\}_{m=1}^{M_{ens}}$.

1: **Sample** $j \sim \mathcal{U}\left(\{1, \ldots, M_{ens}\}\right)$.
2: **For** each $a \in \mathcal{A}$, calculate

$$\mathbb{E}_{\tilde{\theta}_{2,t}^j}\left[Y \mid a, x_t\right] = \int_{\mathcal{Z}} m_2\left(z, x_t; \tilde{\theta}_{2,t}^j\right) f_1^{off}(z; a, x_t) dz.$$

3: **Set** $A_t := \arg\max\limits_{a \in \mathcal{A}} \mathbb{E}_{\tilde{\theta}_{2,t}^j}\left[Y \mid a, x_t\right]$.
4: **Given** $(z_t, x_t, y_t)$, update $\tilde{\theta}_{2,t+1}^1, \ldots, \tilde{\theta}_{2,t+1}^{M_{ens}}$.

---

# E    Further Theoretical Discussion

In this appendix, we provide proofs for the results discussed in Section 5, along with additional theoretical developments related to Thompson sampling in the GAMBITTS framework. The appendix is organized as follows:

- Appendix E.1: Proof of the regret upper bound presented in Section 5 for standard Thompson sampling in a GAMBIT environment.

- Appendix E.2: Proofs of the regret upper bounds presented in Section 5 for `poGAMBITTS`-based algorithms, as well as a bound on the number of samples needed for the offline (empirical) treatment distribution to approximate the true distribution.

- Appendix E.3: Proof of the regret upper bound presented in Section 5 for `foGAMBITTS`.

- Appendix E.4: Statement and proof of a regret lower bound for `foGAMBITTS`.

## E.1    Upper Regret Bound for Standard Thompson Sampling

In this section, we present a Bayesian regret bound of a standard Thompson sampling algorithm that models the problem as a conventional multi-armed contextual bandit over $\mathcal{X} \times \mathcal{A}$, ignoring the generator-mediation structure. Our analysis adapts the proof from Chapter 36.1 of Lattimore and Szepesvári [2020] [31].

**Claim 1.** *The Bayesian regret of the standard Thompson sampling agent that models the problem as a multi-armed contextual bandit over $\mathcal{X} \times \mathcal{A}$ is bounded by*

$$BR_T^{standard} \leq \mathcal{O}((\sigma_1 + \sigma_2)\sqrt{CKT \log T} + BCK)$$

*where $B$ is a uniform bound on the mean reward such that $|m_2(z, x; \theta)| \leq B$.*

*Proof.* Recall that the reward received when the agent selects action $a \in \mathcal{A}$ under context $x \in \mathcal{X}$ is given by $m_2(Z, x; \theta_2) + \eta$, where $\eta$ is $\sigma_2$-subgaussian and $Z \in \mathbb{R}^d$ is drawn from the distribution $\xi_{x,a}^{\theta_1}$.[6] As discussed in Section 5, we assume that $m_2(Z, x; \theta_2)$ is $\sigma_1$-subgaussian. It follows that the overall reward for a given context-action pair $(x, a)$ is $(\sigma_1 + \sigma_2)$-subgaussian.

We use $m(x, a)$ to denote the mean reward for the context-action pair $(x, a)$; i.e.,

$$m(x, a) := \int_{\mathcal{Z}} m_2(z, x; \theta_2) f_1(z; x, a, \theta_1) dz.$$

Since we assume that the mean reward is bounded by $|m_2(z, x; \theta_2)| \leq B$, it follows that $|m(x, a)| \leq B$.

Now, we adapt the Bayesian regret analysis for a standard Thompson sampling algorithm for multi-armed bandit settings to the *contextual* multi-armed bandit setting where the context is assumed to be drawn i.i.d. every round.

Well, for each $a \in \mathcal{A}$, $x \in \mathcal{X}$ and $t = 1, \ldots, T$, define

$$U_t(x, a) := \text{clip}_{[-B, B]}\left(\widehat{m}_{x,a}(t-1) + (\sigma_1 + \sigma_2)\sqrt{\frac{2\log(1/\delta)}{1 \vee N_{x,a}(t-1)}}\right),$$

where $N_{x,a}(t-1) = \sum_{s=1}^{t-1} \mathbb{1}_{\{X_t = x, A_t = a\}}$ is the count of the pair $(x, a)$ up to time $t - 1$ and $\widehat{m}_{x,a}(t-1)$ denotes the empirical estimate of the reward of action $a$ under context $x$ based on observations up to time $t - 1$. If $N_{x,a}(t-1) = 0$, we set $\widehat{m}_{x,a}(t-1) = 0$.

Let $E$ be the event that, for all $t = 1, \ldots, T$ and $(x, a) \in \mathcal{X} \times \mathcal{A}$, the empirical estimate satisfies

$$|\widehat{m}_{t-1}(x, a) - m(x, a)| \leq (\sigma_1 + \sigma_2)\sqrt{\frac{2\log(1/\delta)}{1 \vee N_{x,a}(t-1)}}.$$

---

[6]Recall $\xi_{x,a}^{\theta_1}$ has density $f_1(z; x, a, \theta_1)$.

Under this setup, the standard Hoeffding concentration bound gives $\mathbb{P}\left[E^c\right] \leq 2CKT\delta$ where $C = |\mathcal{X}|$ and $K = |\mathcal{A}|$.

Let $\mathcal{F}_t = \sigma(X_1, A_1, Y_1, \ldots, X_t, A_t, Y_t, X_{t+1})$ be the $\sigma$-algebra generated by the interaction sequence by the end of time $t$ (and the following context at time $t + 1$). Note that $U_t(x, a)$ is $\mathcal{F}_{t-1}$-measurable. Using the fact that, under Thompson sampling, the conditional distributions of $A_t^* = \arg\max_a m(X_t, a)$ and $A_t$ given $\mathcal{F}_{t-1}$ are identical, we obtain

$$\mathrm{BR}_T = \mathbb{E}\left[\sum_{t=1}^{T}(m(X_t, A_t^*) - m(X_t, A_t))\right]$$

$$= \mathbb{E}\left[\sum_{t=1}^{T}\mathbb{E}\left[m(X_t, A_t^*) - m(X_t, A_t) \mid \mathcal{F}_{t-1}\right]\right]$$

$$= \mathbb{E}\left[\sum_{t=1}^{T}(m(X_t, A_t^*) - U_t(X_t, A_t^*)) + \sum_{t=1}^{T}(U_t(X_t, A_t) - m(X_t, A_t))\right].$$

On the event $E^c$, the terms inside the expectation are bounded by $2BT$. On the event $E$, the first sum is negative and the second sum is bounded by

$$\mathbb{1}_{\{E\}}\sum_{t=1}^{T}(U_t(X_t, A_t) - m(X_t, A_t))$$

$$= \mathbb{1}_{\{E\}}\sum_{t=1}^{T}\sum_{x \in \mathcal{X}}\sum_{a \in \mathcal{A}}\mathbb{1}_{\{X_t=x, A_t=a\}}(U_t(x, a) - m(x, a))$$

$$\leq (\sigma_1 + \sigma_2)\sum_{x \in \mathcal{X}}\sum_{a \in \mathcal{A}}\sum_{t=1}^{T}\mathbb{1}_{\{X_t=x, A_t=a\}}\sqrt{\frac{8\log(1/\delta)}{1 \vee N_{x,a}(t-1)}}$$

$$\leq (\sigma_1 + \sigma_2)\sum_{x \in \mathcal{X}}\sum_{a \in \mathcal{A}}\int_0^{N_{x,a}(T)}\sqrt{\frac{8\log(1/\delta)}{s}}ds$$

$$= (\sigma_1 + \sigma_2)\sum_{x \in \mathcal{X}}\sum_{a \in \mathcal{A}}\sqrt{32N_{x,a}(T)\log(1/\delta)} \leq (\sigma_1 + \sigma_2)\sqrt{32CKT\log(1/\delta)}.$$

The result follows from choosing $\delta = \frac{1}{T^2}$ and applying the bound $\mathbb{P}\left[E^c\right] \leq 2CKT\delta$. $\qquad\square$

### E.2 Upper Regret Bounds for `poGAMBITTS`

The contextual bandit environment of interest with the two-stage reward generating process is fully specified by

$$\mathcal{E}(\theta_1, \theta_2) = \left(P_{\mathcal{X}}, \{\xi_{x,a}^{\theta_1}\}_{x \in \mathcal{X}, a \in \mathcal{A}}, \{m_2(z, x; \theta_2)\}_{x \in \mathcal{X}, a \in \mathcal{A}}, P_\eta\right),$$

where $P_{\mathcal{X}}$ is the context distribution, $\xi_{x,a}^{\theta_1}$ is the distribution of the representation $Z \in \mathbb{R}^d$ given the context-action pair $(x, a)$, $\theta_2$ is the parameter that governs the mean reward function $m_2(z, x; \theta_2)$, and $P_\eta$ is the $\sigma_2$-subgaussian noise distribution.

In the analysis of `poGAMBITTS`, we consider the Bayesian regret with $\theta_1$ fixed to $\theta_1^*$ and $\theta_2$ is integrated over the distribution $\pi_2$. We denote the environment with $\theta_1$ fixed to the true parameter $\theta_1^*$ by

$$\mathcal{E}(\theta_2) = \mathcal{E}(\theta_1^*, \theta_2) = \left(P_{\mathcal{X}}, \{\xi_{x,a}^{\theta_1^*}\}_{x \in \mathcal{X}, a \in \mathcal{A}}, \{m_2(z, x; \theta_2)\}_{x \in \mathcal{X}, a \in \mathcal{A}}, P_\eta\right).$$

Theorem 1 assumes access to an Alg with the Bayesian regret bound of $\mathrm{BR}_T^{Alg}$ under the approximate environment

$$\widetilde{\mathcal{E}}(\theta_2) = \left(P_{\mathcal{X}}, \{\xi_{x,a}^{off}\}_{x \in \mathcal{X}, a \in \mathcal{A}}, \{m_2(z, x; \theta_2)\}_{x \in \mathcal{X}, a \in \mathcal{A}}, P_\eta\right),$$

where $\xi_{x,a}^{\theta_1^*}$ in the definition of $\mathcal{E}(\theta_2)$ is replaced with its offline empirical estimate $\xi_{x,a}^{off}$ (with density $f_1^{off}(\cdot|x, a)$), and the same mean reward function $m_2(z, x; \theta_2)$ parameterized by $\theta_2$ is used. To be

more precise, the Bayesian regret of the algorithm Alg under the approximate environment $\widetilde{\mathcal{E}}(\theta_2)$ is defined as

$$\mathrm{BR}_T^{Alg} = \mathbb{E}_{\theta_2 \sim \pi_2}\left[\mathbb{E}_{\widetilde{\mathcal{E}}(\theta_2)}\left[\sum_{t=1}^{T}\left(\mathbb{E}_{\xi_{X_t,\widetilde{A}_t^*}^{off}}\left[m_2(Z,X_t) \mid X_t\right] - \mathbb{E}_{\xi_{X_t,A_t}^{off}}\left[m_2(Z,X_t) \mid X_t\right]\right)\right]\right],$$

where $\widetilde{A}_t^* = \arg\max_a \mathbb{E}_{\xi_{X_t,a}^{off}}\left[m_2(Z,X_t) \mid X_t, \theta_2\right]$; i.e., the optimal action in context $X_t$ under the environment $\widetilde{\mathcal{E}}(\theta_2)$. Theorem 1 shows that running the algorithm Alg under the true environment $\mathcal{E}(\theta_2)$ achieves a similar regret bound as long as the approximation of $\mathcal{E}(\theta_2)$ by $\widetilde{\mathcal{E}}(\theta_2)$ is sufficiently close.

**Theorem 1** (Restated). *Let $f_1^{off}$ be an empirical model for the treatment distribution satisfying (2). Let* Alg *be a contextual bandit algorithm that selects actions based on the estimated reward function $\widehat{m}_2$. Then, running* Alg *in the partially online stochastic treatment setting gives $T$-step Bayesian regret $BR_T^{Alg}$, with*

$$BR_T^{Alg} \leq \mathcal{O}\left(\widehat{BR}_T^{Alg} + T\sqrt{\varepsilon T}\right),$$

*where $\widehat{BR}_T^{Alg}$ is the $T$-step Bayesian regret of* Alg *with respect to $\widehat{m}_2$.*

We provide a proof of Theorem 1 below, which relies on Pinsker's inequality.

**Lemma 1** (Pinsker's inequality). *For any pair of measures $P$ and $Q$ on the same probability space $(\Omega, \mathcal{F})$, we have*

$$\delta(P,Q) = \sup_{A \in \mathcal{F}} P(A) - Q(A) \leq \sqrt{\frac{1}{2}D_{KL}\left(P\|Q\right)},$$

*where $D_{KL}$ denotes KL divergence in nats.*

We now turn to the proof of Theorem 1.

*Proof of Theorem 1.* Let $P_{x,a}^{\theta_2}$ be the distribution of the reward given the context-action pair $(x,a)$ under the environment $\mathcal{E}(\theta_2)$; i.e., the distribution of the random variable $m_2(Z,x;\theta_2) + \eta$ where $Z \sim \xi_{x,a}^{\theta_1^*}$ and $\eta \sim P_\eta$. Similarly, let $\widetilde{P}_{x,a}^{\theta_2}$ be the distribution of the reward given the context-action pair $(x,a)$ under the environment $\widehat{\mathcal{E}}(\theta_2)$; i.e., the distribution of the random variable $m_2(\widetilde{Z},x;\theta_2) + \eta$ where $\widetilde{Z} \sim \xi_{X_t,a}^{off}$ and $\eta \sim P_\eta$.

Recalling the assumption that $f_1^{off}$ satisfies (2) (i.e., $KL\left(f_1^{off}(\cdot;x,a)\|f_1(\cdot;x,a,\theta_1)\right) \leq \varepsilon$), we observe

$$\begin{aligned} KL\left(P_{x,a}^{\theta_2}\|\widetilde{P}_{x,a}^{\theta_2}\right) &= KL\left(\mathcal{L}(g(Z,\eta))\|\mathcal{L}(g(\widetilde{Z},\eta))\right) \\ &\leq KL\left(\mathcal{L}(Z,\eta)\|\mathcal{L}(\widetilde{Z},\eta)\right) \\ &= KL\left(\mathcal{L}(Z)\|\mathcal{L}(\widetilde{Z})\right) \\ &\leq \varepsilon, \end{aligned}$$

where $g(z,\eta) = m_2(z,x;\theta_2) + \eta$ and $\mathcal{L}(W)$ denotes the law/distribution of random variable $W$. The first inequality is due to the data processing inequality.

The interaction between the algorithm Alg and the environment $\mathcal{E}(\theta_2)$ determines the distribution of the sequence $(X_1, A_1, Z_1, Y_1, \ldots, X_T, A_T, Z_T, Y_T)$. With a slight abuse of notation, let $P(\theta_2)$ be the probability measure on the sequence generated by Alg under the environment $\mathcal{E}(\theta_2)$ and let $\widetilde{P}(\theta_2)$ be the probability measure on the sequence under the environment $\widetilde{\mathcal{E}}(\theta_2)$. Then, standard KL divergence decomposition results[7] for bandit environments gives

$$KL\left(P(\theta_2)\|\widetilde{P}(\theta_2)\right) = \sum_{x \in \mathcal{X}}\sum_{a \in \mathcal{A}} N_{x,a}(T)KL\left(P_{x,a}^{\theta_2}\|\widetilde{P}_{x,a}^{\theta_2}\right) \leq \varepsilon T,$$

---

[7]E.g. Lemma 15.1 in Lattimore and Szepesvári [2020] [31].

where $N_{x,a}(T)$ is the count of the pair $(x,a)$ up to time $T$, as in Appendix E.1. Hence, the Bayesian regret of the algorithm Alg under the environment $\mathcal{E}(\theta_2)$ averaged over the prior distribution on $\theta_2$ is[8]

$$
\begin{aligned}
\mathrm{BR}_T &= \mathbb{E}_{\theta_2}\left[\mathbb{E}_{P(\theta_2)}\left[\sum_{t=1}^{T}\left(\mathbb{E}_{\xi^{\theta_1^*}_{X_t,A_t^*}}[m_2(Z,X_t;\theta_2)] - \mathbb{E}_{\xi^{\theta_1^*}_{X_t,A_t}}[m_2(Z,X_t;\theta_2)]\right)\right]\right] \\
&\leq \mathbb{E}_{\theta_2}\left[\mathbb{E}_{\widetilde{P}(\theta_2)}\left[\sum_{t=1}^{T}\left(\mathbb{E}_{\xi^{\theta_1^*}_{X_t,A_t^*}}[m_2(Z,X_t;\theta_2)] - \mathbb{E}_{\xi^{\theta_1^*}_{X_t,A_t}}[m_2(Z,X_t;\theta_2)]\right)\right.\right. \\
&\qquad\qquad \left.\left. + 2TB\cdot\delta\left(P(\theta_2),\widetilde{P}(\theta_2)\right)\right]\right] \\
&\leq \mathbb{E}_{\theta_2}\left[\mathbb{E}_{\widetilde{P}(\theta_2)}\left[\sum_{t=1}^{T}\left(\mathbb{E}_{\xi^{off}_{X_t,A_t^*}}[m_2(Z,X_t;\theta_2)] - \mathbb{E}_{\xi^{off}_{X_t,A_t}}[m_2(Z,X_t;\theta_2)]\right)\right.\right. \\
&\qquad\qquad \left.\left. + 2TB\cdot\delta\left(\xi^{\theta_1^*}_{X_t,A_t^*},\xi^{off}_{X_t,A_t^*}\right) + 2TB\cdot\delta\left(P(\theta_2),\widetilde{P}(\theta_2)\right)\right]\right] \\
&\leq \mathbb{E}_{\theta_2}\left[\mathbb{E}_{\widetilde{P}(\theta_2)}\left[\sum_{t=1}^{T}\left(\mathbb{E}_{\xi^{off}_{X_t,A_t^*}}[m_2(Z,X_t;\theta_2)] - \mathbb{E}_{\xi^{off}_{X_t,A_t}}[m_2(Z,X_t;\theta_2)]\right)\right.\right. \\
&\qquad\qquad \left.\left. + 2TB\sqrt{\frac{1}{2}KL\left(\xi^{\theta_1^*}_{X_t,A_t^*}\|\xi^{off}_{X_t,A_t^*}\right)} + 2TB\sqrt{\frac{1}{2}KL\left(P(\theta_2)\|\widetilde{P}(\theta_2)\right)}\right]\right] \\
&\leq \mathrm{BR}_T^{Alg} + TB\sqrt{2(T+1)\varepsilon}.
\end{aligned}
$$

Here, we use the notations $\delta(X,Y)$ for the total variation between $X$ and $Y$, and $\widetilde{A}_t^* = \arg\max_a \mathbb{E}_{\xi^{off}_{X_t,a}}[m_2(Z,X_t;\theta_2)\mid X_t,\theta_2]$, which is the optimal action in context $X_t$ under the environment $\widetilde{\mathcal{E}}(\theta_2)$. The first inequality follows from the fact that the absolute value of the expectant is bounded by $2TB$. The third inequality follows from Pinsker's inequality (Lemma 1). The final inequality follows from the assumption that the algorithm Alg achieves Bayesian regret $\mathrm{BR}_T^{Alg}$ under the environment $\widetilde{\mathcal{E}}(\theta_2)$, averaged over $\theta_2 \sim \pi_2$. The result follows.

$\square$

### E.2.1 Linear Reward Function

Now, we use the reduction result in Theorem 1 to obtain a Bayesian regret bound for poGAMBITTS under the environment $\mathcal{E}(\theta_2)$, where the mean reward function $m_2(z,x;\theta_2)$ is linear in $z$. To apply the reduction, we first establish a Bayesian regret bound for the Thompson sampling algorithm under the environment $\widetilde{\mathcal{E}}(\theta_2)$, assuming knowledge of $f_1^{off}$.

This analysis relies on two standard results from the theory of linear contextual bandits. The first is the concentration bound for the regression-based estimate of the linear parameter:

**Lemma 2** (Theorem 20.5 in [31])**.** *With probability at least $1-\delta$, we have, for all $t = 1,\ldots,T$, that*

$$\|\theta - \widehat{\theta}_t\|_{V_t} \leq \beta,$$

*where $\beta := 1 + \sigma_2\sqrt{2\log(1/\delta) + d\log(1+T/d)}$.*

The second result is a technical lemma commonly used to bound the cumulative sum of upper confidence bounds in the analysis of optimism-based linear contextual bandit algorithms:

**Lemma 3** (Elliptical potential lemma)**.** *Let $V_0 \in \mathbb{R}^{d\times d}$ be positive definite and $a_1,\ldots,a_n \in \mathbb{R}^d$ be a sequence of vectors with $\|a_t\|_2 \leq L$ for all $t = 1,\ldots,n$. Furthermore, let $V_t = V_0 + \sum_{s=1}^{t} a_s a_s^\top$. Then,*

$$\sum_{t=1}^{n}\left(1\wedge\|a_t\|^2_{V_{t-1}^{-1}}\right) \leq 2\log\left(\frac{\det V_n}{\det V_0}\right) \leq 2d\log\left(\frac{Tr(V_0)+nL^2}{d\det^{1/d}(V_0)}\right).$$

---

[8]Here, we use $\mathbb{E}_{\theta_2}[W]$ to denote $\mathbb{E}_{\theta_2\sim\pi_2}[W]$.

Equipped with the two lemmas above, we now derive a Bayesian regret bound for Thompson sampling in the approximate environment $\widetilde{\mathcal{E}}(\theta_2)$.

**Lemma 4.** *Consider an environment $\widetilde{\mathcal{E}}(\theta_2)$ where the mean reward function is $m_2(z, x; \theta_2) = \langle z, \theta_2 \rangle$. The Thompson sampling algorithm run under this environment with a full knowledge of $f_1^{off}$, and prior $\pi_2$ on $\theta_2$, achieves a Bayesian regret bound of*

$$BR_T \leq \mathcal{O}\left(\sigma_2 d\sqrt{T \log T}\right),$$

*Proof.* We denote by $\psi_{x,a} = \mathbb{E}_{\xi_{x,a}^{off}}[Z]$. The Bayesian regret can be decomposed as:

$$\mathrm{BR}_T = \mathbb{E}\left[\sum_{t=1}^{T}\left(\mathbb{E}_{\xi_{X_t,A_t^*}^{off}}[m_2(Z, X_t; \theta_2)] - \mathbb{E}_{\xi_{X_t,A_t}^{off}}[m_2(Z, X_t; \theta_2)]\right)\right]$$

$$= \mathbb{E}\left[\sum_{t=1}^{T}\langle \psi_{X_t,A_t^*} - \psi_{X_t,A_t}, \theta_2\rangle\right]$$

$$= \mathbb{E}\left[\sum_{t=1}^{T}\langle Z_t^* - Z_t, \theta_2\rangle\right] + \mathbb{E}\left[\sum_{t=1}^{T}\langle \psi_{X_t,A_t^*} - Z_t^*, \theta_2\rangle\right] + \mathbb{E}\left[\sum_{t=1}^{T}\langle Z_t - \psi_{X_t,A_t}, \theta_2\rangle\right],$$

where $A_t^* = \arg\max_a \langle \psi_{X_t,a}, \theta_2 \rangle$ is the best action in context $X_t$ and $Z_t^*$ is a random variable sampled from $\xi_{X_t,A_t^*}^{off}$. The second term above can be bounded by

$$\sum_{t=1}^{T}\langle \psi_{X_t,A_t^*} - Z_t^*, \theta_2\rangle \leq \mathcal{O}\left(B\sqrt{T \log\left(\frac{1}{\delta}\right)}\right),$$

which holds with probability at least $1 - \delta$ by the Azuma-Hoeffding inequality, since the sequence $\{\langle \psi_{X_t,A_t^*} - Z_t^*, \theta_2\rangle\}_{t=1}^{T}$ forms a martingale difference sequence adapted to the filtration $\mathcal{F}_t = \sigma(\theta_2, X_1, A_1, Z_1, Y_1, \ldots, X_t, A_t, Z_t, Y_t, X_{t+1})$. Choosing $\delta = \frac{1}{T}$, we can bound $\mathbb{E}\left[\sum_{t=1}^{T}\langle \psi_{X_t,A_t^*} - Z_t^*, \theta_2\rangle\right] \leq \mathcal{O}\left(B\sqrt{T \log T}\right)$. The third term can be bounded similarly. To bound the first term, we first define an upper confidence bound function $U_t : \mathcal{Z} \to \mathbb{R}$ as

$$U_t(z) = \langle z, \widehat{\theta}_{t-1}\rangle + \beta\|z\|_{V_{t-1}^{-1}},$$

where $V_t = I + \sum_{s=1}^{t} Z_s Z_s^\top$ and $\widehat{\theta}_t = V_t^{-1}\sum_{s=1}^{t} Y_s Z_s$. By the concentration inequality provided in Lemma 2, we are guaranteed that, for all $t = 1, \ldots, T$, $\|\theta_2 - \widehat{\theta}_t\|_{V_t} \leq \beta$ with probability at least $1 - \frac{1}{T}$, where we use $\beta$ defined in the lemma with $\delta = \frac{1}{T}$. Let $E_t$ be the event that $\|\theta_2 - \hat{\theta}_{t-1}\|_{V_{t-1}} \leq \beta$ and $E = \cap_{t=1}^{T} E_t$. Then,

$$\mathbb{E}\left[\sum_{t=1}^{T}\langle Z_t^* - Z_t, \theta_2\rangle\right] = \mathbb{E}\left[\mathbb{1}_{\{E^c\}}\sum_{t=1}^{T}\langle Z_t^* - Z_t, \theta_2\rangle\right] + \mathbb{E}\left[\mathbb{1}_{\{E\}}\sum_{t=1}^{T}\langle Z_t^* - Z_t, \theta_2\rangle\right]$$

$$\leq 2 + \mathbb{E}\left[\sum_{t=1}^{T}\mathbb{1}_{\{E_t\}}\langle Z_t^* - Z_t, \theta_2\rangle\right].$$

By the Thompson sampling algorithm, $\mathbb{P}[A_t^* = \cdot \mid \mathcal{F}_{t-1}] = \mathbb{P}[A_t = \cdot \mid \mathcal{F}_{t-1}]$, implying $\mathbb{P}[Z_t^* = \cdot \mid \mathcal{F}_{t-1}] = \mathbb{P}[Z_t = \cdot \mid \mathcal{F}_{t-1}]$. With this observation, the rest of the proof follows the standard Bayesian regret analysis of linear contextual bandits (e.g., Chapter 36.3 in Lattimore and Szepesvári [2020]) [31].

Since $U_t(z)$ for any fixed $z \in \mathcal{Z}$ is $\mathcal{F}_{t-1}$-measurable, $\mathbb{E}_{t-1}[U_t(Z_t^*)] = \mathbb{E}_{t-1}[U_t(Z_t)]$.[9] Therefore, it follows that the second term in the display above is bounded by

$$
\begin{aligned}
\mathbb{E}_{t-1}\left[\mathbb{1}_{\{E_t\}}\langle Z_t^* - Z_t, \theta_2 \rangle\right] &= \mathbb{1}_{\{E_t\}}\mathbb{E}_{t-1}\left[\langle Z_t^*, \theta_2 \rangle - U_t(Z_t^*) + U_t(Z_t) - \langle Z_t, \theta_2 \rangle\right] \\
&\leq \mathbb{1}_{\{E_t\}}\mathbb{E}_{t-1}\left[U_t(Z_t) - \langle Z_t, \theta_2 \rangle\right] \\
&\leq \mathbb{1}_{\{E_t\}}\langle Z_t, \widehat{\theta}_{t-1} - \theta_2 \rangle + \beta\|Z_t\|_{V_{t-1}^{-1}} \\
&\leq 2\beta\|Z_t\|_{V_{t-1}^{-1}}.
\end{aligned}
$$

Combining the above with $\mathbb{1}_{\{E_t\}}\langle Z_t^* - Z_t, \theta_2 \rangle \leq 2$ produces

$$
\begin{aligned}
\mathbb{E}\left[\sum_{t=1}^{T}\mathbb{1}_{\{E_t\}}\langle Z_t^* - Z_t, \theta_2 \rangle\right] &\leq 2\beta\sum_{t=1}^{T}\left(1 \wedge \|Z_t\|_{V_{t-1}^{-1}}\right) \\
&\leq 2\beta\sqrt{T\mathbb{E}\left[\sum_{t=1}^{T}(1 \wedge \|Z_t\|_{V_{t-1}^{-1}}^2)\right]} \\
&\leq 2\beta\sqrt{2dT\log(1 + T/d)}
\end{aligned}
$$

where the second inequality follows from Cauchy-Schwartz and the last inequality from the elliptical potential lemma (Lemma 3). Combining the bounds yields

$$
\mathrm{BR}_T \leq \mathcal{O}\left(\sigma_2 d\sqrt{T\log T}\right),
$$

as required. $\qquad\square$

Corollary 1 in the main body provides a Bayesian regret bound for `poGAMBITTS` in the linear reward setting. The corollary follows directly by viewing the algorithm as an instance of Thompson sampling for a linear contextual bandit, and applying the reduction in Theorem 1 along with the regret bound in Lemma 4.

### E.2.2  General Reward Function

In this section, we analyze the Bayesian regret of `poGAMBITTS` for general reward functions. We consider function classes $\mathcal{F}$[10] that contains the mean reward function $m_2(z, x; \theta_2)$, and derive a regret bound that holds for any such class $\mathcal{F}$. Our bound is expressed in terms of the eluder dimension of $\mathcal{F}$, a complexity measure introduced by Russo and Van Roy [2013] [44]. We begin by recalling the definition of the eluder dimension.

**Definition 1** ($\epsilon$-dependence). *A pair $(z, x) \in \mathcal{Z} \times \mathcal{X}$ is $\epsilon$-dependent on pairs $\{(z_1, x_1), \ldots, (z_n, x_n)\} \subseteq \mathcal{Z} \times \mathcal{X}$ with respect to $\mathcal{F}$ if any pair of functions $m, \widetilde{m} \in \mathcal{F}$ satisfying $\sqrt{\sum_{i=1}^{n}(m(z_i, x_i) - \widetilde{m}(z_i, x_i))^2} \leq \epsilon$ also satisfies $m(z, x) - \widetilde{m}(z, x) < \epsilon$. Further, $(z, x)$ is $\epsilon$-independent of $\{(z_1, x_1), \ldots, (z_n, x_n)\}$ with respect to $\mathcal{F}$ if $(z, x)$ is not $\epsilon$-dependent on $\{(z_1, x_1), \ldots, (z_n, x_n)\}$.*

**Definition 2** (Eluder dimension [44]). *The $\epsilon$-eluder dimension of a function class $\mathcal{F}$, $dim_E(\mathcal{F}, \epsilon)$, is the length $d$ of the longest sequence of pairs in $\mathcal{Z} \times \mathcal{X}$ such that, for some $\epsilon' \geq \epsilon$, every element is $\epsilon'$-independent of its predecessors.*

The Bayesian regret of an algorithm under the generator-mediated bandit framework measures the regret of the action $A_t$ taken in context $X_t$ against the best action $A_t^*$. As a first step toward bounding the Bayesian regret, we bound the regret of the embedding $Z_t$ generated by the action against the counterfactual embedding $Z_t^*$ that would have been generated by the best action. The analysis adapts the proof of Proposition 1 in Russo and Van Roy [2013] ([44]), and expresses the bound in terms of $w_{\mathcal{C}_t}(Z_t, X_t)$ which measures the maximum variation across the reward functions in the confidence set $\mathcal{C}_t$ at $(Z_t, X_t)$, where

$$
w_{\mathcal{C}}(z, x) := \sup_{m, m' \in \mathcal{C}}(m(z, x) - m'(z, x)).
$$

---

[9]Where $\mathbb{E}_t[W] := \mathbb{E}[W \mid \mathcal{F}_t]$ for any random variable $W$.

[10]For $\mathcal{F}$ consisting of real-valued functions $f : \mathcal{Z} \times \mathcal{X} \to \mathbb{R}$.

**Lemma 5.** *Fix any sequence of confidence sets $\{\mathcal{C}_t\}_{t=1}^T$ where $\mathcal{C}_t \subseteq \mathcal{F}$ is measurable with respect to $\mathcal{G}_{t-1} = \sigma(X_1, A_1, Z_1, Y_1, \ldots, X_{t-1}, A_{t-1}, Z_{t-1}, Y_{t-1}, X_t)$. Then, for any $t = 1, \ldots, T$, we have*

$$\mathbb{E}\left[\sum_{t=1}^T (m_2(Z_t^*, X_t; \theta_2) - m_2(Z_t, X_t; \theta_2))\right] \leq \mathbb{E}\left[\sum_{t=1}^T \left(w_{\mathcal{C}_t}(Z_t, X_t) + 2B \cdot \mathbb{1}_{\{m_2(\cdot, \cdot; \theta_2) \notin \mathcal{F}\}}\right)\right].$$

*Proof.* For any $z \in \mathcal{Z}$ and $x \in \mathcal{X}$, define the upper and lower bounds

$$U_t(z, x) := \sup\{m(z, x) : m \in \mathcal{C}_t\}, \quad L_t(z, x) = \inf\{m(z, x) : m \in \mathcal{C}_t\}.$$

Given any $\theta_2$, if $m_2(\cdot, \cdot; \theta_2) \in \mathcal{C}_t$, then

$$\mathbb{E}\left[m_2(Z_t^*, X_t; \theta_2) - m_2(Z_t, X_t; \theta_2)\right]$$
$$\leq \mathbb{E}\left[U_t(Z_t^*, X_t) - L_t(Z_t^*, X_t) + 2B \cdot \mathbb{1}_{\{m_2(\cdot, \cdot; \theta_2) \notin \mathcal{F}\}}\right]$$
$$\leq \mathbb{E}\left[w_{\mathcal{C}_t}(Z_t, X_t) + 2B \cdot \mathbb{1}_{\{m_2(\cdot, \cdot; \theta_2) \notin \mathcal{F}\}} + (U_t(Z_t^*, X_t) - U_t(Z_t, X_t))\right],$$

By the definition of Thompson sampling, we have $\mathbb{P}[A_t = \cdot \mid \mathcal{G}_{t-1}] = \mathbb{P}[A_t^* = \cdot \mid \mathcal{G}_{t-1}]$, implying $\mathbb{P}[Z_t = \cdot \mid \mathcal{G}_{t-1}] = \mathbb{P}[Z_t^* = \cdot \mid \mathcal{G}_{t-1}]$.

Since $U_t$ is $\mathcal{G}_{t-1}$-measurable, $\mathbb{E}[U_t(Z_t^*, X_t) - U_t(Z_t, X_t) \mid \mathcal{G}_{t-1}] = 0$. It follows that

$$\mathbb{E}\left[\sum_{t=1}^T (m_2(Z_t^*, X_t; \theta_2) - m_2(Z_t, X_t; \theta_2))\right] \leq \mathbb{E}\left[\sum_{t=1}^T \left(w_{\mathcal{C}_t}(Z_t, X_t) + 2B \cdot \mathbb{1}_{\{m_2(\cdot, \cdot; \theta_2) \notin \mathcal{F}\}}\right)\right],$$

concluding the proof. $\square$

Russo and Van Roy [2013] ([44]) bounds the sum of the errors $w_{\mathcal{C}_t}(Z_t, X_t)$, where the confidence set $\mathcal{C}_t$ is centered at the least squares estimate $\widehat{m}_t^{LS}$, where

$$\widehat{m}_t^{LS} := \arg\min_{m \in \mathcal{F}} \sum_{s=1}^{t-1} (m(Z_s, X_s) - Y_s)^2.$$

The bound, presented in Lemma 6, is in terms of the $\alpha_T^{\mathcal{F}}$-eluder dimension, where

$$\alpha_t^{\mathcal{F}} := \max\left\{\frac{1}{t^2}, \ \inf\{\|m_1 - m_2\|_\infty : m_1, m_2 \in \mathcal{F}, m_1 \neq m_2\}\right\}.$$

**Lemma 6** (Lemma 2 in [44]). *If $\{\beta_t\}_{t=1}^T$ is a nondecreasing sequence and $\mathcal{C}_t := \{m \in \mathcal{F} : \|m - \widehat{m}_t^{LS}\|_{2, E_t} \leq \sqrt{\beta_t}\}$, then the following holds almost surely:*

$$\sum_{t=1}^T w_{\mathcal{C}_t}(Z_t, X_t) \leq \frac{1}{T} + B \cdot \min\{dim_E(\mathcal{F}, \alpha_T^{\mathcal{F}}), T\} + 4\sqrt{dim_E(\mathcal{F}, \alpha_T^{\mathcal{F}})\beta_T T}.$$

Russo and Van Roy [2013] ([44]) also shows that, if the confidence width $\beta_t$ is chosen as

$$\beta_t^*(\alpha) := 8\sigma_2^2 \log\left(\frac{\mathcal{N}(\mathcal{F}, \alpha, \|\cdot\|_\infty)}{\delta}\right) + 2\alpha t\left(8B + \sqrt{8\sigma_2^2 \log\left(\frac{4t^2}{\delta}\right)}\right), \qquad (3)$$

where $\mathcal{N}(\mathcal{F}, \alpha, \|\cdot\|_\infty)$ denotes the $\alpha$-covering number of $\mathcal{F}$ with respect to $\|\cdot\|_\infty$, then the associated confidence set (centered at the least squares estimate) contains the true mean reward function $m_2(\cdot, \cdot; \theta_2)$ with high probability. Lemma 7 formalizes this claim.

**Lemma 7** (Proposition 2 in [44]). *For all $\delta > 0$ and $\alpha > 0$, if*

$$\mathcal{C}_t = \left\{m \in \mathcal{F} : \sum_{s=1}^{t-1} (m(Z_s, X_s) - Y_s)^2 \leq \beta_t^*(\alpha)\right\}$$

*for all $t = 1, \ldots, T$, then*

$$\mathbb{P}\left[m_2(\cdot, \cdot; \theta_2) \in \bigcap_{t=1}^T \mathcal{C}_t \mid \theta_2\right] \geq 1 - 2\delta.$$

We next establish a Bayesian regret bound for `poGAMBITTS` in the general reward function setting. As a first step, we analyze the algorithm under the approximate environment $\widetilde{\mathcal{E}}(\theta_2)$. We assume that the given function class $\mathcal{F}$ is sufficiently rich to satisfy $\alpha_T^{\mathcal{F}} = T^{-2}$.

**Lemma 8.** *Consider an environment $\widetilde{\mathcal{E}}(\theta_2)$ with mean reward function $m_2(z, x; \theta_2) \in \mathcal{F}$ for some function class $\mathcal{F}$. The Thompson sampling algorithm run under this environment with a full knowledge of $f_1^{off}$ and a prior $\pi_2$ on $\theta_2$ achieves a Bayesian regret bound of*

$$BR_T \leq \mathcal{O}\left(B \cdot dim_E\left(\mathcal{F}, T^{-2}\right) + \sqrt{dim_E(\mathcal{F}, T^{-2}) \log\left(\mathcal{N}(\mathcal{F}, T^{-2}, \|\cdot\|_\infty)\right) T}\right),$$

*where the Bayesian regret is defined over the distribution $\pi_2$ on $\theta_2$.*

*Proof.* Well, the Bayesian regret for such a Thompson sampling agent can be decomposed as

$$BR_T = \mathbb{E}\left[\sum_{t=1}^{T}\left(\mathbb{E}_{\xi_{X_t, A_t^*}^{off}}[m_2(Z, X_t; \theta_2)] - \mathbb{E}_{\xi_{X_t, A_t}^{off}}[m_2(Z, X_t; \theta_2)]\right)\right]$$

$$= \mathbb{E}\left[\sum_{t=1}^{T}(m_2(Z_t^*, X_t; \theta_2) - m_2(Z_t, X_t; \theta_2))\right]$$

$$+ \mathbb{E}\left[\sum_{t=1}^{T}\mathbb{E}_{\xi_{X_t, A_t^*}^{off}}[m_2(Z, X_t; \theta_2)] - m_2(Z_t^*, X_t; \theta_2)\right]$$

$$- \mathbb{E}\left[\sum_{t=1}^{T}\mathbb{E}_{\xi_{X_t, A_t}^{off}}[m_2(Z, X_t; \theta_2)] - m_2(Z_t, X_t; \theta_2)\right]$$

$$\leq \mathbb{E}\left[\sum_{t=1}^{T}(m_2(Z_t^*, X_t; \theta_2) - m_2(Z_t, X_t; \theta_2))\right] + \mathcal{O}\left(B\sqrt{T \log T}\right),$$

where the final inequality follows from the Azuma-Hoeffding inequality on the martingale difference sequences

$$\left\{\mathbb{E}_{\xi_{X_t, A_t^*}^{off}}[m_2(Z, X_t; \theta_2)] - m_2(Z_t^*, X_t; \theta_2)\right\}_{t=1}^{T}$$

and

$$\left\{\mathbb{E}_{\xi_{X_t, A_t}^{off}}[m_2(Z, X_t; \theta_2)] - m_2(Z_t, X_t; \theta_2)\right\}_{t=1}^{T},$$

both of which are adapted to $\mathcal{G}_t = \sigma(X_1, A_1, Z_1, Y_1, \ldots, X_t, A_t, Z_t, Y_t, X_{t+1})$.

As before, we consider $A_t^* := \arg\max_a \mathbb{E}_{\xi_{X_t, a}^{off}}[m_2(Z, X_t; \theta_2) \mid X_t, \theta_2]$ and we consider $Z_t^*$ sampled from $\xi_{X_t, A_t^*}^{off}$. Considering a fixed sequence of confidence sets $\mathcal{C}_1, \ldots, \mathcal{C}_T$, as defined in Lemma 6, and applying Lemma 5 gives the following bound on the first term:

$$\mathbb{E}\left[\sum_{t=1}^{T}(m_2(Z_t^*, X_t; \theta_2) - m_2(Z_t, X_t; \theta_2))\right] \leq \mathbb{E}\left[\sum_{t=1}^{T}\left(w_{\mathcal{C}_t}(Z_t, X_t) + 2B \cdot \mathbb{1}_{\{m_2(\cdot, \cdot; \theta_2) \notin \mathcal{F}\}}\right)\right]$$

$$\leq \mathcal{O}\left(B \cdot \dim_E\left(\mathcal{F}, \alpha_T^{\mathcal{F}}\right) + \sqrt{\dim_E\left(\mathcal{F}, \alpha_T^{\mathcal{F}}\right) \beta_T^*\left(\alpha_T^{\mathcal{F}}\right) T}\right),$$

where the second inequality follows by Lemma 6 and Lemma 7, with $\delta = \frac{1}{T}$. The desired bound follows by plugging $\alpha = \alpha_T^{\mathcal{F}}$ into the definition of $\beta_T^*(\alpha)$ (Equation 3). $\qquad\square$

Finally, Theorem 2 directly follows by the reduction provided in Theorem 1 and the lemma above.

**Theorem 2** (Restated). *Let $f_1^{off}$ satisfy (2). For any function class $\mathcal{F}$ with $\widehat{m}_2 \in \mathcal{F}$, the Bayesian regret of `poGAMBITTS` (Algorithm 2) satisfies*

$$BR_T^{po:\mathcal{F}} \leq \widetilde{\mathcal{O}}\left(\sigma_2\sqrt{dim_E(\mathcal{F}, T^{-2}) \log \mathcal{N}(\mathcal{F}, T^{-2}, \|\cdot\|_\infty)T} + T\sqrt{\varepsilon T}\right).$$

### E.2.3 Bounding the Error in Offline Treatment Models

poGAMBITTS requires access to an estimate $f_1^{off}(\cdot; a, x)$ that is $\varepsilon$-close to the true treatment density $f_1(\cdot; a, x, \theta_1)$ in KL divergence. Below, we consider the example where $\xi_{x,a}^{\theta_1}$ is Gaussian for each $(x, a) \in \mathcal{X} \times \mathcal{A}$, demonstrating that a sample-based empirical estimate can meet this requirement.

**Lemma 9** (Multivariate Gaussian Distribution). *Consider a multivariate normal distribution $\mathcal{N}(\mu, \Sigma)$ and let $\{Z_1, \dots, Z_n\}$ be i.i.d. samples drawn from this distribution. Then, the estimated distribution $\mathcal{N}(\widehat{\mu}, \widehat{\Sigma})$ satisfies*

$$KL\left(\mathcal{N}(\mu, \Sigma) \| \mathcal{N}\left(\widehat{\mu}, \widehat{\Sigma}\right)\right) \leq \mathcal{O}\left(\frac{d}{\lambda_{min}(\Sigma)} \sqrt{\frac{d + \log(1/\delta)}{n}}\right)$$

*with probability at least $1 - \delta$, where $\widehat{\mu}$ is the sample mean and $\widehat{\Sigma}$ is the empirical covariance matrix.*

*Proof.* The KL divergence has the following exact form:

$$KL\left(\mathcal{N}(\mu, \Sigma) \| \mathcal{N}(\widehat{\mu}, \widehat{\Sigma})\right) = \frac{1}{2}\left[\text{tr}\left(\widehat{\Sigma}^{-1}\Sigma\right) + (\widehat{\mu} - \mu)^\top \widehat{\Sigma}^{-1}(\widehat{\mu} - \mu) - d + \log \det \widehat{\Sigma} - \log \det \Sigma\right].$$

We now turn to bounding each term. A standard concentration result for empirical covariance matrices, based on the matrix Bernstein inequality, gives:

$$\|\widehat{\Sigma} - \Sigma\| \leq \sqrt{\frac{d + \log(1/\delta)}{n}}$$

with probability at least $1 - \delta$, where $\| \cdot \|$ is the operator norm. Hence, the inverse $\widehat{\Sigma}$ concentrates and it follows that

$$\|\widehat{\Sigma}^{-1}\| \leq 2\|\Sigma^{-1}\| = \frac{2}{\lambda_{\min}(\Sigma)}$$

for $n \geq \Omega\left(\frac{d + \log(1/\delta)}{\lambda_{\min}(\Sigma)^2}\right)$.

Hence, we can bound

$$\text{tr}\left(\widehat{\Sigma}^{-1}\Sigma\right) = \text{tr}\left(I + \widehat{\Sigma}^{-1}\left(\Sigma - \widehat{\Sigma}\right)\right)$$

$$\leq d + \|\widehat{\Sigma}^{-1}\|\|\Sigma - \widehat{\Sigma}\|$$

$$\leq d + \mathcal{O}\left(\frac{1}{\lambda_{\min}(\Sigma)} \sqrt{\frac{d + \log(1/\delta)}{n}}\right).$$

To bound the log-determinant term, observe that

$$\log \det \widehat{\Sigma} - \log \det \Sigma = \log \det\left(I + \Sigma^{-1/2}\left(\widehat{\Sigma} - \Sigma\right)\Sigma^{-1/2}\right)$$

$$\leq d\|\Sigma^{-1/2}\left(\widehat{\Sigma} - \Sigma\right)\Sigma^{-1/2}\|$$

$$\leq d\|\Sigma^{-1}\|\|\widehat{\Sigma} - \Sigma\|$$

$$\leq \mathcal{O}\left(\frac{d}{\lambda_{\min}(\Sigma)} \sqrt{\frac{d + \log(1/\delta)}{n}}\right).$$

Finally, we bound

$$(\widehat{\mu} - \mu)^\top \widehat{\Sigma}^{-1}(\widehat{\mu} - \mu) \leq \|\widehat{\mu} - \mu\|_2^2 \|\widehat{\Sigma}^{-1}\|$$

$$\leq \mathcal{O}\left(\frac{d + \log(1/\delta)}{n}\right),$$

applying the vector Bernstein inequality to bound $\|\widehat{\mu} - \mu\|_2$.

Combining all the bounds yields

$$KL\left(\mathcal{N}(\mu, \Sigma) \| \mathcal{N}\left(\widehat{\mu}, \widehat{\Sigma}\right)\right) \leq \mathcal{O}\left(\frac{d}{\lambda_{\min}(\Sigma)} \sqrt{\frac{d + \log(1/\delta)}{n}}\right),$$

as required. $\qquad\square$

### E.3 Upper Regret Bound for foGAMBITTS

In this section, we derive a Bayesian regret for foGAMBITTS in the environment $\mathcal{E}(\theta_1, \theta_2)$. The key step in the analysis is that the action selected by Thompson sampling has the same conditional distribution as the optimal action, given the agent's posterior. Formally, let $\mathcal{F}_t = \sigma(X_1, A_1, Z_1, Y_1, \ldots, X_t, A_t, Z_t, Y_t, X_{t+1})$ be the $\sigma$-algebra generated by the interaction up to time $t$. Thompson sampling logic guarantees that the conditional distributions of $A_t^*$ and $A_t$ given $\mathcal{F}_{t-1}$ are identical; i.e.,

$$\mathbb{P}\left[A_t^* = \cdot \mid \mathcal{F}_{t-1}\right] = \mathbb{P}\left[A_t = \cdot \mid \mathcal{F}_{t-1}\right], \quad \text{almost surely.} \tag{4}$$

We will use the following concentration inequality to control deviations of a martingale sequence in the proof of Theorem 3:

**Lemma 10** (Freedman's inequality). *Let $(\mathcal{F}_k)$ be a filtration and let the sequence $\{X_k\}$ of random variables satisfy $\mathbb{E}[X_k|\mathcal{F}_{k-1}] = 0$ and $|X_k| \leq R$ almost surely. Let $S_k = \sum_{i=1}^{k} X_i$ and $V_k = \sum_{i=1}^{k} \mathbb{E}\left[X_i^2 \mid \mathcal{F}_{i-1}\right]$. Then, with probability at least $1 - \delta$, we have, for all $k$, that*

$$|S_k| \leq \sqrt{2V_k \log(2/\delta)} + \frac{2R}{3} \log(2/\delta).$$

We now restate Theorem 3 and proceed with its proof.

**Theorem 3** (Restated). *If $m_2(z, x_t; \theta)$ is linear in $z \in \mathbb{R}^d$, the Bayesian regret of foGAMBITTS (Algorithm 1) satisfies*

$$BR_T^{fo:lin} \leq \widetilde{\mathcal{O}}\left(\sigma_1 \sqrt{CKT} + \sigma_2 d\sqrt{T}\right).$$

*Proof of Theorem 3.* We are interested in bounding the Bayesian regret defined as

$$BR_T = \mathbb{E}\left[\sum_{t=1}^{T} \mathbb{E}_{\xi_{X_t, A_t^*}^{\theta_1}}[m_2(Z_t, X_t; \theta_2)] - \mathbb{E}_{\xi_{X_t, A_t}^{\theta_1}}[m_2(Z_t, X_t; \theta_2)]\right]$$

where $A_t^* = \arg\max_{a \in \mathcal{A}} \mathbb{E}_{\xi_{X_t, a}^{\theta_1}}[m_2(Z, X_t; \theta_2) \mid X_t, \theta_2]$ is the best action in context $X_t$. Under the linear reward model $m_2(z, x; \theta_2) = \langle z, \theta_2 \rangle$, we can write the Bayesian regret as

$$BR_T = \mathbb{E}\left[\sum_{t=1}^{T} \langle \psi_{X_t, A_t^*} - \psi_{X_t, A_t}, \theta_2 \rangle\right]$$

where we define $\psi_{x,a} := \mathbb{E}_{\xi_{x,a}^{\theta_1}}[Z]$, suppressing the superscript $\theta_1$ for convenience.

Since the agent observes the embedding $Z_t$ and receives a noisy reward of the form $Y_t = \langle Z_t, \theta_2 \rangle + \eta_t$, it can estimate $\theta_2$ by regressing $Y_t$ on $Z_t$. Consider the following estimate for $\theta_2$:

$$\widehat{\theta}_t = V_t^{-1} \sum_{s=1}^{t} Y_s Z_s$$

where $V_t = I + \sum_{s=1}^{t} Z_s Z_s^\top$. Lemma 2 guarantees $\|\theta_2 - \widehat{\theta}_t\|_{V_t} \leq \beta$ for all $t = 1, \ldots, T$ with probability at least $1 - \delta$, where $\beta$, dependent on $\delta$, is defined in the lemma. Let $E_t$ be the event that $\|\theta_2 - \widehat{\theta}_{t-1}\|_{V_{t-1}} \leq \beta$, and let $E := \cap_{t=1}^{T} E_t$. Setting $\delta = \frac{1}{T}$ in the definition of $\beta$ gives $\mathbb{P}[E] \geq 1 - \frac{1}{T}$. We can then bound the Bayesian regret under $E^c$ and $E$ separately, as follows.

$$
\begin{aligned}
BR_T &= \mathbb{E}\left[\sum_{t=1}^{T} \langle \psi_{X_t, A_t^*} - \psi_{X_t, A_t}, \theta_2 \rangle\right] \\
&= \mathbb{E}\left[\mathbb{1}_{\{E^c\}} \sum_{t=1}^{T} \langle \psi_{X_t, A_t^*} - \psi_{X_t, A_t}, \theta_2 \rangle\right] + \mathbb{E}\left[\mathbb{1}_{\{E\}} \sum_{t=1}^{T} \langle \psi_{X_t, A_t^*} - \psi_{X_t, A_t}, \theta_2 \rangle\right] \\
&\leq 2 + \mathbb{E}\left[\mathbb{1}_{\{E\}} \sum_{t=1}^{T} \langle \psi_{X_t, A_t^*} - \psi_{X_t, A_t}, \theta_2 \rangle\right] \\
&\leq 2 + \mathbb{E}\left[\sum_{t=1}^{T} \mathbb{1}_{\{E_t\}} \langle \psi_{X_t, A_t^*} - \psi_{X_t, A_t}, \theta_2 \rangle\right]. \tag{5}
\end{aligned}
$$

Before proceeding with the bound, we clarify the probability space governing the random variables $X_t, A_t, Z_t, Y_t$. We adopt the random table model (see Chapter 4.6 in Lattimore and Szepesvári [2020]) and, conditional on $\theta_1$, define the embedding $Z_t(x, a) \sim \xi_{x,a}^{\theta_1}$ independently for each $t = 1, \ldots, T$ and $(x, a) \in \mathcal{X} \times \mathcal{A}$ [31]. We define $Z_t = Z_t(X_t, A_t)$ to be the embedding observed by the agent. This random table model provides a convenient framework for the analysis that follows.

To proceed, we define the upper confidence bound function $U_t : \mathcal{Z} \to \mathbb{R}$ by

$$U_t(z) = \langle z, \widehat{\theta}_{t-1} \rangle + \beta \|z\|_{V_{t-1}^{-1}}.$$

Then, under the event $E_t$, it follows that, for all $(x, a) \in \mathcal{X} \times \mathcal{A}$,

$$\langle \psi_{x,a}, \theta_2 \rangle - U_t(Z_t(x, a)) = \langle \psi_{x,a}, \theta_2 \rangle - \langle Z_t(x, a), \widehat{\theta}_{t-1} \rangle - \beta \|Z_t(x, a)\|_{V_{t-1}^{-1}}$$

$$= \langle \psi_{x,a} - Z_t(x, a), \theta_2 \rangle + \langle Z_t(x, a), \theta_2 - \widehat{\theta}_{t-1} \rangle - \beta \|Z_t(x, a)\|_{V_{t-1}^{-1}}$$

$$\leq \langle \psi_{x,a} - Z_t(x, a), \theta_2 \rangle.$$

The inequality follows from using Cauchy-Schwarz to bound $\langle Z_t(x, a), \theta_2 - \widehat{\theta}_{t-1} \rangle \leq \|Z_t(x, a)\|_{V_{t-1}^{-1}} \|\theta_2 - \widehat{\theta}_{t-1}\|_{V_{t-1}}$. Furthermore, under the event $E_t$, $\|\theta_2 - \widehat{\theta}_{t-1}\|_{V_{t-1}} \leq \beta$. Similarly,

$$U_t(Z_t(x, a)) - \langle \psi_{x,a}, \theta_2 \rangle = \langle Z_t(x, a), \widehat{\theta}_{t-1} \rangle + \beta \|Z_t(x, a)\|_{V_{t-1}^{-1}} - \langle \psi_{x,a}, \theta \rangle$$

$$= \langle Z_t(x, a), \widehat{\theta}_{t-1} - \theta \rangle + \beta \|Z_t(x, a)\|_{V_{t-1}^{-1}} - \langle \psi_{x,a} - Z_t(x, a), \theta \rangle$$

$$\leq \langle Z_t(x, a) - \psi_{x,a}, \theta \rangle + 2\beta \|Z_t(x, a)\|_{V_{t-1}^{-1}}.$$

Since $U_t(\cdot)$ and $X_t$ are $\mathcal{F}_{t-1}$-measurable, (4) implies

$$\mathbb{P}\left[U_t(Z_t(X_t, A_t^*)) = \cdot \mid \mathcal{F}_{t-1}\right] = \mathbb{P}\left[U_t(Z_t(X_t, A_t)) = \cdot \mid \mathcal{F}_{t-1}\right],$$

and that

$$\mathbb{E}_{t-1}\left[U_t(Z_t(X_t, A_t^*))\right] = \mathbb{E}_{t-1}\left[U_t(Z_t(X_t, A_t))\right],$$

where, as before, $\mathbb{E}_{t-1}[\cdot]$ denotes $\mathbb{E}[\cdot \mid \mathcal{F}_{t-1}]$. Hence, recalling the definition $Z_t = Z_t(X_t, A_t)$,

$$\mathbb{E}_{t-1}\left[\mathbb{1}_{\{E_t\}} \langle \psi_{X_t, A_t^*} - \psi_{X_t, A_t}, \theta_2 \rangle\right]$$

$$= \mathbb{1}_{\{E_t\}} \mathbb{E}_{t-1}\left[\langle \psi_{X_t, A_t^*}, \theta_2 \rangle - U_t(Z_t(X_t, A_t^*)) + U_t(Z_t(X_t, A_t)) - \langle \psi_{X_t, A_t}, \theta_2 \rangle\right]$$

$$\leq \mathbb{1}_{\{E_t\}} \mathbb{E}_{t-1}\left[\langle \psi_{X_t, A_t^*} - Z_t(X_t, A_t^*), \theta_2 \rangle + \langle Z_t - \psi_{X_t, A_t}, \theta_2 \rangle + 2\beta \|Z_t\|_{V_{t-1}^{-1}}\right].$$

Continuing the regret bound from (5) and using $\mathbb{E}_{t-1}\left[\mathbb{1}_{\{E_t\}} \langle \psi_{X_t, A_t^*} - \psi_{X_t, A_t}, \theta_2 \rangle\right] \leq 2$, we observe

$$\mathrm{BR}_T \leq 2 + \mathbb{E}\left[\sum_{t=1}^{T} \mathbb{E}_{t-1}\left[\mathbb{1}_{\{E_t\}} \langle \psi_{X_t, A_t^*} - \psi_{X_t, A_t}, \theta_2 \rangle\right]\right]$$

$$\leq 2 + \mathbb{E}\left[\underbrace{\sum_{t=1}^{T} \mathbb{1}_{\{E_t\}} \langle \psi_{X_t, A_t^*} - Z_t(X_t, A_t^*), \theta_2 \rangle}_{(a)}\right]$$

$$+ \mathbb{E}\left[\underbrace{\sum_{t=1}^{T} \mathbb{1}_{\{E_t\}} \langle Z_t - \psi_{X_t, A_t}, \theta_2 \rangle}_{(b)}\right] + 2\beta \mathbb{E}\left[\underbrace{\sum_{t=1}^{T} (\|Z_t\|_{V_{t-1}^{-1}} \wedge 3)}_{(c)}\right]$$

To bound term $(a)$, we express it as a summation over $(x, a)$:

$$\sum_{t=1}^{T} \mathbb{1}_{\{E_t\}} \langle \psi_{X_t, A_t^*} - Z_t(X_t, A_t^*), \theta_2 \rangle$$

$$= \sum_{x \in \mathcal{X}} \sum_{a \in \mathcal{A}} \sum_{t=1}^{T} \mathbb{1}_{\{E_t\}} \mathbb{1}_{\{X_t = x, A_t^* = a\}} \langle \psi_{x,a} - Z_t(x, a), \theta_2 \rangle.$$

For any given context-action pair $(x, a) \in \mathcal{X} \times \mathcal{A}$, the sequence $\{W_t(x, a)\}_{t=1}^T$, defined by

$$W_t(x, a) = \mathbb{1}_{\{E_t\}} \mathbb{1}_{\{X_t=x, A_t^*=a\}} \langle Z_t(x, a) - \psi_{x,a}, \theta_2 \rangle$$

is a martingale difference sequence adapted to

$$\mathcal{G}_t = \sigma\left(\theta_1, \theta_2, X_1, A_1, \{Z_1(x, a)\}_{x,a}, Y_1, \ldots, X_t, A_t, \{Z_t(x, a)\}_{x,a}, Y_t, X_{t+1}, A_{t+1}\right).$$

This follows because $\mathbb{1}_{\{E_t\}}$ and $\mathbb{1}_{\{X_t=x, A_t^*=a\}}$ are $\mathcal{G}_{t-1}$-measurable, implying

$$\mathbb{E}\left[\mathbb{1}_{\{E_t\}} \mathbb{1}_{\{X_t=x, A_t^*=a\}} \langle Z_t(x, a) - \psi_{x,a}, \theta_2 \rangle \mid \mathcal{G}_{t-1}\right]$$
$$= \mathbb{1}_{\{E_t\}} \mathbb{1}_{\{X_t=x, A_t^*=a\}} \langle \mathbb{E}\left[Z_t(x, a) - \psi_{x,a} \mid \mathcal{G}_{t-1}\right], \theta_2 \rangle$$
$$= 0.$$

Furthermore, $|W_t(x, a)| \le 2$ for all $t = 1, \ldots, T$, and

$$\sum_{t=1}^T \mathbb{E}\left[W_t(x, a)^2 \mid \mathcal{G}_{t-1}\right] \le \sum_{t=1}^T \mathbb{1}_{\{X_t=x, A_t^*=a\}} \mathbb{E}\left[(\langle Z_t(x, a) - \psi_{x,a}, \theta_2 \rangle)^2 \mid \mathcal{G}_{t-1}\right]$$
$$\le \sigma_1^2 \sum_{t=1}^T \mathbb{1}_{\{X_t=x, A_t^*=a\}}$$
$$= \sigma_1^2 N_T(x, a),$$

where $N_T(x, a) = \sum_{t=1}^T \mathbb{1}_{\{X_t=x, A_t^*=a\}}$, as before. The first inequality follows from the assumption that $m_2(Z, x; \theta_2)$ is $\sigma_1$-subgaussian. Hence, Freedman's inequality (Lemma 10) coupled with a union bound over $(x, a) \in \mathcal{X} \times \mathcal{A}$, guarantees that, with probability at least $1 - \delta$,

$$\sum_{t=1}^T W_t(x, a) \le \sigma_1 \sqrt{2 N_T(x, a) \log\left(\frac{2CK}{\delta}\right)} + \frac{4}{3} \log\left(\frac{2CK}{\delta}\right).$$

It follows that

$$\sum_{t=1}^T \mathbb{1}_{\{E_t\}} \langle \psi_{X_t, A_t^*} - Z_t(X_t, A_t^*), \theta_2 \rangle = \sum_{x \in \mathcal{X}} \sum_{a \in \mathcal{A}} \sum_{t=1}^T W_t(x, a)$$
$$\le \sum_{x \in \mathcal{X}} \sum_{a \in \mathcal{A}} \left(\sigma_1 \sqrt{2 N_T(x, a) \log\left(\frac{2CK}{\delta}\right)} + \frac{4}{3} \log\left(\frac{2CK}{\delta}\right)\right)$$
$$\le \sigma_1 \sqrt{2CKT \log\left(\frac{2CK}{\delta}\right)} + \frac{4}{3} CK \log\left(\frac{2CK}{\delta}\right).$$

where the last inequality is by Cauchy-Schwarz and the fact that $\sum_x \sum_a N_T(x, a) = T$. Similar analysis, with $A_t$ in place of $A_t^*$, yields the following bound on the term $(b)$:

$$\sum_{t=1}^T \mathbb{1}_{\{E_t\}} \langle \psi_{X_t, A_t} - Z_t(X_t, A_t), \theta \rangle \le \sigma_1 \sqrt{2CKT \log\left(\frac{2CK}{\delta}\right)} + \frac{4}{3} CK \log\left(\frac{2CK}{\delta}\right).$$

Finally, the last term $(c)$ can be bounded using the elliptical potential lemma (Lemma 3) as

$$\sum_{t=1}^T (3 \wedge \|Z_t\|_{V_{t-1}^{-1}}) \le 3 \sum_{t=1}^T (1 \wedge \|Z_t\|_{V_{t-1}^{-1}})$$
$$\le 3 \sqrt{T \sum_{t=1}^T (1 \wedge \|Z_t\|_{V_{t-1}^{-1}}^2)}$$
$$\le 3 \sqrt{2dT \log\left(1 + \frac{T}{d}\right)}.$$

Combining the bounds above, we observe

$$\mathrm{BR}_T \le 2 + \sigma_1 \sqrt{8CKT \log\left(\frac{2CK}{\delta}\right)} + \frac{8}{3}CK \log\left(\frac{2CK}{\delta}\right) + 6\beta\sqrt{2dT \log\left(1+\frac{T}{d}\right)}$$

$$\le \mathcal{O}\left(\sigma_1 \sqrt{CKT \log\left(\frac{CA}{\delta}\right)} + \sigma_2 d\sqrt{T \log\left(1+\frac{T}{d}\right)\left(\log\left(1+\frac{T}{d}\right) + \log\left(\frac{1}{\delta}\right)\right)}\right.$$

$$\left. + CK \log\left(\frac{CA}{\delta}\right)\right).$$

The result follows. $\qquad\square$

### E.4 Lower Regret Bound for foGAMBITTS

In this section, we establish a minimax lower bound on the regret of foGAMBITTS.

**Claim 2.** *The minimax regret of foGAMBITTS satisfies*

$$R_T \ge \Omega\left(\max\left\{\sigma_1\sqrt{CKT}, \sigma_2\sqrt{dT \log K}\right\}\right).$$

*Proof.* To establish the lower bound, we construct instances in which the agent cannot efficiently resolve uncertainty, leading to provably high regret. We first show $R_T \ge \Omega\left(\sigma_1\sqrt{CKT}\right)$.

Fix $K$, $C$ and $d$, and let $\mathcal{A} = \{1, \ldots, K\}$, $\mathcal{X} = \{1, \ldots, C\}$. Let $\theta_2 = (\sigma_1, 0, \ldots, 0) \in \mathbb{R}^d$ be the parameter for the linear reward model. Furthermore, choose the reward noise $\sigma_2 = 0$.

Let $\theta_1 \in \{\mathcal{A} \cup \{0\}\}^C$ parameterize the distributions $\xi_{x,a}^{\theta_1}$ such that, for each $(x,a) \in \mathcal{X} \times \mathcal{A}$,

$$\xi_{x,a}^{\theta_1} \sim \text{ Bernoulli on } \{e_1, \mathbf{0}\} \text{ with probabilities} \begin{cases} \left(\frac{1}{2}, \frac{1}{2}\right) & \text{if } a \ne \theta_{1,x} \\ \left(\frac{1}{2}-\Delta, \frac{1}{2}+\Delta\right) & \text{if } a = \theta_{1,x} \end{cases},$$

for some $\Delta \in \left[0, \frac{1}{4}\right]$.[11] Here, we use $e_1$ to denote the first basis vector in $\mathbb{R}^d$.

Under this environment, the mean reward of each context-action pair is $\sigma_1$-subgaussian. Also, in context $x$, the mean reward of every action is $\frac{\sigma_1}{2}$ except for one optimal action $a = \theta_{1,x}$, which gives mean reward $\frac{1}{2}(\sigma_1 + \Delta)$.

Because rewards in different contexts are generated from independent distributions, observations collected in one context carry no statistical information about any other context. Hence, from the learner's point of view, the problem decomposes into $C$ independent instances of standard $K$-armed bandits, each with horizon $N_T(1), \cdots, N_T(C)$, where $N_T(x)$ denotes the number of rounds in which context $x$ appears.

For a context $x$, the regret lower bound over $N_T(x)$ rounds is $c\sigma_1\sqrt{KN_T(x)}$ for some constant $c$ (see Chapter 15.2 in Lattimore and Szepesvári [2020]) [31]. Summing over $x$, and choosing $N_T(x) = \frac{T}{C}$ to ensure the environment chooses contexts evenly, we obtain

$$R_T \ge \Omega\left(\sum_{x \in \mathcal{X}} \sqrt{KN_T(x)}\right) = \Omega\left(\sigma_1\sqrt{CKT}\right).$$

We now show that $R_T \ge \Omega\left(\sigma_2\sqrt{dT \log K}\right)$. Consider an environment where $\xi_{x,a}^{\theta_1}$ is deterministic and known to the learner. This reduces to a standard contextual linear bandit environment with dimension $d$. Applying a known lower bound for this setting [32], we obtain

$$R_T \ge \Omega\left(\sigma_2\sqrt{dT \log K}\right).$$

---

[11]Note that $\{\mathcal{A} \cup \{0\}\}^C$ here denotes the $C$-dimensional space with components elements of $\{\mathcal{A} \cup \{0\}\}$, rather than the complement of $\{\mathcal{A} \cup \{0\}\}$.

Combining the two lower bounds yields

$$R_T \geq \Omega\left(\max\left\{\sigma_1\sqrt{CKT}, \sigma_2\sqrt{dT\log K}\right\}\right),$$

as required.

□

## F   Further Simulations

This appendix presents additional simulations designed to probe specific factors influencing GAM-BITTS performance. Unless otherwise specified, all data-generating environments assume a linear outcome model with optimism, formality, and severity as predictors.

### F.1   Variance Decomposition

Section 5 and Appendix E discuss GAMBITTS performance in terms of the relative variance of the treatment and outcome processes. In our setup, the treatment is generated by an LLM and fixed in advance, but we can control the residual variance of the outcome-generative model. This allows us to manipulate the ratio of treatment variance to outcome noise directly. Throughout the main simulations, we held this ratio fixed by setting $\sigma_1 = \sigma_2$.[12] In Figure 7, we increase $\sigma_2$ to explore how performance changes under high outcome noise. We scale $\sigma_2$ gradually, up to the level observed when fitting the outcome-generative model on held-out data from the 2023 IHS study.

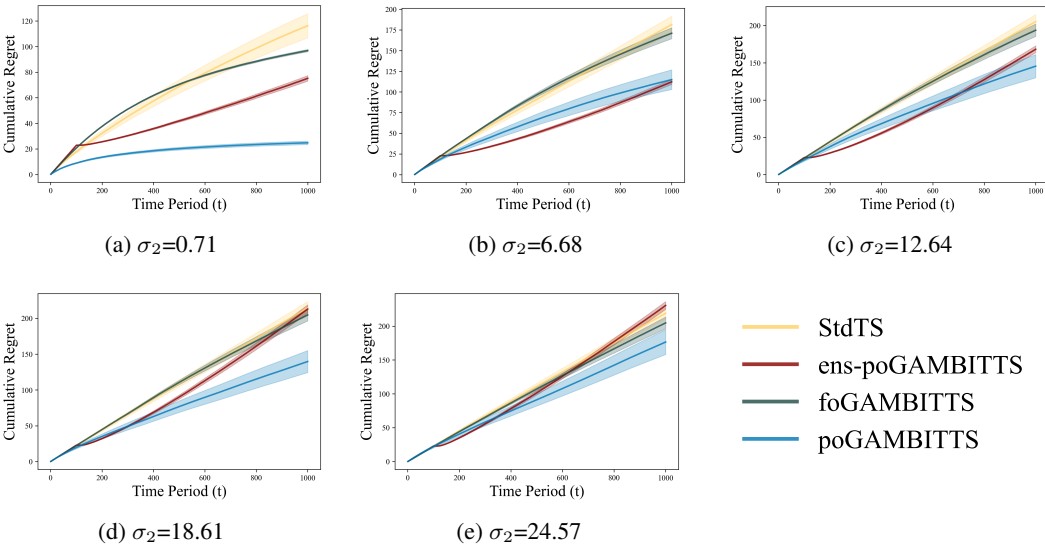

Figure 7: Cumulative Regret Across Outcome Variances

Corollary 1 and Theorem 3 suggest that the performance advantage of linear GAMBITTS-based algorithms over standard Thompson sampling increases with the outcome noise level $\sigma_2$. Figure 7 shows that linear GAMBITTS-based methods achieve lower regret across values of $\sigma_2$; however, as $\sigma_2$ increases, all algorithms exhibit performance deterioration and begin to converge. One possible explanation is that higher outcome noise reduces the overall signal-to-noise ratio, making it more difficult for *any* bandit agent to quickly learn a good policy. Future work may examine this phenomenon over longer time horizons to better understand the asymptotic behavior of the competing methods.

---

[12]We computed $\sigma_1$ from the generated text interventions used for outcome simulation; i.e., from the `response_db` described in Appendix G.

## F.2 Nonlinear Data-Generative Reward Model

Section 6.2 examined GAMBITTS performance under style embedding misspecification. Figure 8 explores a related question, focusing instead on misspecification in the structure of the outcome model. In these experiments, all GAMBITTS agents used the correct style embeddings, but the true outcome-generating process $\mathbb{E}[Y \mid Z, X]$ followed an MLP structure. To assess how GAMBITTS agents respond to this nonlinear specification, we revisit the variance-scaling analysis from Figure 7, now under an MLP outcome model.

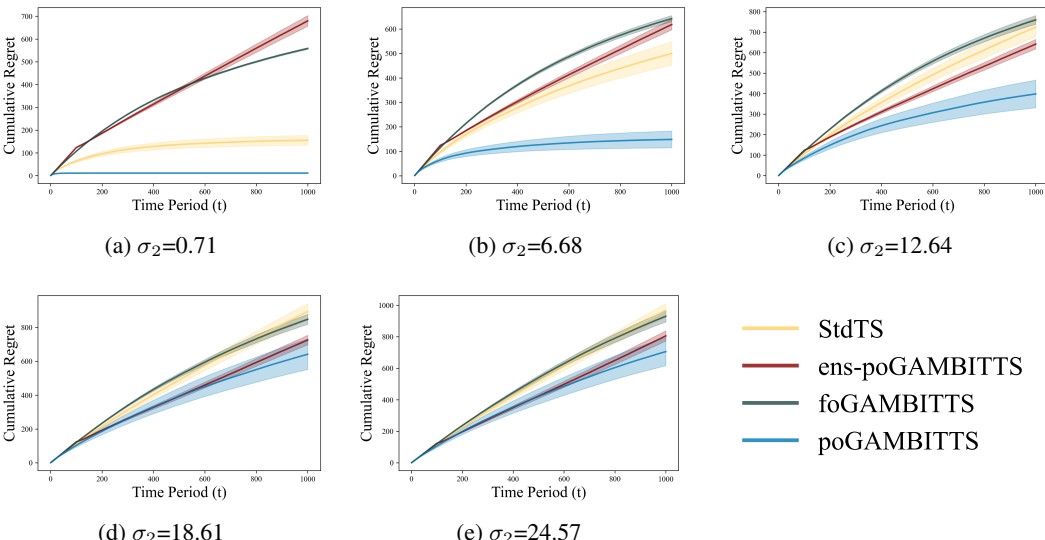

Figure 8: Cumulative Regret Across Outcome Variances: MLP Outcome Generative Model

In a nonlinear data-generating environment, we expected `ens-poGAMBITTS` to outperform its linear counterparts, given its more flexible outcome model. However, as in other simulation settings, `ens-poGAMBITTS` exhibits volatile performance. Interestingly, the linear `poGAMBITTS` agent performs relatively well, despite the misspecification of the outcome model. One possible explanation for this pattern, and for the broader volatility observed in `ens-poGAMBITTS` performance, is that, within the GAMBITTS framework (and in Thompson sampling more generally), the outcome model is used to support decision-making rather than to produce accurate predictions. As described in Appendix G.2, the neural network models are substantially more parameterized than their linear counterparts, and may require far more data to converge. In contrast, while the linear models are misspecified, they may still capture sufficient structure to support effective decision-making.

The volatility in `ens-poGAMBITTS` performance suggests that the approach holds promise, but it is not a panacea. Without tuning tailored to the specifics of the application, the agent may perform poorly. Future work could develop principled strategies for hyperparameter selection to make `ens-poGAMBITTS` more robust to variation in the data-generating process.

## F.3 `poGAMBITTS` Simulator Access

As discussed in Section 4, partially online GAMBITTS variants use simulator access to construct $f_1^{off}$, an empirical estimate of the treatment-generating distribution. Figure 9 examines the sensitivity of `poGAMBITTS` and `ens-poGAMBITTS` performance to the number of samples per $(x, a)$ pair used to construct $f_1^{off}$.

We expected the partially online GAMBITTS variants to perform better with increased access to the treatment-generating distribution. Figure 9 supports this hypothesis: cumulative regret decreases as the number of simulator draws increases. Interestingly, performance stabilizes after roughly 50 draws, suggesting that limited simulation access may be sufficient to realize most of the benefits of the partially online approach.

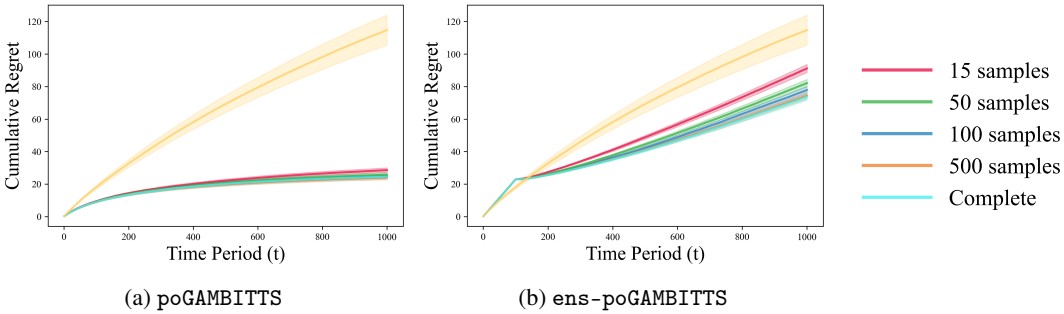

(a) poGAMBITTS   (b) ens-poGAMBITTS

Figure 9: Cumulative Regret Under Varying Levels of Simulator Access

## F.4 Treatment Dimension

We also examine the effect of $d$, the dimensionality of the treatment embedding. To do so, we introduce 15 additional style dimensions, supplementing the five used in Section 6, for a total of 20 dimensions: optimism, formality, encouragement, severity, clarity, humor, complexity, vision, detail, threat-level, urgency, politeness, personalization, conciseness, actionability, emotiveness, authoritativeness, authenticity, supportiveness, and gender-codedness.[13]

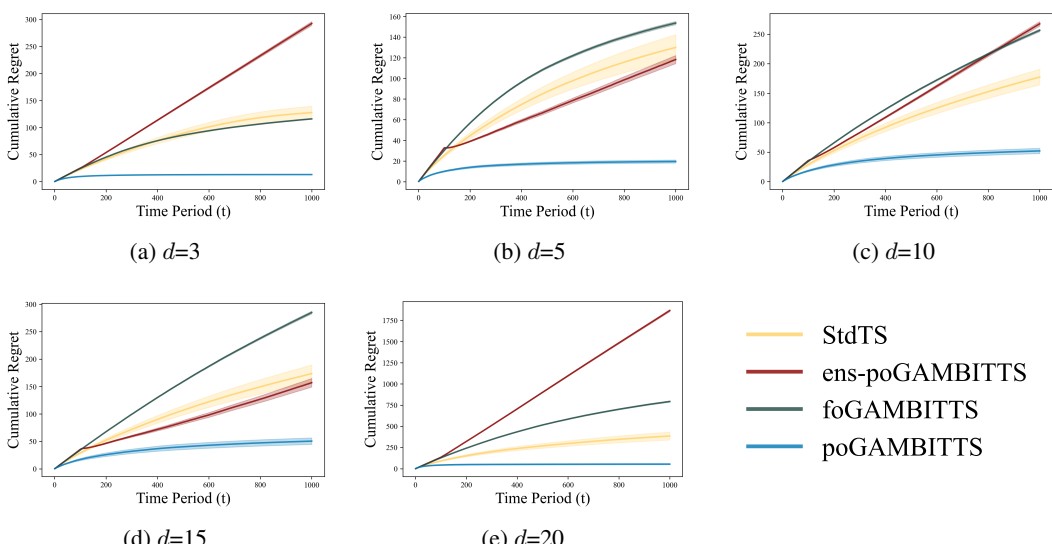

(a) $d$=3   (b) $d$=5   (c) $d$=10

(d) $d$=15   (e) $d$=20

Figure 10: Cumulative Regret for Varying Treatment Embedding Dimension

We anticipated that GAMBITTS agents would perform increasingly poorly relative to standard Thompson sampling as the embedding dimension $d$ increased. Figure 10 confirms this trend for the foGAMBITTS agent. However, the linear poGAMBITTS agent remains competitive even as $d$ grows. As discussed in Appendix F.2, the volatility in ens-poGAMBITTS performance further underscores the need for future work on robust hyperparameter tuning strategies within the GAMBITTS framework.

## F.5 Direct Contextual Influence on Reward

The simulations in Sections 6.1 and 6.2 (along with Appendix C) use reward models linear in $Z^{\text{optimism}}$ with Gaussian noise. As described in Section 6.3, the other data-generating environments

---

[13]To select the dimensions used in Figures 10a-10e, we clustered the style dimensions by their correlation in response_db. For each $d$, we used $d$-means clustering and randomly sampled one dimension from each cluster.

extend this structure to include $Z^{\text{optimism}}$, $Z^{\text{formality}}$, and $Z^{\text{severity}}$, again with Gaussian noise.[14] This appendix considers a simulation designed to evaluate GAMBITTS performance when rewards depend directly on context as well as semantic dimensions.[15] Given the substantial performance advantage of GAMBITTS agents over standard Thompson sampling in our main experiments, we expect this advantage to persist under additional covariate influence.

In this appendix, we extend the setup from Figure 4b[16] by allowing rewards to depend directly on context. As described in Appendix G, the simulated contextual variables were current location and steps taken the previous day. Specifically, we use the data-generating model[17]

$$Y_t = \beta^{\text{optimism}} Z_t^{\text{optimism}} + \beta^{\text{formality}} Z_t^{\text{formality}} + \beta^{\text{severity}} Z_t^{\text{severity}}$$
$$+ \beta^{\text{steps}} X_t^{\texttt{stepsprevday}} + \sum_{\text{loc} \in Locs} \beta^{\text{loc}} \mathbb{1}_{X_t^{\texttt{currloc}} = \text{loc}} + \varepsilon_t.$$

Figure 11 reports results under correctly specified GAMBITTS agents. The figure supports the intuition that the advantage of GAMBITTS persists under direct covariate influence, showing little deviation from Figure 4b.[18]

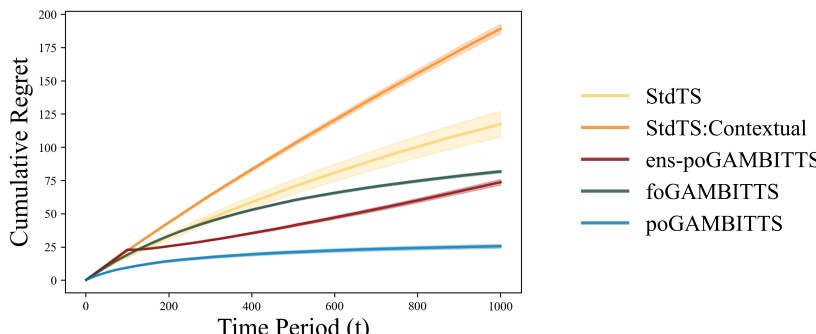

Figure 11: Cumulative Regret Under Reward Model with Direct Context Influence

## F.6 Scaling Number of Arms: Revisited

As observed in Section 6.3, `foGAMBITTS` underperforms standard Thompson sampling when the number of available arms, $K$, is large. However, the more appropriate comparison is to a contextual Thompson sampler, since `foGAMBITTS` incorporates covariate information through the treatment embedding. Moreover, Section 5 shows that `foGAMBITTS` is expected to outperform standard Thompson sampling when $\sigma_2 \gg \sigma_1$, whereas the experiments in Section 6.3 assume $\sigma_1 = \sigma_2$. To investigate this further, we examine `foGAMBITTS` performance with $K = 40$ under varying levels of outcome noise.

---

[14]With two exceptions: Appendix F.2 instead uses an MLP mean reward model, and Appendix F.4 varies the number of semantic dimensions in the mean reward model.

[15]In all cases, rewards depend on text indirectly through the semantic dimensions. Semantic dimensions are functions of LLM-produced text. Although other data-generative environments do not specify a direct effect of context on reward, context enters implicitly through the LLM query, which shapes the generated text and hence the semantic dimensions (described further in Appendix G).

[16]We use Figure 4b as the baseline because it employs the default choices for $(i)$ the number of arms, $(ii)$ the set of style dimensions, and $(iii)$ the reward noise level used throughout the experiments in this manuscript.

[17]Here $Locs := \{$"home", "work", "a location other than home or work"$\}$ and $X^{\texttt{stepsprevday}}$ was recoded to be 1, 2, 3, or 4 if the simulated previous day's steps was 0-4,999, 5,000-9,999, 10,000-15,000, or more than 15,000, respectively.

[18]The standard contextual Thompson sampler continues to underperform in this setting. Incorporating covariates requires a more heavily parameterized model for the contextual Thompson sampling agent than for its non-contextual counterpart. When covariate effects are modest, this added complexity increases variance and can reduce performance (a known issue in contextual bandit learning [47]). Furthermore, in this simulation, the optimal action does not vary across context, limiting the potential gain from contextual information. Further work could examine GAMBITTS performance under different forms and strengths of contextual influence.

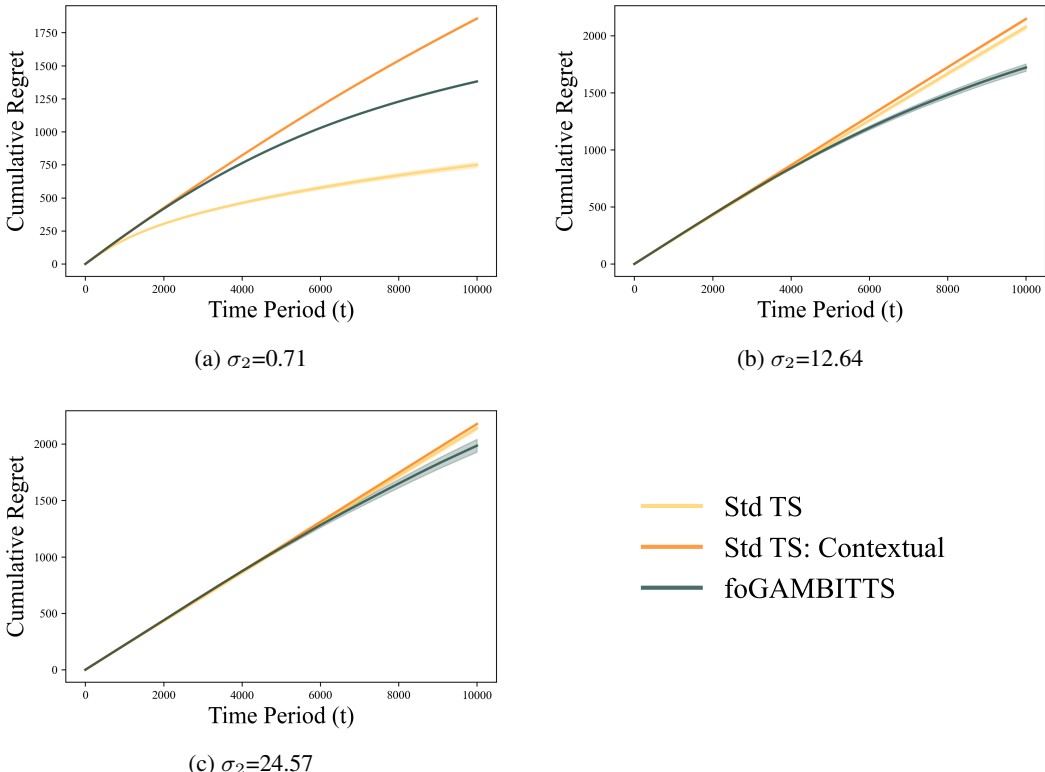

Figure 12: Cumulative Regret for `foGAMBITTS` with $K = 40$ under Varying Reward Noise

Figure 12 supports the results in Section 5: `foGAMBITTS` performs increasingly well relative to standard Thompson sampling as $\sigma_2$ grows. Moreover, even for small $\sigma_2$, `foGAMBITTS` outperforms a contextual Thompson sampling agent. To better observe asymptotic behavior under high noise, we extend the time horizon to 10,000 rounds. Even so, cumulative regret does not appear to level off, suggesting that the agents are still learning the optimal policy. This persistent learning phase likely reflects the low signal-to-noise ratio estimated from the IHS study that generated the outcome parameters. Future research directions to address such scenarios could incorporate partial pooling methods (e.g., Huch et al. [2020]) to share reward model information across users and accelerate policy learning [20].

## G    Simulation Study Design

To evaluate algorithm performance in a setting where ground truth is available, we needed full knowledge of the data-generating process. This ruled out querying a black-box language model during simulation runs, since doing so would obscure the underlying conditional expectations and make performance comparisons difficult to interpret. Instead, we simulated on-the-fly text generation by constructing a response database (referred to as `response_db`): for each prompt-context pair, we sampled 1,000 responses from an LLM[19] and computed the style embeddings associated with each response, as detailed in Appendix G.3. At runtime, we drew from this pool to mimic LLM-generated responses, as described in Algorithm 6.

---

[19]In particular, Llama 3.1 with 8.0B parameters, 131,072 context length, 2,095 embedding length, and Q4_K_M quantization, accessed via Ollama [42]. Additionally, we re-ran select analyses using a Qwen-based LLM. The results were consistent with those in the main manuscript, suggesting that the simulation conclusions are not sensitive to the choice of LLM, though further work would be needed to establish this more formally.

We derived the outcome-generating model using data from the 2023 IHS study.[20] For each outcome model structure (linear or neural network) and relevant style dimensions, we fit the corresponding model to the IHS data and used the estimated parameters to define the outcome surface. This provided a data-driven but controlled setting for comparing policies.[21] We provide the exact parameters used in the data-generating process on GitHub.

Lastly, in our simulations, we included two context variables:

$$\texttt{stepsprevday} \in \{\text{"0-4,999", "5,000-9,999", "10,000-15,000", "more than 15,000"}\} \text{ and }$$
$$\texttt{currloc} \in \{\text{"home", "work", "a location other than home or work"}\}.$$

These context variables varied uniformly across time points and were sampled independently of the action.

---

**Algorithm 6** `Pseudocode for Outcome Generation`

---

**Inputs**: `response_db`, style embedding functions $f_1, \ldots, f_D$, mean model structure $m_2(z, x; \theta_2)$, data-generative outcome model parameters $\theta_2^{IHS}$, conditional variance $\sigma_2^2$.

1: **Given** a prompt $a$ and context $x$, randomly sample response $g$ from `response_db`, restricted to responses generated from $(a, x)$.
2: **Compute** style embedding vector $z = [f_1(g) \ldots f_D(g)]^\top$.
3: **Simulate** outcome $y$ by drawing $\varepsilon \sim \mathcal{N}(0, \sigma_2)$ and setting $y = m_2(z, x; \theta_2^{IHS}) + \varepsilon$.

---

### G.1 Prompt Specifications

To initialize the LLM environment for crafting intervention text, we provided the following system prompt:

> You are tasked with writing a single message to an individual. The message should be two to three sentences long and will be delivered to the individual through a mobile app. The intent of the message is to encourage the individual to take more steps, with the ultimate goal of improving their physical health. The app collects the following information on its users: whether they took 0-4,999, 5,000-9,999, 10,000-15,000, or more than 15,000 steps the previous day, and whether they are currently at home, at work, or at another location.

To generate the intervention text used in our simulations, we constructed prompts that explicitly manipulated specific stylistic dimensions. Each prompt was designed to target a particular user context and asked the LLM to produce a message in which a given dimension (e.g., optimism, formality) was either high or low. This approach yielded a total of 40 prompts (one high and one low version for each of the 20 dimensions) attempting to promote alignment among the variation in the generated text with the interpretable axes used in the outcome model.

To generate simulation messages (for given contexts `stepsprevday` and `currloc`), we used the following prompt structure:

> The app provides the following context about the individual: they took `stepsprevday` steps the previous day and are currently at `currloc`. This context may be included in the message but does not need to dominate it. Please make the message mildly `<style polarity>`. `<dimension description>`. Please do not include anything in your response other than the text of the message.

Here, `<style polarity>` refers to the high/low level of the target stylistic dimension (e.g., "optimistic"/"pessimistic" for the optimism dimension). `<dimension description>` is a descriptor

---

[20]We used the square root of the number of steps taken by the user on the day following the message as the reward. This transformation, rather than using raw step counts, yielded approximately normally distributed errors.

[21]The one exception is our use of $\sigma_1 = \sigma_2$ in the main simulations. We observed $\sigma_1$ from the variability in `response_db` and explored sensitivity to larger values of $\sigma_2$ in Appendix F.1.

of the stylistic quality, intended to guide the LLM toward the appropriate tone.[22] Unless otherwise noted (e.g., Section 6.3), all experiments were run using five prompts available to the agent, chosen to be evenly distributed by expected outcome.[23]

## G.2 Agent Specifications

We evaluate several bandit agents in Section 6 and Appendix F. Unless otherwise noted, each agent type followed the base specifications described in the corresponding subsection below. Furthermore, all GAMBITTS-based algorithms used a correctly specified mean outcome model (unless otherwise noted; e.g., Section 6.2). When possible, we used data from the 2022 IHS study to inform prior specification. For example, the average square root of daily steps was approximately 77, which guided our choice of prior for intercept terms.

### G.2.1 Standard Thompson Samplers

The standard Thompson sampling agents modeled the reward for each action using a normal likelihood, with a prior of the form $\mu_a \sim \mathcal{N}\left(77, \ \sigma_a^2\right)$, where $\sigma_a^2 \sim$ Inverse-Gamma$(1, 10)$. The contextual Thompson sampling agents used the same model and priors, but indexed them by each context–action pair $(x, a)$.

### G.2.2 `poGAMBITTS` Agent

Linear `poGAMBITTS` agents modeled the reward as $Y \sim \mathcal{N}(\beta_0 + Z^\top \beta', \Sigma)$ for $Z, \beta' \in \mathbb{R}^d$ and $\Sigma \in \mathbb{R}^{d \times d}$. Letting $\beta := [\beta_0 \ \ \beta']^\top \in \mathbb{R}^{d+1}$, these agents placed the following priors on model parameters:

$$\theta_2 \sim \mathcal{N}\left(\mu_0, \ B_0\right)$$
$$\mu_0 = [77 \ \ 0 \ \ldots \ 0]^\top \in \mathbb{R}^{d+1}$$
$$B_0 = \text{diag}(10^{-2}, 1, \ldots, 1) \in \mathbb{R}^{(d+1) \times (d+1)}.$$

### G.2.3 `ens-poGAMBITTS` Agent

Our implementation of `ens-poGAMBITTS` uses PyTorch [1], with each neural network instantiated using PyTorch's default initialization: weights and biases were drawn from a uniform distribution $\mathcal{U}\left(-\dfrac{1}{\sqrt{k}}, \dfrac{1}{\sqrt{k}}\right)$, where $k$ is the number of input features. The ensembles consisted of $M_{ens} = 60$ networks. Each network was single-layer feedforward model with 64 hidden units and ReLU activation. Online training was performed with a batch size of 100 and a learning rate of 0.1. The ensembles maintained a replay buffer of size 1,024. To approximate the expectation in Algorithm 4, we used 100 Monte Carlo samples. In each experiment, `ens-poGAMBITTS` training began after a burn-in period of $t = 100$ steps, during which the neural networks did not update, allowing sufficient data to accumulate for batch-based optimization. Lastly, the agents used the data-generative $\sigma_2$ as the perturbation noise standard deviation.

### G.2.4 `foGAMBITTS` Agent

`foGAMBITTS` agents used the same prior $\pi_2$ on $\theta_2$ as defined for the `poGAMBITTS` agent in Appendix G.2.2 above. `foGAMBITTS` agents used a normal likelihood for $Z \in \mathbb{R}^d$ with mean $\theta_1 \in \mathbb{R}^d$. The prior $\pi_1$ for $\theta_1$ is given below:

$$\Sigma_1 \sim \text{Inverse-Wishart}\left(\mathbf{I}_d, 1\right)$$
$$\theta_1 \sim \mathcal{N}\left(\mathbf{0}_d, \ \Sigma_1\right),$$

where $\mathbf{I}_d \in \mathbb{R}^{d \times d}$, $\mathbf{0}_d \in \mathbb{R}^d$, and $d$ is the dimension of the latent treatment space $\mathcal{Z}$.

---

[22]For example, the following is the description of the optimism dimension: "Here, optimism is defined as the degree of hopeful or confident language in the message. A lower optimism level may include a more matter-of-fact or pessimistic tone, while a higher optimism level may include uplifting and buoyant language."

[23]E.g., if an experiment included only three prompts, we selected the highest, lowest, and median prompt based on true expected value under the data-generating process.

## G.3  Text Embedding Construction

We aimed to quantify LLM-generated responses along five stylistic dimensions: optimism, encouragement, formality, clarity, and severity.[24] For each dimension, $d$, we constructed a mapping $f_d : \mathcal{G} \to \mathbb{R}$, as described in Algorithm 7 below. This appendix details the construction of these embeddings and provides evidence that they capture the intended stylistic dimensions. This evidence is included primarily to aid interpretability of the simulation experiments. While high-quality embeddings are crucial in applied settings, they are not required for demonstrating the validity of the GAMBITTS approaches.

We learned each $f_d$ by prompting an LLM to generate a set of JITAI-style messages (brief behavioral messages aimed at encouraging users to walk more, mirroring the setup of the IHS and HeartSteps studies) [33, 40]. Appendix G.3.1 below contains the exact prompt template used for this purpose. For each prompt, we asked the LLM to vary the degree to which its response reflected a particular stylistic dimension ($d$). The responses were intended to isolate variation along that dimension while holding other properties relatively constant. The goal was not to assign explicit scores or labels, but to induce embeddings in which each axis captured targeted semantic variation.

We then trained a variational autoencoder (VAE) on the resulting text set, constraining the model to use a single latent dimension. The output of this latent space was used as the embedding $f_d(g)$ for each message $g$.[25] This process was repeated separately for each dimension. Algorithm 7 below provides pseudocode for this process.

---

**Algorithm 7** `Pseudocode for Style Dimension` $d$ `Embedding Construction`

---

**Input**: Style dimension label $d$, LLM, high-dimensional embedding model, VAE architecture, training size $M_{train}^d$.

1: **Generate** $M_{train}^d$ random prompts $\{p_i^d\}_{i=1}^{M_{train}^d}$ using the template presented in Appendix G.3.1.
2: **For** each prompt, $p_i^d$, in Step 1, query the LLM with the prompt to receive $r_i^d$
3: **For** each response, $r_i^d$ in Step 2, obtain a high-dimensional numerical embedding ($e_i^{d,h}$) by using the high-dimensional embedding model [55].
4: **Given** embeddings $e_1^{d,h}, \ldots, e_{M_{train}^d}^{d,h}$, train a variational auto-encoder to obtain a one-dimensional latent representation of the response [6]. Call the resulting mapping $f_d$.

---

As with the outcome generative model, we used Llama 3.1 as the LLM for embedding construction. Additionally, we used BERT as our high-dimensional embedding model [10], and set $M_{train}^d = 22,000$.

### G.3.1  Embedding Generation Prompt Creation

To initialize the LLM environment, we provided a system prompt describing the context and purpose of the message generation task (e.g., encouraging physical activity through behavioral messages). This global instruction was held fixed across all prompt constructions and consisted of the following:

> You are tasked with writing eleven separate messages to an individual. Each message should be two to three sentences long and will be delivered to the individual through a mobile app. The messages should be independent and must not reference each other. The intent of each message is to encourage the individual to take more steps, with the ultimate goal of improving their physical health. The app collects the following information on its users: whether they took 0-4,999, 5,000-9,999, 10,000-15,000, or more than 15,000 steps the previous day, and whether they are currently at home, at work, or at another location.

To simulate messages that varied along a given stylistic axis, we used a template-based user prompt, shown below. The specific wording of the prompt varied by stylistic dimension; we present the version used for *optimism* as an illustrative example.

---

[24]Along with the additional 15 dimensions for the experiments in Appendix F.4.

[25]We note that the messages used to train the VAE were distinct from those contained in the `response_db` database used for outcome simulation.

> The app provides the following context about the individual: they took `stepsprevday` steps the previous day and are currently at `currloc`. This context may be included in the messages but should not dominate them.
>
> The primary axis of variation in the messages should be optimism, with no intentional variation along other dimensions such as tone, length, or formality. For this task, create eleven messages where the optimism level varies as follows: Message 1 should represent the least optimistic tone (optimism level 0). Message 11 should represent the most optimistic tone (optimism level 10). Messages in between should gradually and evenly increase in optimism. Here, optimism is defined as the degree of hopeful or confident language in the message. A lower optimism level may include a more matter-of-fact or pessimistic tone, while a higher optimism level may include uplifting and buoyant language. Remember, low levels of optimism should have a pessimistic tone.
>
> Please write each message on a separate line, and remember that the target length is two to three sentences for each message. Do not include anything in your response other than the text of the messages. Start every new message with its number. The optimism of the message should be independent of the user specific information.

For this purpose, we randomly drew context variables `stepsprevday` and `currloc` uniformly from their possible values.[26] We varied the user context across prompts to encourage the VAE to learn stylistic variation independently of context. For each style dimension $d$, we generated 2,000 responses (each consisting of eleven JITAI messages spanning the stylistic axis) yielding a total of 22,000 messages per dimension.

### G.3.2 Text Embedding Results

After running Algorithm 7, we compared the target input rating with the VAE output. Figure 13 shows this relationship.[27]

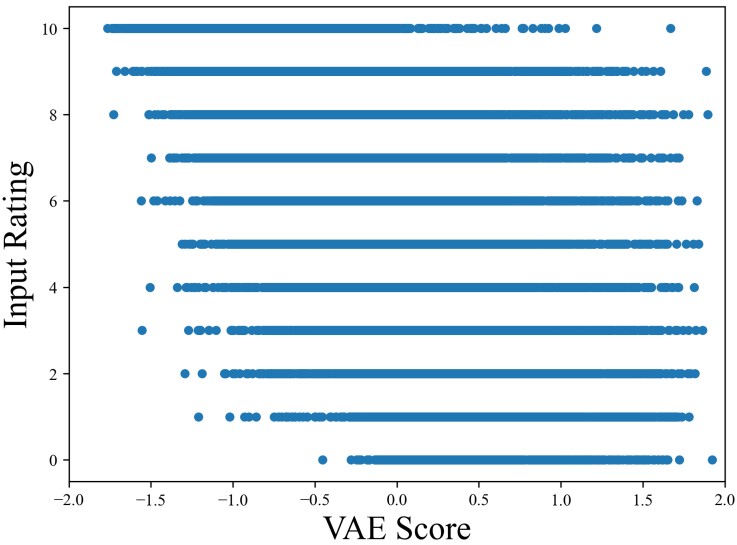

Figure 13: Optimism Scores versus Prompted Rating

We find that the VAE captured some signal related to the intended tone of the message. While the relationship in Figure 13 is noisy, two caveats are worth noting. *(i)* Precisely quantifying variation along a semantic dimension is likely a domain-specific challenge and, while important, it is outside the scope of this work. Our goal here is not to validate the embedding as a standalone

---

[26]We used the same context variables for the embedding construction as for the outcome generation.

[27]Note that we removed two outliers with high VAE scores, one with rating ten and one with rating nine, which distorted the scale of the plot.

construct; nonetheless, alignment between prompted tone and latent score adds interpretability to the simulation setup. *(ii)* Some of the observed noise reflects inconsistencies in LLM behavior: the model occasionally generates responses that do not align with the requested tone. For example, the following message was produced in response to a prompt with optimism rating 1, but appears highly optimistic: "You're on a roll after yesterday's success - keep up the pace and see where it takes you!" Larger LLMs or more carefully tuned unsupervised methods may improve the correspondence between prompted tone and latent score.

### G.3.3 Embedding Construction Summary

Appendix G.3.1 presents the prompt instructions used for generating text that varied in optimism, and Appendix G.3.2 shows how the resulting VAE scores relate to the specified input levels. As illustrated in Figure 13, the VAE captures variation along the target semantic dimension, but its unsupervised nature means the sign of the scores is arbitrary (for example, positive scores appear to align with pessimistic rather than optimistic text). Table 1 summarizes the style definitions provided in the prompts for each of the five primary semantic dimensions and reports the styles of text the VAE associated with positive and negative scores.[28]

Table 1: Primary Semantic Styles, Prompted Definitions, and Interpretation of VAE Scores

| Style | Style Definition | VAE Score Interpretations | |
| --- | --- | --- | --- |
| | | Negative | Positive |
| Optimism | "The degree of hopeful or confident language in the message. A lower optimism level may include a more matter-of-fact or pessimistic tone, while a higher optimism level may include uplifting and buoyant language." | Optimistic | Pessimistic |
| Severity | "The degree of dire or drastic language in the message. A lower severity level may include a more lax, calm, or gentle tone, while a higher severity level may include more dark, intense, worrying and distressing language." | Severe | Lax |
| Formality | "The degree of which the message has an objective, academic, or professional tone. A lower formality level may include a more personal, casual, or emotional tone and include colloquial language. A higher formality level may include more matter-of-fact, impersonal, professional and serious language." | Informal | Formal |
| Clarity | "The degree to which the intent of the message is intelligibly communicated. A lower clarity level should be more vague and ambiguous in its language, while a higher clarity level may include more precise, coherent, and intelligible instructions or comments." | Clear | Unclear |
| Encouragement | "The degree of persuasion using positivity, confidence, and hope. A lower encouragement level may include a more depressed, dispirited or hopeless tone, while a higher encouragement level may include more excited, heartening, and motivating language." | Discouraging | Encouraging |

### G.3.4 Relationship between Style Embeddings

Table G.3.4 below shows the empirical correlation (within the `response_db` dataset constructed for outcome generation) among the five stylistic embeddings discussed in Section 6.2.

---

[28]These associations were identified by manually examining a sample of generated text and its corresponding VAE scores.

Table 2: Style Dimension Correlation Table

|  | Optimism | Formality | Severity | Clarity | Encouragement |
|---|---|---|---|---|---|
| Optimism | 1.000 | -0.095 | 0.787 | 0.184 | -0.960 |
| Formality | -0.095 | 1.000 | -0.568 | -0.248 | -0.121 |
| Severity | 0.787 | -0.568 | 1.000 | 0.066 | -0.644 |
| Clarity | 0.184 | -0.248 | 0.066 | 1.000 | -0.201 |
| Encouragement | -0.960 | -0.121 | -0.644 | -0.201 | 1.000 |

Furthermore, we analyzed the structure of the relationship between the different style dimensions. Figure 14 shows pairwise scatterplots of style embeddings for the five primary dimensions. Recall that, in addition to the five primary dimensions, we created 15 more for the experiments in Appendix F.4. The pairwise scatterplots for all 20 dimensions are shown in Figure 15. These figures provide a qualitative view of the dependencies among dimensions. However, the direction of the embeddings is not directly comparable across dimensions; e.g., a high encouragement score corresponds to an encouraging tone, while a high optimism score reflects greater pessimism.

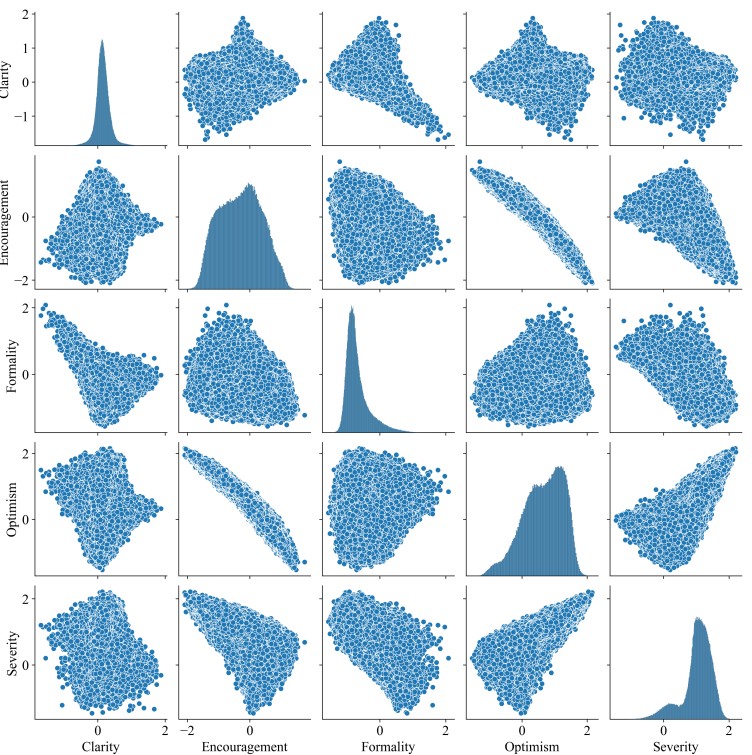

Figure 14: Pairwise Scatterplot of Five Primary Style Dimensions

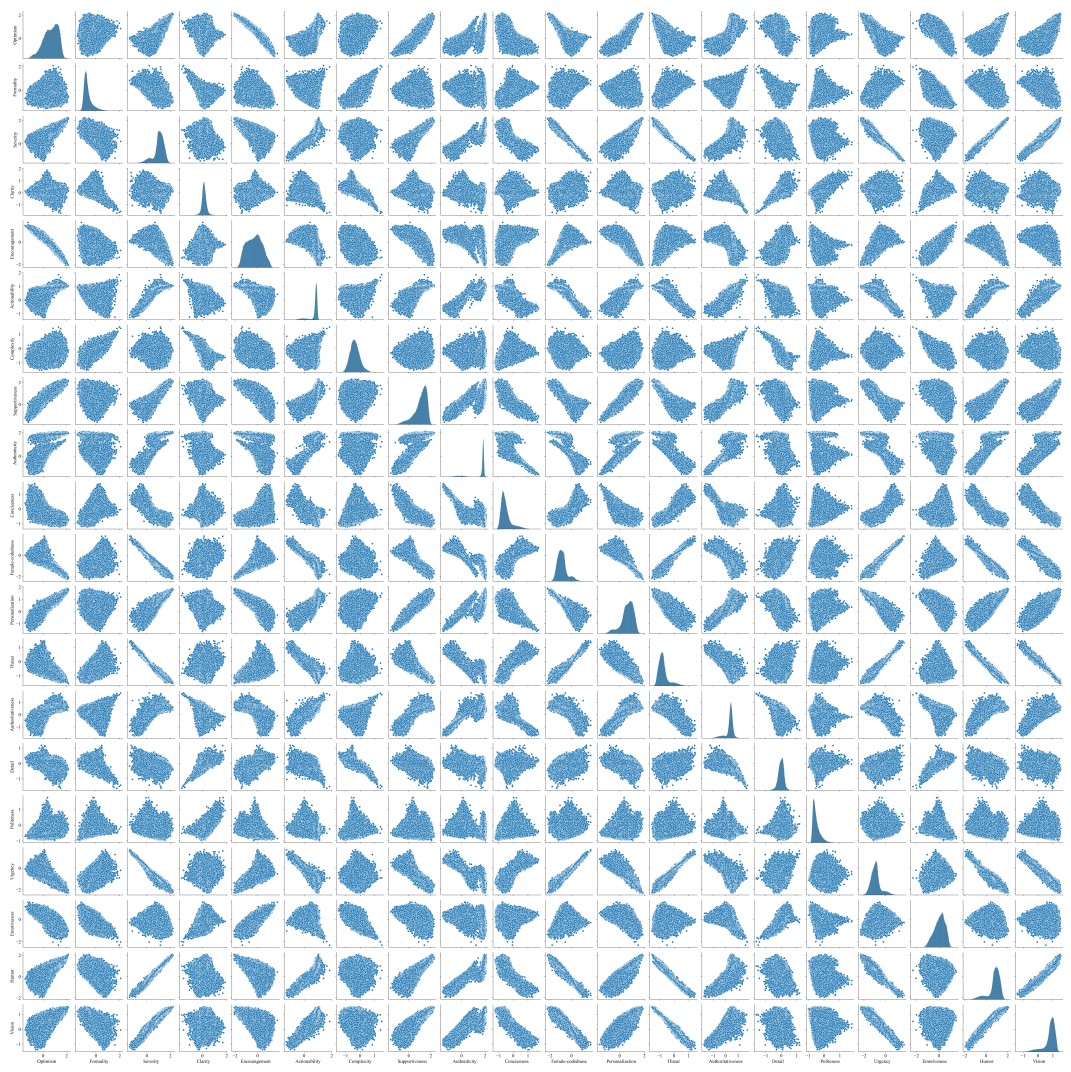

Figure 15: Pairwise Scatterplot of All Style Dimensions

### G.4 Computing Resources

The simulation of text-based treatments generated by Llama 3.1 was implemented on a high-performance compute cluster. Each node in the cluster included two 2.9 GHz Intel Xeon Gold 6226R processors, 8 GB of allocated RAM, and a single NVIDIA A40 GPU with 48 GB of memory. VAE training was performed on a similar setup, except each node was allocated 16 GB of RAM. The simulation studies were conducted on nodes equipped with two 3.0 GHz Intel Xeon Gold 6154 processors, 16 CPU cores, and between 16–32 GB of allocated RAM. Under these specifications, each experiment presented in Section 6 and Appendix F required approximately 50-70 minutes to complete.

## H  Broader Impact

As discussed in Section 1, text-based sequential interventions are used across domains such as mobile health and education to support behavioral change and improve outcomes. We view the GAMBIT framework and associated algorithms as a step toward more personalized decision-making in such settings. At the same time, personalization introduces risks (particularly in sensitive domains) where automatically generated content may influence individuals in subtle or unintended ways. In some applications, it may be inappropriate to deliver GenAI-generated content without human oversight or safety review. We discuss these concerns and potential adaptations to the framework in Section 7, with the hope that variants of GAMBIT can be developed to address such challenges directly.

