# OpenReview forum: "Generator-Mediated Bandits: Thompson Sampling for GenAI-Powered Adaptive Interventions"
_NeurIPS.cc/2025/Conference — NeurIPS 2025 poster_

### Official Review · Reviewer_XiMn · 2025-06-27

**Clarity:** 3
**Significance:** 2
**Originality:** 3
**Rating:** 5
**Confidence:** 4

**Summary:**

The paper introduces the notion of Generator-Mediated Bandits with Thompson Sampling. These bandit methods take into account a two stage problem architecture, where the bandit's output is fed to an LLM along with additional context, and the output of the LLM is what determines the reward. This is primarily motivated by personalized decision-making in mobile health contexts, where study participants are at the receiving end of text based intervention prompts. The proposed method, attributed to as GAMBITTS, takes this two stage problem into account by modeling the action as well as the reward generated - part of which is achieved by projecting the LLM's output into a lower dimension embedding. The paper provides regret bounds and experimental results wrt the methods proposed.

**Questions:**

See above.

**Ethical Concerns:**

["NO or VERY MINOR ethics concerns only"]

**Final Justification:**

I am satisfied with the author rebuttal and the clarifications presented. The authors have also offered to clarify a lot of my concerns in the paper. Hence, I am raising my score.

**Limitations:**

See above.

**Paper Formatting Concerns:**

Listed above.

**Quality:**

2

**Strengths And Weaknesses:**

Strengths:
1. The paper is well written!
2. The methods proposed to model the LLMs output is very interesting and novel to me.

Weaknesses and Questions:
1. The paper talks about personalizing JITAIs or pJITAIs on line 39-40, yet none of the references (40 and 26) even mention or elaborate that term - they only talk about JITAIs. Can the paper please be edited to cite proper references?
2. Line 269 - What are the definition of these dimensions? While some may be apparent like encouragement, it is difficult to ascertain what severity means. Also, what does it mean to have encouragement of 1 - does it mean the prompt is discouraging or just not encouraging (there is a subtle difference between the two). Related - Table F.3.3: What does -0.9 correlation mean between optimism and encouragement?
3. The authors should be upfront that this paper assumes that there exists an LLM with nice properties like safety. I could only find a small one-liner wrt this in the Discussion section - while adoption of LLMs in these mHealth contexts are held back by the lack of safety guarantees.
4. Some of the details wrt experimentation should be moved into the main paper.
5. It would be interesting to see if the results are different with a different LLM. Especially for something like Figure 10.
6. Appendix E.3 - How is one supposed to interpret the VAE score?
7. Appendix E.3 - Why 1-10 scaling? Why not use low medium high instead? Given that messages are short in length, I personally do not expect much difference between messages of score 1 and of score 3 for a given prompt. The stochasticity combined with this gives a noisy Figure 10.
8. Appendix E.3 - Continued from above, seems like the VAE just learnt noise. Why not instead use IRR to assign labels, and let the VAE learn those labels corresponding to where those messages fall under the given category instead? The problem currently is that because it is learning noise, the VAE score is extremely unreliable.
9. Perception of a message is very "human", and it would differ across people. The entire point of Gen-AI is to deliver personalized intervention text. However using a single, population-level scoring does not reflect the heterogeneity in individual perceptions. The paper should address that, or acknowledge it.

---

> ### Author Rebuttal · Authors · 2025-07-30
>
> # Point 1: JITAIs vs pJITAIs
> >*The paper talks about personalizing JITAIs or pJITAIs on line 39-40, yet none of the references (40 and 26) even mention or elaborate that term - they only talk about JITAIs. Can the paper please be edited to cite proper references?*
>
> Just-in-Time Adaptive Interventions (JITAIs) are sequential decision frameworks designed to deliver support at moments of need. Personalized JITAIs (pJITAIs) form a subclass that use RL agents to learn decision rules online. [26] details the HeartSteps (V2) pJITAI; however, it does not explicitly use the term "pJITAI." We will add a citation which uses the term (e.g., https://www.ijcai.org/proceedings/2024/0805.pdf).
>
> # Point 2: Dimension Interpretation
> >*Line 269 - What are the definition of these dimensions? While some may be apparent like encouragement, it is difficult to ascertain what severity means. Also, what does it mean to have encouragement of 1 - does it mean the prompt is discouraging or just not encouraging (there is a subtle difference between the two). Related - Table F.3.3: What does -0.9 correlation mean between optimism and encouragement?*
>
> As described in Algorithm 7 (lines 1238-1246), we obtained the embeddings (for each style dimension $d$) using the following steps:
>
> * Generate prompts designed to elicit variation along style $d$, using a predefined template.
> * Query an LLM with each prompt to generate a text response.
> * Embed each response into a high-dimensional vector space using an embedding model (BERT).
> * Train a VAE on these embeddings to learn a 1D latent variable capturing variation along style $d$.
>
> This mapping serves as the style embedding function $f_d$. As described in Algorithm 6 (below line 1171), we used these $f_d$ functions to generate style embeddings for the simulated JITAI messages. Correlations refer to the empirical correlation between the embedding values among all the simulated JITAI messages.
>
> To elicit variation along a style dimension $d$, we crafted prompts that included a definition of $d$. For example, the prompt defining optimism appears on line 1258. Definitions for the remaining style dimensions are available in the code supplement (code/simulation_setup/utils.py); we will link to the corresponding GitHub repository in the main paper upon de-anonymization. When eliciting a style dimension, we aimed to span the full range of intensity, from strongly negative to strongly positive. E.g., for optimism, we prompted the LLM to generate pessimistic, neutral, and optimistic messages.
>
> We note the unsupervised nature of the VAE means it does not guarantee alignment between the sign of the latent variable and the intensity of the style dimension. For instance, in the case of optimism, the VAE empirically assigned positive values to pessimistic messages. In contrast, for encouragement, positive values corresponded to more encouraging messages (and an encouragement value of 1 would mean a highly encouraging message). This also helps explain the observed negative correlation between optimism and encouragement. While this inconsistency is acceptable in our current setting, where embeddings are used as internal agent features, future work focused on embedding construction could aim to enforce consistent interpretability across style dimensions.
>
> # Point 3: LLM Safety
> >*The authors should be upfront that this paper assumes that there exists an LLM with nice properties like safety. I could only find a small one-liner wrt this in the Discussion section - while adoption of LLMs in these mHealth contexts are held back by the lack of safety guarantees.*
>
> We agree earlier framing would help clarify the scope/focus of the paper. We will add a brief discussion in Section 1 to more clearly acknowledge the assumption that an LLM with desirable properties (e.g., safety) exists.
>
> # Point 4: Experimentation Details
> >*Some of the details wrt experimentation should be moved into the main paper.*
>
> We agree that a summary of how the style dimension embeddings were generated, along with clearer references to the relevant appendix sections, would help frame the simulation study and improve accessibility. We will revise Section 6 accordingly.
>
> # Point 5: LLM Sensitivity
> >*It would be interesting to see if the results are different with a different LLM. Especially for something like Figure 10.*
>
> We also ran simulations using prompts generated by Qwen, while keeping the original VAE-based embeddings trained on Llama-generated text. The results were similar to those in the main paper.
>
> When we plotted optimism ratings against the original VAE scores, the resulting curve closely matched Figure 10. This consistency across LLMs suggests that the VAE picked up a real signal, despite substantial noise. We see this as a promising direction for future work, particularly with custom VAEs and improved prompts to reduce noise.
>
> # Point 6: VAE Score Interpretation
> >*Appendix E.3 - How is one supposed to interpret the VAE score?*
>
> We address this in our response to Point 2. The VAE score reflects the intensity and direction of a semantic dimension, but the sign is not consistently aligned across dimensions. For example, positive scores for optimism and encouragement correspond to pessimistic and encouraging messages (resp.). This inconsistency stems from the unsupervised nature of the VAE, which does not constrain the direction of the latent variable. In future work focused on embedding construction, we would take steps to ensure sign consistency.
>
>
> # Point 7: Scaling Approach
> >*Appendix E.3 - Why 1-10 scaling? Why not use low medium high instead? Given that messages are short in length, I personally do not expect much difference between messages of score 1 and of score 3 for a given prompt. The stochasticity combined with this gives a noisy Figure 10.*
>
> Our goal was to generate text spanning a wide range of intensities along each style dimension. Since we ultimately wanted a continuous embedding, we opted for a finer-grained 1–10 input scale rather than coarse labels like "low," "medium," or "high." That said, the input score is only used to guide LLM generation, we discard it once the VAE embedding functions are trained.
>
> We agree that LLMs are unlikely to consistently reflect subtle distinctions between nearby scores (e.g., 1 vs. 3), and this likely contributes to the noise observed in Figure 10. As discussed in lines 1272–1277, some of this noise stems from erratic LLM behavior. In future work focused more directly on embedding construction, additional fine-tuning and more targeted prompt engineering could help reduce this variability.
>
>
> # Point 8: VAE Reliability
> >*Appendix E.3 - Continued from above, seems like the VAE just learnt noise. Why not instead use IRR to assign labels, and let the VAE learn those labels corresponding to where those messages fall under the given category instead? The problem currently is that because it is learning noise, the VAE score is extremely unreliable.*
>
> As discussed in our response to Point 5, the pattern (while noisy) persists across LLMs, which suggests the VAE is capturing *some* underlying signal. That said, we agree that the embeddings are not perfectly reliable. As noted in lines 1268-1272, precisely mapping text to a continuous style intensity is inherently challenging and likely requires domain-specific tuning. Developing more reliable embeddings is an important direction, but not the focus of this work. Our goal is to demonstrate the feasibility of the GAMBITTS framework, which relies on projecting high-dimensional treatments into a low-dimensional space. The alignment between prompted tone and VAE score helps interpret the example projection, despite the noise, but is not necessary to show the validity of the GAMBITTS framework at-large.
>
> We agree that alternative strategies, such as using IRR-based labels or coarser scoring schemes (as the reviewer suggested in Point 7), are promising directions for future work, particularly for projects that foreground the embedding construction itself.
>
> # Point 9: Embedding Heterogeneity
> >*Perception of a message is very "human", and it would differ across people. The entire point of Gen-AI is to deliver personalized intervention text. However using a single, population-level scoring does not reflect the heterogeneity in individual perceptions. The paper should address that, or acknowledge it.*
>
> We agree that accounting for heterogeneity in message perception is an important concern, especially in settings where the goal is to deliver personalized intervention text. As discussed in Point 1 of our responses to Reviewers 1gsV and tZeG, we consider two classes of approaches for robustness to embedding misspecification: $(i)$ approaches within the scope of the current paper, and $(ii)$ broader directions for future work. Many of these approaches address embedding heterogeneity as well, and we briefly describe these approaches below. Our response to Reviewer 1gsV contains more detailed descriptions.
>
> ### $(i)$ Methods Within the Current Framework
>
> #### $(i.1)$ Maintaining a Set of Candidate Reward Models
> *Labelled i.2 in earlier responses.*
>
> Maintain a set of candidate reward models, each potentially using different treatment embeddings. The agent can periodically switch to the model with best predictive performance. This allows the algorithm to adaptively tailor embeddings to individuals, as different models may provide better fit for different users.
>
> ### $(ii)$ Future Directions
>
> #### $(ii.1)$ Online Embedding Learning
> Learn treatment embeddings online, providing a data-driven approach to specifying $Z$ for each individual (lines 317–321).
>
> #### $(ii.2)$ Corralling Bandit
> An extension of the candidate model approach is to use a corralling bandit, where a master algorithm adaptively selects among reward models.
>
> We plan to include a *brief* discussion of these approaches in the supplement, as noted in our responses to Reviewers 1gsV and tZeG.

---

> ### Comment · Reviewer_XiMn · 2025-08-05
>
> I thank the authors for their detailed responses and clarifications. Please see below:
>
> > This mapping serves as the style embedding function $f_d$. As described in Algorithm 6 (below line 1171), we used these $f_d$ functions to generate style embeddings for the simulated JITAI messages. Correlations refer to the empirical correlation between the embedding values among all the simulated JITAI messages.
>
> > To elicit variation along a style dimension $d$, we crafted prompts that included a definition of $d$. For example, the prompt defining optimism appears on line 1258. Definitions for the remaining style dimensions are available in the code supplement (code/simulation_setup/utils.py); we will link to the corresponding GitHub repository in the main paper upon de-anonymization. When eliciting a style dimension, we aimed to span the full range of intensity, from strongly negative to strongly positive. E.g., for optimism, we prompted the LLM to generate pessimistic, neutral, and optimistic messages.
>
> > We note the unsupervised nature of the VAE means it does not guarantee alignment between the sign of the latent variable and the intensity of the style dimension. For instance, in the case of optimism, the VAE empirically assigned positive values to pessimistic messages. In contrast, for encouragement, positive values corresponded to more encouraging messages (and an encouragement value of 1 would mean a highly encouraging message). This also helps explain the observed negative correlation between optimism and encouragement. While this inconsistency is acceptable in our current setting, where embeddings are used as internal agent features, future work focused on embedding construction could aim to enforce consistent interpretability across style dimensions.
>
> It would greatly benefit the reader if the definitions are not limited to the code and are also put in the appendix (even a table, just stating what 1 and 10 meant in the prompt along a dimension). The same goes for the last paragraph here, explaining the sign misalignment and scope for future work.

---

> > ### Author Response · Authors · 2025-08-05
> >
> > Thank you for your thoughtful review. We especially appreciate the attention you gave to the simulation study details in both the main text and the supplement.
> >
> > While lines 1255-1263 provide the style definition and explain the 0-10 scoring scheme in the context of optimism, we agree that presenting these details more generally would benefit the reader. We will revise the supplement to include a broader discussion of the style definitions and scoring, including a table summarizing the endpoints for the main style dimensions.
> >
> > In addition, we will expand the discussion on lines 1285-1287 to better clarify the potential for style/sign misalignment and outline directions for future work.

---

> > > ### Comment · Reviewer_XiMn · 2025-08-05
> > >
> > > I am satisfied with the responses and clarifications, and have adjusted my score accordingly.

---

### Official Review · Reviewer_q3H4 · 2025-06-30

**Clarity:** 2
**Significance:** 3
**Originality:** 3
**Rating:** 5
**Confidence:** 4

**Summary:**

This paper presents a Thompson sampling based algorithm for contextual bandits that interact with the environment through a generative model. The action taken by the algorithm is a prompt that is fed into the generative model. The response that is put into the environment is a sample from the generative model. They relate their approach to causal graphs in RL and provide theoretical guarantees for their approach under assumptions on the causal graph. They also empirically evaluate their approach and show that their method improves upon standard Thompson sampling empirically.

**Questions:**

- I found the Z and Z* discussion sometimes a bit hard to follow. After reading more closely I think I understand it, but it might help to better clarify what are the assumptions of your algorithm and what is an assumption on the data generating environment. For example, in Algorithm 1, I spent a some time thinking about in the integration in step 3 what distribution of z was being integrated over Z or Z*. I’m pretty sure it is Z, but might help to make this more overt for future readers.
- When I was reading, in step 3 of algorithm 1, isn’t the action space (the space of all possible prompts) huge? I feel there needs to be more acknowledgement of the computational challenges in the main text (even if you say you will discuss it further in a future section), as it made me think either this algorithm is not practical or I am misunderstanding what an example action space could be. Later in algorithm 1, there is X_t being used to select actions. Could you clarify this? Also is this $X_t$ or $X_t^G$?
- Could you talk a bit more about what “context is sent to generator”? It appears this is not learned by the algorithm but pre-specified?

**Ethical Concerns:**

["NO or VERY MINOR ethics concerns only"]

**Final Justification:**

I thank the authors for their detailed response. I feel their response has clarified many of the questions that I had and I have accordingly raised my score.

**Limitations:**

I feel the authors could have a more thorough discussion of the limitations of their algorithm, e.g., when it is appropriate to apply versus not. As well as discuss computational challenges of implementing their algorithm further.

**Quality:**

3

**Strengths And Weaknesses:**

## Strengths
- The problem tackled by this work is well-motivated and timely.
- The introduction and problem motivation was very well-written
- The algorithm seems relatively straightforward to understand and has both theoretical guarantees (under assumptions) and performs well empirically in simulations

## Weaknesses
- I found the “Problem Formulation and Notation” section 3 hard to follow. I feel it would benefit from a rewrite that discusses what all the variables are in a specific problem setup. For example, I am not sure exactly what the context $X_t$ is here (sometimes I thought this was the prompt). Then I wasn’t clear why the action space was finite (as I thought the prompt space could be very large). Should I be thinking about $A$ as a policy and $G$ as a sample from that policy? The general problem setup is still confusing for me.
- The biggest weakness I see (which could be clarified) is why the action does not depend on the context $X$ in Figure 1. It seems like it is a contextual bandit problem, but the algorithm is not allowed to use the context to select actions, which seems odd.

---

> ### Author Rebuttal · Authors · 2025-07-30
>
> # Point 1: Clarity of Problem Formulation and Notation
> >*I found the “Problem Formulation and Notation” section 3 hard to follow. I feel it would benefit from a rewrite that discusses what all the variables are in a specific problem setup. For example, I am not sure exactly what the context $X_t$ is here (sometimes I thought this was the prompt). Then I wasn’t clear why the action space was finite (as I thought the prompt space could be very large). Should I be thinking about $A$ as a policy and $G$ as a sample from that policy? The general problem setup is still confusing for me.*
>
> We begin by clarifying the notation.
>
> * $X_t$ - User context at time $t$. This could include any static user covariates (e.g., gender, race/ethnicity), as well as time-varying characteristics (e.g., steps taken the previous day, current location).
> * $A_t$ - The query sent to the generator at time $t$. This is the message the agent sends to the generator that requests a certain type of output (e.g. "Please write a message that encourages the user to walk more.").
> * $\mathcal{A}$ - The set of possible queries to send to the generator. This set is finite and pre-specified.
> * $G_t$ - The generator's response. This is output of the generator given the query $A_t$ and context $X_t$.
> * $Z_t$ - The working low-dimensional embedding of the full generator output $G_t$. This is used for modeling $Y_t$.
> * $Y_t$ - The reward at time $t$.
>
> Using LLM-assisted mHealth interventions as an illustrative example, a GAMBIT environment sees the following at time $t$:
>
> * The agent observes user context $X_t$ (e.g., demographic information and steps taken the previous day).
> * The agent selects query $A_t$ and sends it to the generator, along with the user information $X_t$. (E.g., "Please write a message that encourages the user to walk more. The user is male and took 10,000 steps yesterday.")
> * The LLM outputs a message $G_t$ (e.g., "You're doing great! Let's see if you can take even more than 10,000 steps today!"), which is then sent to the user.
> * The mHealth application records a user outcome (e.g., daily steps), which is stored as reward $Y_t$.
>
>
> We are considering $\mathcal{A}$ to be a finite set of prompts that are determined before the GAMBITTS algorithm is deployed; i.e., we are not assuming $\mathcal{A}$ is the set of all possible prompts anyone could send to a generator.
>
> One could think of $A$ as the chosen action and $G$ as a noisy representation of that action that is sent to the environment. In the GAMBIT framework, the agent can only exert direct control over $A$, not $G$ (even though the environment experiences $G$, not $A$). Subsequently, the policy concerns choices of $A$.
>
>
> Lastly, we agree that adding a concrete example (as above) could help clarify Section 3. We plan to revise the manuscript accordingly.
>
>
> # Point 2: Action Dependence on Context
> >*The biggest weakness I see (which could be clarified) is why the action does not depend on the context $X$ in Figure 1. It seems like it is a contextual bandit problem, but the algorithm is not allowed to use the context to select actions, which seems odd.*
>
>
> Indeed, **GAMBITTS algorithms consider context at time $t$ ($x_t$) when selecting actions**. For example, Step 4 of Algorithm 1 (foGAMBITTS, line 149) states:
> $$\textbf{Set } A\_t = \displaystyle\text{argmax}\_{a'\in\mathcal{A}} \mathbb{E}\_{\theta_{1,t}, \theta_{2,t}}\left[Y\mid a', x_t\right].$$
>
> Here, the foGAMBITTS agent selects the action which maximizes expected reward (under the sampled $\theta$ parameters from Step 2) *given* current context $x_t$. Furthermore, Step 4 of Algorithms 2 and 3 (poGAMBITTS and ens-foGAMBITTS, respectively), as well as Step 3 of Algorithm 5 (ens-poGAMBITTS) involve similar action selection.
>
> Figure 1, however, depicts the causal dynamics of the GAMBIT data-generating process, not the logic of the decision algorithm. The proposed GAMBITTS algorithms (which do incorporate context) are external to the causal model. Accordingly, we omit arrows from $X$ to $A$. We recognize that conventions may differ on this point and will add a note to the Figure 1 caption clarifying the absence of this arrow.
>
>
>
> # Point 3: $Z$ versus $Z^*$
> >*I found the Z and $Z^\ast$ discussion sometimes a bit hard to follow. After reading more closely I think I understand it, but it might help to better clarify what are the assumptions of your algorithm and what is an assumption on the data generating environment. For example, in Algorithm 1, I spent a some time thinking about in the integration in step 3 what distribution of z was being integrated over $Z$ or $Z^\ast$. I’m pretty sure it is Z, but might help to make this more overt for future readers.*
>
>
> In the GAMBIT framework, the generator produces content $G$, which is delivered to the user, who then responds with reward $Y$. In practice, modeling $Y\mid G$ is intractable due to the high dimensionality of $G$. To address this, we project $G$ into a low-dimensional space, taking $Z:=h(G)$, and model $Y\mid Z$. This working model is used throughout for reward prediction and action selection.
>
> As discussed in lines 126-131, GAMBITTS performance depends on how well $Y\mid Z$ approximates $Y\mid G$. Poor embeddings (where $Y\mid Z, X$ is a bad proxy for $Y\mid G, X$) can degrade performance relative to embeddings that capture more reward-relevant signal.
>
> With this in mind, we introduce $Z^\ast$ as a conceptual benchmark: the ideal low-dimensional embedding that captures all reward-relevant information from $G$. It is used solely in the theoretical analysis to characterize regret and establish performance guarantees.
>
> All algorithmic steps (including the integration in Step 3 of Algorithm 1) operate on the observed working embedding $Z$, not $Z^\ast$. Since the agent has no access to $Z^\ast$, it plays no role in implementation.
>
> We agree that clarifying this point can help avoid confusion, and will revise the introduction to Section 4 in response
>
>
>
>
> # Point 4: Size of Prompt Space
> >*When I was reading, in step 3 of algorithm 1, isn’t the action space (the space of all possible prompts) huge? I feel there needs to be more acknowledgement of the computational challenges in the main text (even if you say you will discuss it further in a future section), as it made me think either this algorithm is not practical or I am misunderstanding what an example action space could be. Later in algorithm 1, there is X_t being used to select actions. Could you clarify this? Also is this $X_t$ or $X_t^G$?*
>
> As discussed in Point 1, we consider $\mathcal{A}$ to be a finite set of prompts specified prior to deploying the GAMBITTS algorithm. However, as shown in Section 6.3 (lines 298-309), poGAMBITTS regret does not materially increase with the size of $\mathcal{A}$. Thus, in settings with simulation access to the generator, the size of the action space can be quite large without sacrificing algorithmic performance. However, the computational cost does scale linearly with the number of arms. We agree this is worth noting and will revise Section 6.3 to include a comment on computational scaling.
>
> Regarding action selection: as discussed in Point 2, the full context $X_t$ is used to select actions. We discuss the distinction between $X_t$ and $X_t^G$ in our response to Point 5.
>
>
>
>
> # Point 5: Context Sent to Generator
> >*Could you talk a bit more about what “context is sent to generator”? It appears this is not learned by the algorithm but pre-specified?*
>
> We take $X_t$ to represent the full observed context for the user at time $t$. This might include static variables (e.g., demographics) and time-varying ones (e.g., prior behavior). While some components of $X_t$ may be useful for outcome modeling, they may not be appropriate for inclusion in the prompt sent to the generator. For example, a variable like weight could improve reward prediction but would likely be excluded from the generator input for ethical or design reasons.
>
> To reflect this, we consider interventions which send a subset of user context to the generator (denoted $X_t^G\subseteq X_t$). We treat the choice of which features to include in $X_t^G$ as pre-specified, though learning this mapping online could be a promising direction for future work.
>
> Throughout the paper, we assume $X_t^G=X_t$ for simplicity and narrative clarity. The proposed algorithms require only superficial changes to accommodate $X_t\neq X_t^G$ (namely, the treatment model introduced below line 121, $f_1(z; A_t, X_t, \theta_1)$, becomes $f_1(z; A_t, X_t^G, \theta_1)$). Since this distinction does not affect the main development, we agree it is clearer to move discussion of $X_t^G$ to the supplement and will revise the manuscript accordingly.
>
>
> # Point 6: Appropriate Applications
> >*I feel the authors could have a more thorough discussion of the limitations of their algorithm, e.g., when it is appropriate to apply versus not. As well as discuss computational challenges of implementing their algorithm further.*
>
> As noted in Point 4, we will revise Section 6.3 to explicitly discuss how computational cost scales with the size of the action space. In response to Reviewer XiMn Point 3, we also plan to revise Section 1 to include a brief discussion of appropriate settings for algorithm deployment.

---

### Official Review · Reviewer_SpBA · 2025-07-01

**Clarity:** 3
**Significance:** 3
**Originality:** 3
**Rating:** 4
**Confidence:** 3

**Summary:**

In GenAI-powered interventions, the agent selects a query, but the environment experiences a stochastic response drawn from the generative model. Standard bandit methods do not explicitly account for this structure, where actions influence rewards only through stochastic, observed treatments. This paper introduces generator-mediated bandit - Thompson sampling (GAMBITTS), a bandit approach designed for this action/treatment split, using mobile health interventions with large language model-generated text as a motivating case study. GAMBITTS explicitly models both the treatment and reward generation processes, using information in the delivered treatment to accelerate policy learning relative to standard methods. The authors establish regret bounds for GAMBITTS by decomposing sources of uncertainty in treatment and reward, identifying conditions where it achieves stronger guarantees than standard bandit approaches. The empirical results show that GAMBITTS consistently outperforms conventional algorithms by leveraging observed treatments to more accurately estimate expected rewards.

**Questions:**

Please see the weaknesses above.

**Ethical Concerns:**

["NO or VERY MINOR ethics concerns only"]

**Final Justification:**

After reading the response and other reviewers' comments, I tend to weak accept.

**Limitations:**

Please see the weaknesses above.

**Quality:**

3

**Strengths And Weaknesses:**

Strengths:

1.	The studied problem, generator-mediated bandits, is related to large language models (LLMs) and may receive attention in the online learning and LLM community.
2.	The authors propose algorithmic instantiations for the GAMBITTS framework, and provide regret bounds for GAMBITTS.
3.	Empirical results show the limitations of standard bandit algorithms and the performance gains achieved by the proposed GAMBITTS approaches.

Weaknesses:

1.	The technical novelty in algorithm design and theoretical analysis is unclear. It seems that this paper just applies Thompson sampling to the proposed generator-mediated bandit formulation.
2.	The title mentioned “GenAI-Powered”, but it is unclear how the generator-mediated formulation or algorithm brings benefits to the decision-making process in theoretical results or experiments, compared to standard (no generator involved) bandit problem or standard Thompson sampling algorithm. The paragraph following Theorem 3 discusses the advantage when $d<<\sqrt{CK}$. But this seems an advantage brought by a low-dimensional representation. How can we understand the benefits brought by generator for the decision-making or learning process?
3.	The writing of this paper needs to be improved. The authors try to sell some big concepts like “GenAI”, but it is hard to understand the technical contributions of this paper.

---

> ### Author Rebuttal · Authors · 2025-07-30
>
> *We thank Reviewer SpBA for their time and feedback. In our response, we seek to clarify the setting of our paper. Subsequently, we address Point 2 before Point 1.*
>
> # Point 2: Generator Integration
> >*The title mentioned “GenAI-Powered”, but it is unclear how the generator-mediated formulation or algorithm brings benefits to the decision-making process in theoretical results or experiments, compared to standard (no generator involved) bandit problem or standard Thompson sampling algorithm. The paragraph following Theorem 3 discusses the advantage when $d<<\sqrt{CK}$. But this seems an advantage brought by a low-dimensional representation. How can we understand the benefits brought by generator for the decision-making or learning process?*
>
>
> The integration of the generator is motivated by applied settings where generative models are increasingly used to deliver personalized sequential interventions (see lines 21–24). In such cases, the planner selects prompts, but the treatment actually delivered is the generator’s output. This motivates the GAMBIT framework, where the generator mediates the action-to-treatment mapping.
>
> Our goal is not to show that introducing a generator improves the learning process; indeed, it *complicates* the action-to-reward feedback loop. Rather, we show how, *in a GAMBIT environment*, explicitly incorporating the generator’s role (via GAMBITTS) can make learning feasible and efficient.
>
> The theory compares standard Thompson sampling, which models the reward directly as a function of the prompt, with GAMBITTS, which accounts for the treatment generated in response. The referenced discussion around $d<<\sqrt{CK}$ (lines 254-255) pertains to regret bounds for fully online GAMBITTS relative to standard Thompson sampling, illustrating when the generator-aware approach can provide tighter guarantees in a GAMBIT environment. Earlier theoretical results discuss this comparison in the context of partially online GAMBITTS.
>
>
>
> # Point 1: Technical Novelty
> >*The technical novelty in algorithm design and theoretical analysis is unclear. It seems that this paper just applies Thompson sampling to the proposed generator-mediated bandit formulation.*
>
> Indeed, standard Thompson sampling can be directly applied in the GAMBIT setting by directly modeling reward $Y$ as a function of action $A$. However, as introduced in lines 27-33, and formalized in lines 102-106, the GAMBIT environment induces a separation between the chosen action and delivered treatment. Here, the action ($A$) represents the query sent to the generator. Meanwhile, the treatment delivered to the environment is the generator's response ($G$). While the agent can directly control their action (generator query), they cannot directly control the generated response. In this environment, standard Thompson sampling ignores the mediation by $G$ and directly models $Y\mid A$.
>
> Unlike standard Thompson sampling, GAMBITTS is designed specifically to handle this mediation by modeling the generator’s output and using that information in the decision-making process. Our formulation explicitly incorporates the generator mediation structure, allowing the agent to update beliefs over treatment effects rather than over prompt effects directly. Theoretical results (e.g., Theorems 1, 2, and 3, Corollary 1) and empirical comparisons in Section 6 and Appendix E show that GAMBITTS can outperform standard Thompson sampling under various data generative environments.
>
> Lastly, as discussed throughout Appendix B, there is limited prior work on Thompson sampling in bandit settings with mediated feedback, particularly in the presence of continuous mediators. To our knowledge, no existing Thompson sampling-based methods address the high-dimensional setting induced by generator-mediation or account for the complexities introduced by large-scale generative models producing treatments. GAMBITTS fills this gap.
>
>
> # Point 3: Writing
> >*The writing of this paper needs to be improved. The authors try to sell some big concepts like “GenAI”, but it is hard to understand the technical contributions of this paper.*
>
> Our use of "GenAI" reflects the motivating applications, where generative models are already being used to deliver sequential interventions (see lines 21–24). The goal of this paper is not to propose GenAI as a solution, but to analyze decision-making in settings where such models are already in use. In this sense, the contribution is algorithmic: we develop a method tailored to a structure that arises naturally in these interventions. To our knowledge, there are no existing bandit algorithms specifically designed for this generator-mediated setting.
>
> Regarding the quality of the writing, Reviewer q3H4 noted that the introduction and problem motivation were well written, and Reviewer XiMn expressed the same view about the paper overall. If there are specific passages that were unclear, we would be happy to revise them for clarity.

---

> > ### Comment · Reviewer_SpBA · 2025-08-05
> > **I raised my rating from 3 to 4**
> >
> > Thanks for the response. After reading the response and other reviewers' comments, I raised my rating from 3 to 4.

---

> > > ### Author Response · Authors · 2025-08-05
> > >
> > > Thank you for your time and effort in reviewing our paper.

---

### Official Review · Reviewer_tZeG · 2025-07-03

**Clarity:** 3
**Significance:** 3
**Originality:** 3
**Rating:** 4
**Confidence:** 3

**Summary:**

New contextual bandit model where the planner action is a query to an LLM which then goes to the environment (e.g., mHealth patient) potentially generating a reward. Relates to a broader literature where actions are stochastic, but with the ultra-high dimensionality of a language model. They give regret analysis and empirical results.

**Questions:**

What is the reward model of 6.3?

Why is the contextual TS a “more natural” comparison, and if so, why not include it in the main text figures?

Can you comment on steps one might take toward learning the mapping from G to Z, in either an offline or online way? This will be key to adapting the framework to new domains where planners may have less informed priors on reward structures.

Can you comment on the generality of GAMBITTS – other applications where, as you mention, there may be an interest in learning some randomized embedded action dimension… perhaps there is a connection to causal MDPs, where this is a known latent structure? https://proceedings.mlr.press/v177/lu22a/lu22a.pdf

Should add related work on LLMs and bandits… much study on the LLM as the agent
https://arxiv.org/html/2410.06238v1
https://arxiv.org/pdf/2403.15371

**Ethical Concerns:**

["NO or VERY MINOR ethics concerns only"]

**Final Justification:**

Thank you to the authors for the additional clarity and results. I'm keeping my borderline accept score, as I'm finding myself with additional questions to each response (all about the evaluation) that are leading me to believe there's more to dig into than can be clarified in the rebuttal phase... e.g., we've gone through multiple rounds of additional simulation -- more here than just clarifying questions about the original submission. That said, I feel there's still enough here in the model and theory to accept, but I'm not willing to fight for this submission, as it could certainly benefit from more work on the evaluation.

**Limitations:**

yes, authors have a nice paragraph on limitations and future work

**Quality:**

3

**Strengths And Weaknesses:**

Strengths:
GAMBITTS framework seems to be new and relevant, will likely be relevant to broad set of researchers
They provide algorithms for solving as well as theoretical analysis
They provide empirical analysis on real-world derived datasets
Related work is sufficient, good connections to the literature on bandits with randomized actions, good connections to literature on natural language as an action space.

Weakness
My main takeaway from this paper so far is that the most important thing a planner can do is learn a good projection from action space to embedding space before the policy execution/learning starts… which is a departure from the bandit part of the problem altogether. And further, the problem of learning the projection isn’t explored. Maybe this models some realistic scenario, but it does not capture what seem to be the richer parts of the problem from a methodological perspective. This is likely balanced by the value of introducing the formal model.
Evaluation is somewhat lacking. There is no exploration of a case where the reward is a function of context, despite the choice to model it. What is the reward model of 6.3? Why is the contextual TS a “more natural” comparison, and if so, why not include it in the main text figures?

---

> ### Author Rebuttal · Authors · 2025-07-30
>
> # Point 1: Mapping from $G$ to $Z$
> ## Point 1(a)
> >*The most important thing a planner can do is learn a good projection from action space to embedding space before the policy execution/learning starts*
>
> Certainly, the planner would benefit from a good projection from $G$ to $Z$, as evidenced by Figure 3 (Section 6.2). While these results show GAMBITTS can still outperform standard Thompson sampling under mild-to-moderate $Z$-misspecification, we agree that many applied scientists may wish to pursue modeling strategies robust to embedding misspecification. We briefly outline two directions for addressing this here: $(i)$ approaches within the scope of the current paper, and $(ii)$ broader directions for future work. **Furthermore, we refer readers to our response to Reviewer 1gsV Point 1 for a more in-depth discussion on these directions**, and note that we plan to include a brief supplement outlining these directions.
>
>
>
> ### $(i)$ Robustness Within the Current Framework
>
> #### $(i.1)$ Adding Prompt Indicator to Reward Model
> Below line 121, we discuss a model for expected reward $(m_2(Z_t, X_t; \theta_2))$. Given an embedding construct $Z$ and model $m_2$, we consider the modified form:
> $$\tilde{m}\_2(A\_t, Z\_t, X\_t; \beta^., \theta\_2)=\sum\_{a\in \mathcal{A} }\beta^a \mathbb{I}(A\_t=a) + m_2(Z\_t, X\_t; \theta\_2).$$
> This helps the agent hedge against embedding misspecification when the goal is online learning of the optimal action $A$.
>
> We revisited the simulation results in Section 6.2 using this hedged model. Brief summaries of these results can be found in our response to Reviewer 1gsV Point 1.
>
>
> #### $(i.2)$ Maintaining a Set of Candidate Reward Models
> A second approach is to maintain a set of candidate reward models, each potentially using different treatment embeddings. Periodically, the agent can evaluate these models based on predictive performance and switch to the one that best predicts observed outcomes. This allows the algorithm to recover from poor initial choices of $Z$ without requiring strong prior knowledge about which embedding will perform best.
>
>
> ### $(ii)$ Future Directions
>
> #### $(ii.1)$ Online Embedding Learning
> As noted in lines 317–321, a natural direction for future work is to learn individualized treatment embeddings online, providing a data-driven approach to specifying $Z$.
>
> #### $(ii.2)$ Corralling Bandit
> An extension of the candidate model approach is to use a corralling bandit, where a master algorithm selects among reward models based on performance. This allows the agent to adaptively shift across models, potentially combining their strengths while minimizing regret.
>
>
>
>
> ## Point 1(b)
> >*[Learning the mapping from $G$ to $Z$] is a departure from the bandit part of the problem altogether.*
>
>
> From an applied perspective, the goal of the embedding is to recover a model for $Y\mid Z$ that approximates $Y\mid G$, as discussed in lines 126-129. Broadly, this is fundamentally the outcome modeling central to Thompson sampling-based approaches. While this step becomes more challenging with high dimensional treatments (e.g., text), we do not view it as a departure from the bandit framework. Rather, the high-dimensional nature of the treatment (and subsequent inability to directly model $G$ to $Y$) introduces new modeling challenges and distinguishes our work from earlier mediated bandit approaches (see Appendix B).
>
>
> ## Point 1(c)
> >*The problem of learning the projection isn’t explored.*
>
> We aimed to propose a flexible framework that could accommodate the diversity of embedding strategies already in use (e.g., cited works on line 131). Accordingly, we focused on the development of the GAMBITTS model and intentionally kept the approach agnostic to embedding construction to ensure applicability across diverse techniques and modalities (e.g., text, sound, images).
>
> With this in mind, a focused study on embedding strategy would be a valuable contribution in its own right. However, in the meantime, there are several minor modeling choices analysts can adopt to improve robustness to misspecification. As discussed in Point 1(a), we plan to add a *brief* appendix outlining these strategies.
>
>
>
> ## Point 1(d)
> >*Can you comment on steps one might take toward learning the mapping from G to Z, in either an offline or online way?*
>
> As noted in lines 122–125, the appropriate embedding strategy is highly dependent on the application area. In domains like mHealth, planners often have access to predecessor trials or related intervention data, which can support offline construction of a working embedding. For example, the personalized HeartSteps study ([26]) was a second iteration of a JITAI development trial, and prior data could plausibly be used to inform treatment representations.
>
> In cases where such data are limited or the embedding is poorly aligned with outcomes, one can adopt more robust modeling choices as discussed in Point 1(a). Alternatively, embeddings may be adapted online, though the development of these online embedding strategies remains an important direction for future work (see Point 1(a) and lines 317–321).
>
>
>
>
> # Point 2: Evaluation
> ## Point 2(a)
> >*Evaluation is somewhat lacking. There is no exploration of a case where the reward is a function of context, despite the choice to model it.*
>
> We chose to omit context-dependent rewards in the main experiments to streamline the narrative and more directly highlight the benefits of incorporating observed mediation via text. However, we did run simulations where the reward depends on covariates, and the qualitative results were unchanged. As an illustrative example, we revisit the Figure 2 simulation (line 287) with reward depending on location, prior steps, and optimism. The table below reports point estimates and 95% Monte Carlo confidence intervals for cumulative regret at $t=1,000$. All GAMBITTS agents considered remain correctly specified under this setup.
>
>
> | Agent  | Original | With Covariates   |
> |:-------|:---------|:------------------|
> | Std TS: Contextual |    75.9 (74.0, 77.8)  | 76.4 (74.5, 78.2) |
> | Std TS |   70.4 (65.3, 75.5) |  71.3 (66.3, 76.4)  |
> | foGAMBITTS |  38.2 (37.8, 38.6)  |  34.2 (33.8, 34.5)  |
> | poGAMBITTS |  9.6 (9.1, 10.0)  |  8.8 (8.3, 9.2) |
>
> To improve transparency and clarify this point, we will revise the supplement to include a subset of these simulations with covariate-dependent rewards.
>
> ## Point 2(b)
> >*What is the reward model of 6.3?*
>
> It is a linear reward model in optimism, formality, and severity. I.e.,
> $$Y_t = \beta_0 + \beta^{optimism}Z_t^{optimism} + \beta^{formality}Z_t^{formality} + \beta^{severity}Z_t^{severity} + \varepsilon_t.$$
>
> Exact values for all coefficients are located in the code supplement, and we will link to the GitHub repository when producing a de-anonymized draft.
>
> ## Point 2(c)
> >*Why is the contextual TS a "more natural" comparison, and if so, why not include it in the main figures?*
>
> We described contextual Thompson sampling (TS) as a more natural comparison to foGAMBITTS because both approaches employ separate modeling for each action-context pair $(a,x)$: foGAMBITTS by modeling treatment representations separately for each pair, and contextual Thompson sampling by modeling outcomes separately. However, we chose not to include contextual TS in the main figures because it consistently underperformed relative to standard TS in our simulations, likely due to the limited role of context in the reward. Since it was not a competitive baseline, we omitted it to keep figures focused, as noted in lines 283-285.
>
> # Point 3: Generality of GAMBITTS
> >*Can you comment on the generality of GAMBITTS – other applications where, as you mention, there may be an interest in learning some randomized embedded action dimension. Perhaps there is a connection to causal MDPs, where this is a known latent structure?*
>
> We view any decision-making setup where prompts to a generative model yield high-dimensional treatments as falling naturally under the GAMBITTS framework. We discuss broad application potentials in lines 21-24. Other examples include LLM-guided robotic control (e.g., https://www.microsoft.com/en-us/research/articles/chatgpt-for-robotics/ and LLM-based protein design, where models generate novel proteins https://pmc.ncbi.nlm.nih.gov/articles/PMC10701588/#s4. GAMBITTS offers a potential tool for efficiently guiding such generation.
>
> Thank you for noting the connection to causal MDPs, as in Lu et al. (2022). We see two core similarities:
>
> * A causal chain of the form action $\to$ intermediate variable $\to$ reward, where the agent cannot intervene directly on the intermediate.
> * Regret reduction by leveraging the fact that the intermediate lies in a lower-dimensional space than the action space.
>
> Key differences include the nature of the two spaces. Lu et al. consider a combinatorial action space on discrete interventions, while GAMBITTS operates over a large prompt space inducing high-dimensional treatments with low-dimensional structure. GAMBITTS also adopts a two-stage modeling strategy absent from Lu et al.'s framework.
>
> # Point 4: Related Work on LLMs
> >*Should add related work on LLMs and bandits, there is much study on the LLM as the agent.*
>
> We viewed our primary methodological framing as grounded in mediated bandits and causal inference with text. This shaped our focus on connecting to literature on bandits with randomized actions and text as an action space, as noted by the reviewer. That said, we agree that recent work on integrating LLMs into bandit systems (particularly where the LLM acts as the agent) is relevant from an applied perspective. We will add a paragraph in Appendix B to discuss this line of work in more detail.

---

> > ### Comment · Reviewer_tZeG · 2025-08-06
> > **Response to authors**
> >
> > I appreciate the author response and additional experiments with the hedged model. The result, while expected, that hedging improves robustness, would be a nice insight to add to the experiments, and I encourage the authors to do so.
> >
> > Response to 1(b) makes sense -- I should clarify my point more. One key takeaway from the submission was that a user has the best chance of performing well by starting off with the correct guess of how actions map to the embedding space. And that spirit of "the biggest impact I can have on this problem is to have learned something offline" is a departure from the spirit of bandit literature, which is to learn online. I agree that there are still pieces of this problem left to learn (i.e., the reward function), but the biggest delta in regret comes from bringing a pre-specified model from offline, not from improvements in learning.
> >
> > Evaluation points were not well addressed -- streamlining the narrative is not a reason to omit an experiment that explores part of the model -- at least it needs to be included in the appendix. Moreover, why is contextual TS worse than standard TS in a contextual setting? This seems to indicate a problem with the experiment. It would be great if authors could clarify.

---

> > > ### Author Response · Authors · 2025-08-08
> > >
> > > We thank Reviewer tZeG for their time and effort put into reviewing our paper. We greatly appreciate their detailed feedback.
> > >
> > >
> > > ## Hedged Model
> > > We agree that the hedged model results, though expected, offer a valuable contribution to the discussion on robustness-oriented modeling strategies. We will revise the appendix to include this discussion.
> > >
> > > ## Relation to Bandit Literature
> > > Indeed, there is no substitute for a priori knowledge of a good action-to-embedding map. We also agree that one could view constructing such a mapping using offline data (as described in Point 1(d) of our original response) as a departure from a "pure" online learning framework. However, using offline information for online bandit and RL algorithms is an active frontier of research [1-5].
> > >
> > > Within this framework, the online design challenge shifts to how best to incorporate the offline information, and whether doing so leads to demonstrable improvements over "offline-free" baselines. We address both aspects in our theoretical framework and in the experiments presented in the submitted draft, as well as those discussed in Point 1(a) of our original response.
> > >
> > > ## Evaluation
> > >
> > > We will revise the appendix to include an additional section in Appendix E (Further Simulations) presenting simulation results under direct covariate dependence.
> > >
> > > To clarify the setup in the submitted draft, while rewards in our main experiments do not explicitly depend on context, they are stochastic functions of style dimensions, which themselves depend on both action and context (see Appendix F.1). In this sense, rewards in all simulations incorporate context implicitly (as rewards are functions of text messages which are functions of context and action). Furthermore, all GAMBITTS agents use context in estimating treatment distributions. That said, we agree that an active discussion of direct covariate influence in the reward model would be helpful to readers interested in implementing these methods. As discussed above, we will revise the appendix accordingly.
> > >
> > > Regarding the underperformance of contextual TS: Incorporating covariates/context requires more heavily parameterized modeling for the contextual TS agent, compared with its standard TS counterpart. When covariate effects are modest, this modeling burden introduces variance that hurts performance (a general challenge in contextual bandit learning [6]). Furthermore, the reward model in the original rebuttal lacked any interaction between optimism and context, so the optimal action did not vary across context. This dampened any potential gain from using contextual information. To revisit this comparison under more favorable conditions for contextual TS, we re-ran the simulations with two main changes:
> > >
> > > * We simplified the data-generating environment so that current location was the only dynamic context variable, removing previous steps. We then changed the contextual TS agent to use only current location as its context variable.
> > > * We modified the reward model to include an interaction between optimism and current location.
> > >
> > > As shown below, we now observe the expected advantage of contextual TS.
> > >
> > > | Agent         | Regret                |
> > > |:-------------------|:---------------------|
> > > | Std TS             | 690.7 (667.5, 713.8) |
> > > | Std TS: Contextual | 402.4 (372.7, 432.1) |
> > > | foGAMBITTS         | 681.5 (664.5, 698.6) |
> > > | poGAMBITTS         | 15.9 (15.4, 16.3)    |
> > >
> > > *Results are based on 100 Monte Carlo iterations with a time horizon of $T=10,000$, chosen to better assess long-run behavior of non-GAMBITTS agents. To reduce learning delay for non-GAMBITTS agents, we also lowered the reward noise relative to the original low-signal environment used in our original rebuttal.*
> > >
> > > We also observe foGAMBITTS struggles in this environment, consistent with the results in Appendix E.5.
> > >
> > > ## References
> > >
> > > [1] Changxiao Cai, T. Tony Cai, and Hongzhe Li. Transfer learning for contextual multi-armed bandits. *The Annals of Statistics*, 52(1), February 2024.
> > >
> > > [2] Joey Hong, Branislav Kveton, Manzil Zaheer, Yinlam Chow, Amr Ahmed, and Craig Boutilier. Latent bandits revisited. In *Advances in Neural Information Processing Systems*, volume 33, pages 13423–13433. Curran Associates, Inc., 2020.
> > >
> > > [3] Yuda Song, Yifei Zhou, Ayush Sekhari, Drew Bagnell, Akshay Krishnamurthy, and Wen Sun. Hybrid RL: Using both offline and online data can make RL efficient. In *The Eleventh International Conference on Learning Representations*, 2023.
> > >
> > > [4] Kevin Tan, Wei Fan, and Yuting Wei. Hybrid reinforcement learning breaks sample size barriers in linear MDPs. In *Advances in Neural Information Processing Systems*, volume 37. Curran Associates, Inc., 2024.
> > >
> > > [5] Andrew Wagenmaker and Aldo Pacchiano. Leveraging offline data in online reinforcement learning. In *Proceedings of the 40th International Conference on Machine Learning*, ICML’23. JMLR.org, 2023.
> > >
> > > [6] Aleksandrs Slivkins. Introduction to multi-armed bandits, 2019.

---

### Official Review · Reviewer_1gsV · 2025-07-03

**Clarity:** 3
**Significance:** 3
**Originality:** 3
**Rating:** 4
**Confidence:** 3

**Summary:**

This paper introduces GAMBITTS (Generator-Mediated Bandit–Thompson Sampling), a novel bandit framework that explicitly models environments where agent actions (queries/prompts) produce stochastic treatments (e.g., LLM-generated content), which in turn influence rewards (e.g., user response). Motivated by adaptive mobile health interventions, the authors formalize a generator-mediated causal structure, propose both fully online and partially online variants of Thompson Sampling algorithms (foGAMBITTS and poGAMBITTS), and offer theoretical regret bounds. They also propose ensemble extensions for nonlinear reward modeling and empirically demonstrate improved performance over standard bandits in simulations based on real-world mHealth studies.

**Questions:**

Could GAMBITTS be extended to incorporate uncertainty or feedback on the generator output quality (e.g., hallucination risk, toxicity)? In high-dimensional cases, could contrastive or self-supervised embedding learning methods be integrated to learn Z on the fly?

**Ethical Concerns:**

["NO or VERY MINOR ethics concerns only"]

**Limitations:**

While the authors have discussed the limitations of the methods, an additional discussion on the limitations of practical applications would be beneficial to enhance the paper's strength.

**Quality:**

3

**Strengths And Weaknesses:**

Strengths:
- GAMBITTS is well-motivated and grounded in causal reasoning, and both algorithmic and theoretical components are clearly articulated.
- The regret bounds offer insightful decompositions of uncertainty from treatments and rewards.
- While motivated by mHealth, the approach is broadly applicable to marketing, education, and other personalization domains involving LLMs.


Weaknesses:
- Performance critically depends on the quality of the treatment embedding (Z), yet embedding learning is left to future work. In practice, choosing or learning Z is nontrivial.
- While simulations are well-designed, evaluation on real-world GenAI-intervention data is not yet feasible and leaves empirical validation incomplete.
- The framework assumes the generator does not adapt or improve, which might limit its applicability in settings where models are updated over time.

---

> ### Author Rebuttal · Authors · 2025-07-30
>
> *We thank Reviewer 1gsV for their thoughtful comments on both the strengths of the paper and areas for improvement. We especially appreciated their questions about possible extensions (i.e., Points 4 and 5), which were interesting, insightful, and helped shape our thinking going forward.*
>
> # Point 1: $Z$ Embedding
> >*Performance critically depends on the quality of the treatment embedding (Z), yet embedding learning is left to future work. In practice, choosing or learning Z is nontrivial.*
>
> Indeed, performance does depend on quality of the treatment embedding ($Z$), as discussed in lines 126-129. Moreover, we show empirical evidence for this phenomenon in Figure 3 (Section 6.2), although these results suggest GAMBITTS can still outperform standard Thompson sampling under mild-to-moderate $Z$-misspecification.
>
> Our goal was to present an approach that is flexible with respect to embedding strategy, given that there is no consensus on how best to construct embeddings in settings with high-dimensional treatments such as text. To remain agnostic on this point, we did not include an extended discussion of embedding strategy. However, as discussed below, we agree this is an important applied concern and there are several minor modeling choices an analyst can make to improve robustness to embedding misspecification. We plan to add a *brief* appendix outlining these strategies.
>
> As mentioned above, we agree that choice of $Z$ is nontrivial and of material concern for applied scientists implementing the algorithm. We outline two directions for addressing this: $(i)$ approaches within the scope of the current paper that aim to improve robustness to misspecification, and $(ii)$ broader directions for future work. We take each in turn.
>
> ### $(i)$ Robustness Within the Current Framework
>
> #### $(i.1)$ Adding Prompt Indicator to Reward Model
> Below line 121, we discuss a model for expected reward given treatment embedding and user context $(m_2(Z_t, X_t; \theta_2))$. Given any embedding construct $Z$ and model $m_2$, consider the modified form:
> $$\tilde{m}_2(A\_t, Z\_t, X\_t; \beta^., \theta_2)=\sum\_{a\in \mathcal{A} }\beta^a \mathbb{I}(A\_t=a) + m\_2(Z\_t, X\_t; \theta\_2).$$
> When the goal is online learning of the optimal action $A$, this structure offers a hedge against misspecification of $Z$ and $m_2$. In particular, if $m_2$ fails to capture signal, its estimated contribution to $\tilde{m}_2$ will shrink towards zero. In that case, the $\tilde{m}_2$ model asymptotically behaves like a standard Thompson sampling agent that learns a reward distribution over actions directly. Conversely, if $Z$ and $m_2$ are well-specified, then the action $A_t$ provides no additional information beyond what is already captured through $m_2(Z_t, X_t; \theta_2)$. In this case, the estimated $\beta^.$ terms will shrink to zero and $\tilde{m}_2$ will converge to $m_2$. However, the agent must first learn that the embedding carries signal, potentially delaying efficient generalization early in learning.
>
> We revisited Figure 3 (Section 6.2) using hedged reward models (as described above). The table below reports point estimates and 95\% Monte Carlo confidence intervals for cumulative regret at $t=1,000$ for the revised hedged versions for each agent (*a: poGAMBITTS; b: foGAMBITTS*).
>
> | Agent       | Hedged Cumulative Regret    |
> |:---------------------|:----------------------------|
> | Std TS               | 70.4 (65.3, 75.5)  |
> | Clarity (a)          | 53.4 (52.5, 54.4)  |
> | Encouragement (a)    | 53.1 (52.1, 54.0)  |
> | Formality (a)        | 53.2 (52.2, 54.2)  |
> | Optimism (a)         | 52.5 (51.5, 53.5)  |
> | Severity (a)         | 52.5 (51.5, 53.4)  |
> | Clarity (b) | 56.5 (55.6, 57.3)  |
> | Encouragement (b)    | 59.4 (58.5, 60.3)  |
> | Formality (b)        | 55.6 (54.7, 56.5)  |
> | Optimism (b)         | 60.3 (59.3, 61.2)  |
> | Severity (b)         | 59.7 (58.9, 60.6)  |
>
> All poGAMBITTS agents perform similarly, as do all foGAMBITTS agents. While every GAMBITTS variant outperforms standard Thompson sampling in this setting, the well-specified models no longer exhibit the performance advantage they had in the original (unhedged) Figure 3. This supports the hypothesis that hedging reduces downside risk but limits gains under well-specified embeddings.
>
>
>
> #### $(i.2)$ Maintaining a Set of Candidate Reward Models
> A second approach is to maintain a set of candidate reward models, each potentially using different treatment embeddings. Periodically, the agent can evaluate these models based on predictive performance and switch to the one that best predicts observed outcomes. This allows the algorithm to recover from poor initial choices of $Z$ without requiring strong prior knowledge about which embedding will perform best.
>
>
> ### $(ii)$ Future Directions
>
> #### $(ii.1)$ Online Embedding Learning
> A natural direction for future work is to learn individualized treatment embeddings online, providing a data-driven approach to specifying $Z$ (lines 317–321). While we highlight online sufficient dimensionality reduction as one possibility, other approaches may also be effective.
>
> #### $(ii.2)$ Corralling Bandit
> An extension of the candidate model approach is to use a corralling bandit, where a master algorithm selects among reward models based on performance. This allows the agent to adaptively shift across models, potentially combining their strengths while minimizing regret.
>
>
> # Point 2: Evaluation on Real-World Data
>
> >*While simulations are well-designed, evaluation on real-world GenAI-intervention data is not yet feasible and leaves empirical validation incomplete.*
>
>
> We agree that empirical validation on real-world GenAI interventions is important but currently infeasible, as such systems are not yet deployed at scale (line 270). Our simulations are grounded in realistic settings inspired by the 2023 Intern Health Study (lines 271–272), following a common mHealth development pipeline in which algorithms are first evaluated in simulation prior to deployment. For example, the reBandit algorithm was introduced and evaluated entirely in simulation in [Ghosh et al. 2024](https://www.ijcai.org/proceedings/2024/0805.pdf), then later deployed in the MiWaves pilot study. Similarly, the Oralytics algorithm was first developed and tuned using simulation-based reward design, then deployed in a real-world trial [Trella et al. 2023](https://pmc.ncbi.nlm.nih.gov/articles/PMC10457015/). In line with these trajectories, we view this manuscript as part of the foundational algorithmic development needed to support future deployment of GenAI-powered adaptive interventions.
>
>
>
> # Point 3: Generator Drift
> >*The framework assumes the generator does not adapt or improve, which might limit its applicability in settings where models are updated over time.*
>
> Yes, we assume a static generator throughout (line 322). This reflects scenarios where models are siloed and fixed for privacy, regulatory, or engineering reasons. Of course, in many applications the generator may evolve over time. We agree that extending the framework to accommodate non-static generators is an important and interesting concern (lines 322-323). As before, we outline two directions for addressing this: (i) approaches within the scope of the current paper, and (ii) broader directions for future work.
>
> ### $(i)$ Approaches Within the Current Framework
> Within the current framework, there are two natural adjustments to better handle generator drift. First, in poGAMBITTS, the offline dataset used to estimate $f_1^{off}$ can be periodically refreshed by re-generating samples from the current generator. This provides a simple mechanism for accommodating changes in the treatment distribution without fully redesigning the learning algorithm. Second, in foGAMBITTS, the agent can weight recent observations more heavily when estimating the treatment model $f_1$, allowing the agent to adapt if the distribution of generated responses shifts over time.
>
> ### $(ii)$ Future Directions
> In some settings, we may want to deliberately adapt the generator over time to improve treatment quality. As noted in lines 322–323, one promising direction is online fine-tuning of the generator itself, using observed outcomes to improve the quality of the treatments it produces. In this framework, generator drift is not a challenge to be overcome, but can be an opportunity to develop more responsive and personalized intervention pipelines.
>
>
>
> # Point 4: Extensions for Feedback on Generator Output Quality
>
> >*Could GAMBITTS be extended to incorporate uncertainty or feedback on the generator output quality (e.g., hallucination risk, toxicity)?*
>
> Yes: this is an immediate and important direction for extension (lines 323–328). To address safety concerns such as hallucination or toxicity, one approach is to integrate ideas from the learning-to-defer literature, allowing the system to block or reject generated texts deemed too risky. This would naturally connect to the cited work on noncompliant bandits, where the action taken (no text sent) may differ from the action proposed (send the text solicited by prompt $a$). Moreover, we believe such approaches could be extended to incorporate user feedback as well.
>
>
> # Point 5: Extensions for On-The-Fly Embedding Construction
>
> >*In high-dimensional cases, could contrastive or self-supervised embedding learning methods be integrated to learn Z on the fly?*
>
> Yes, we see online learning of individualized treatment embeddings as a promising direction, as discussed in Point 1 and in lines 317–321. Our current focus has been on supervised approaches, such as sufficient dimensionality reduction. However, generator access opens the door to alternative strategies, including self-supervised and contrastive methods that do not rely directly on outcome signals. Exploring these approaches would be a natural focus for a future paper on embedding strategies.

---

### Note · Authors · 2025-08-14

We thank the reviewers for their time in reading our paper and rebuttal, and appreciate their comments regarding the well-motivated nature of the problem and its broad interest to various research communities. Based on the discussion, the only point that may still need clarification from us concerns the simulation results presented in our response to Reviewer tZeG.

As discussed in the submitted draft and our response to Reviewer XiMn, we simulated rewards as stochastic functions of text embeddings. While user context influenced the text embeddings, the rewards in the submitted draft had no additional direct dependence on context. As noted in our response to Reviewer tZeG, we will add simulations to Appendix E where context directly influences both text and reward.

Furthermore, in our discussion with Reviewer tZeG, we presented simulation results showing that GAMBITTS agent performance is not materially affected by the presence of direct covariate influence on reward. Reviewer tZeG asked why the contextual Thompson sampling (TS) agent underperformed relative to the standard TS agent even when context influenced the reward. This was due to the particular data generative structure, in which the optimal action was unaffected by context. To illustrate this point, we considered a modified data-generative model with strong interaction effects between text embeddings and context. In this second experiment, we observed the expected advantage of the contextual TS agent. (*Cumulative regret is not directly comparable between these two experiments due to differing data-generating models and time horizons.*)

Lastly, the second experiment used low reward noise to better study the contextual and standard TS agents. As suggested in our theory and Appendix E.5, foGAMBITTS struggled in this setting. We ran an additional simulation increasing the reward noise; as expected, foGAMBITTS improved relative to the TS agents. Because Reviewer tZeG did not ask about foGAMBITTS performance, we did not include this table in the original discussion. We present it here to preempt any potential misunderstandings.

| Agent | Regret Under Increased Reward Noise |
|:-------------------|:---------------------|
| Std TS | 1159.4 (1129.0, 1189.9) |
| Std TS: Contextual | 1106.9 (1068.5, 1145.4) |
| foGAMBITTS | 1033.5 (964.1, 1102.9)  |
| poGAMBITTS  | 745.0 (665.7, 824.3) |

---

### Decision · Program_Chairs · 2025-09-17

**Decision:**

Accept (poster)

**Comment:**

The paper formalizes generator-mediated bandits and proposes Thompson-sampling algorithms that explicitly model the action→generated-treatment→reward pipeline, with ensemble extensions and explicit randomization probabilities for causal evaluation. It introduces a partially-online variant that pretrains the treatment model via generator simulation, which the authors argue improves online performance in mobile health scenarios.   Reviewers note that performance hinges on the quality of the treatment embedding, whose learning/choice is largely deferred.   They also point to limited real-world validation and the assumption of a static (non-drifting) generator as key weaknesses. To strengthen the paper, I recommend adding a crisp limitations/threats-to-validity section (when the proposed approach helps vs. fails), run ablations across embedding families/sizes with on-the-fly or corralling baselines, and report sensitivity to embedding misspecification.